In this file, we included clean version and marked up manuscript for tc-2018-245 entitled with "Comparison of ERA5 and ERA-Interim near surface air temperature, snowfall and precipitation over Arctic sea ice: Effects on sea ice thermodynamics and evolution"

# Comparison of ERA5 and ERA-Interim near surface air temperature, snowfall and precipitation over Arctic sea ice: Effects on sea ice thermodynamics and evolution

Caixin Wang[1, 2], Robert M. Graham[2], Keguang Wang[1], Sebastian Gerland[2], Mats A. Granskog[2]

[1]Norwegian Meteorological Institute, 9293 Tromsø, Norway

[2]Norwegian Polar Institute, Fram Centre, P.O.Box 6606 Langnes, 9296 Tromsø, Norway

*Correspondence to*: Caixin Wang (caixin.wang@npolar.no)

**Abstract.** Rapid changes are occurring in the Arctic, including a reduction in sea ice thickness and coverage and a shift towards younger and thinner sea ice. Snow and sea ice models are often used to study these ongoing changes in the Arctic, and are

typically forced by atmospheric reanalyses in absence of observations. ERA5 is a new global reanalysis that will replace the widely used ERA-Interim (ERA-I). In this study, we compare the 2 m air temperature (T2M), snowfall (SF) and total precipitation (TP) from ERA-I and ERA5, and evaluate these products using buoy observations from Arctic sea ice for years 2010 to 2016. We further assess how biases in reanalyses can influence the snow and sea ice evolution in the Arctic, when used to force a thermodynamic sea ice model. We find that ERA5 is generally warmer than ERA-I in winter and spring (0-1.2

°C), but colder than ERA-I in summer and autumn (0-0.6 °C) over Arctic sea ice. Both reanalyses have a warm bias over Arctic sea ice relative to buoy observations. The warm bias is smaller in the warm season, and larger in the cold season, especially when the T2M is below -25°C in the Atlantic and Pacific sectors. Interestingly, the warm bias for ERA-I and new ERA5 is on average 3.4 °C and 5.4 °C (daily mean), respectively, when T2M is lower than -25 °C. The TP and SF along the buoy trajectories and over Arctic sea ice is consistently higher in ERA5 than in ERA-I. Over Arctic sea ice, the TP in ERA5

is typically less than 10 mm SWE greater than in ERA-I in any of the seasons, while the SF in ERA5 can be 50 mm SWE higher than in ERA-I in a season. The largest increase in annual TP (40-100 mm) and SF (100-200 mm) in ERA5 occurs in the Atlantic sector. The SF to TP ratio is larger in ERA5 than in ERA-I, on average 0.6 for ERA-I and 0.8 for ERA5 along the buoy trajectories. Thus, the substantial anomalous Arctic rainfall in ERA-I is reduced in ERA5, especially in summer and autumn. Simulations with a 1D thermodynamic sea ice model demonstrate that the warm bias in ERA5 acts to reduce

thermodynamic ice growth. The higher precipitation and snowfall in ERA5 results in a thicker snow pack that allows less heat loss to the atmosphere. Thus, the larger winter warm bias and higher precipitation in ERA5, compared with ERA-I, on ice growth result in thinner ice thickness at the end of growth season when using ERA5, however the effect is small during the freezing period.

# 1 Introduction

The Arctic has been undergoing substantial changes in the recent decades. The decline of Arctic sea ice is seen as one of the most prominent indicators of Arctic climate change (Stroeve et al., 2012). The extent and area of the Arctic sea ice has decreased (Comiso et al., 2008), the length of the sea ice melt season is increasing (Markus et al., 2009; Mortin et al., 2014;

Stroeve et al., 2014; Mortin et al., 2016; Stroeve and Notz, 2018), and large areas of thick multi-year ice (MYI) have been replaced by thinner and more dynamic first-year ice (FYI) (Maslanik et al., 2011; Lindsay and Schweiger, 2015; King et al., 2017). The Arctic is warming more than twice as fast as the global average temperature over the past 50 years (Bekryaev et al., 2010; AMAP, 2017). The fastest warming in the Arctic occurs during the fall and winter season (Graversen et al., 2008; Boisvert and Stroeve, 2015), and is driven in part by an increased number of storms that bring warm winds from the south

(Woods and Caballero, 2016; Dahlke and Maturilli, 2017; Graham et al., 2017a, 2017b; Rinke et al., 2017). The additional heat and moisture carried by these storms could contribute to a reduction in the winter ice growth (Woods and Caballero, 2016; Alexeev et al., 2017; Stroeve et al., 2018).

Despite the rapid ongoing changes in the Arctic, there are relatively few direct observations of the atmosphere, sea ice and ocean conditions, especially during winter. Due to the lack of in-situ observations, most studies documenting changes in the

Arctic rely heavily on atmospheric reanalyses (Screen and Simmonds, 2010; Kapsch et al., 2014; Woods and Caballero, 2016; Sato and Inoue, 2017). In addition, reanalyses are also frequently used to force snow and sea ice models (Schweiger et al., 2011; Merkouriadi et al., 2017; Stroeve et al., 2018). However, there are inherent biases and uncertainties within these reanalyses, and large differences can exist among the different products (Tjernstöm and Graversen, 2009; Decker, et al., 2012; Jakobson et al., 2012; Lindsay et al., 2014; Wesslén et al., 2014; Graham et al., 2017b). Thus the choice of reanalysis, and

inherent biases within that product, will ultimately influence the simulation of Arctic sea ice mass balance (Cheng et al., 2008; Wang et al., 2015).

The European Centre for Medium-range Weather Forecasts (ECMWF) reanalysis product, ERA-Interim (ERA-I, Dee et al., 2011), has been widely used for studying changes in the Arctic and forcing ocean and sea ice models (e.g., Cheng et al., 2008; Maksimovich and Vihma, 2012; Kapsch et al., 2014; Woods and Caballero, 2016; Graham et al., 2017b). In 2017, the ECMWF

released a new reanalysis ERA5 (Hersbach and Dee, 2016). There are several major improvements in ERA5 compared with ERA-I, including much higher spatial and temporal resolutions, and more consistent sea surface temperature and sea ice concentration (Hersbach and Dee, 2016). Evaluations of the performance of ERA5 have been conducted over the land and revealed a higher performance of ERA5 than ERA-I (Albergel et al., 2018; Urraca et al., 2018), and other commonly used reanalysis, such as, MERRA-2 (the second version of the Modern-Era Retrospective Analysis for Research and Applications)

(Olausen, 2018; Urraca et al., 2018). However, the performance of ERA5 over Arctic sea ice is yet to be fully investigated.

In this study, we compare and evaluate the performance of ERA-I and ERA5 over Arctic sea ice. For this, we use data from Ice Mass Balance buoys (IMB) (Perovich et al., 2018) and Snow Buoys (Grosfeld et al., 2016; Nicolaus et al., 2017) deployed in 2010 to 2015. The buoys record position, the 2 m air temperature (T2M), mean sea level pressure (MSLP), and snow depth

at regular intervals. Hence, these observations can be used to evaluate the variables of T2M, precipitation and MSLP in the reanalyses. The former two variables are critical parameters for sea ice simulation (Cheng et al., 2008; Wang et al. 2015), and form the focus of our study. We use the T2M and snow depth observations from these buoys to assess the performance of ERA5 and ERA-I over Arctic sea ice. We further use the reanalyses to force a 1-D thermodynamic sea ice model. The simulations are compared with snow and ice thickness observations from the buoys to evaluate how differences in the T2M and precipitation influence the evolution of sea ice in the model.

## 2 Materials and Methods

### 2.1 Buoy data

IMBs autonomously measure thermodynamic changes in sea ice mass balance (Richter-Menge et al., 2006; Polashenski et al., 2011). They are part of a network of drifting buoys over the Arctic Ocean that provide meteorological and oceanographic data for real-time operational requirements and research purposes (Rigor et al., 2000). These instruments typically record GPS position, T2M and mean sea level pressure (MSLP) at hourly intervals, as well as temperature profiles through the air, snow, ice, and upper-ocean, and distances to snow/ice surface and ice bottom at four hour intervals. Snow depth and ice thickness can be estimated from the distances measured by acoustic sounders, if the initial thickness of snow and ice are known when the IMB is deployed (Wang et al., 2013). If the acoustic sounders fail but the temperature string works, the positions of the ice surface and bottom can be determined from the temperature readings. Similar to IMBs, Snow Buoys also record GPS position, T2M, MSLP, and snow depth at hourly intervals (Grosfeld et al., 2016; Nicolaus et al., 2017). However, Snow Buoys do not measure temperature profiles, and provide no information on ice thickness.

Since 2000, a large number of IMBs have been deployed across the Arctic, in regions such as the Central Arctic, the Beaufort Sea, the Chukchi Sea, the Laptev Sea, the North Pole, Canadian Islands and Svalbard (Perovich et al., 2018) (http://imb-crrel-dartmouth.org/archived-data/). In this study, we use data from 13 IMBs deployed in these different regions between 2010-2015 (Fig. 1, Table 1). The IMBs were typically deployed in the Central Arctic during April/May, while deployments in the Beaufort, the Laptev, and Chukchi Seas generally took place in August/September (Fig. 1, Table 1). For additional coverage, we also use observations from 3 Snow Buoys deployed in 2015, two of which in the Laptev Sea and one in the Central Arctic (Table 1; Fig. 1) (http://www.meereisportal.de/en). For simplicity, hereafter we refer to IMBs and Snow Buoys as buoys.

### 2.2 ERA5 and ERA-I reanalysis data

ERA5 is the ECMWF's latest reanalysis product, and will replace the widely used ERA-I. The first batch of ERA5, covering the period 2010-2016, was released in July 2017. The entire ERA5 dataset, extending back to 1950 will be available for use in late 2019. ERA5 and ERA-I both have global coverage, with a horizontal spatial resolution of 80 km for ERA-I, and 31 km for ERA5. In the vertical, ERA5 resolves the atmosphere using 137 levels from the surface up to a height equalling 0.01 hPa,

and ERA-I uses 60 levels from the surface up to an equivalent height of 0.1 hPa. ERA5 provides hourly analysis and forecast fields, while ERA-I provides 6-hourly analysis and 3-hourly forecast fields. For the data assimilation, both apply 4-dimensional variational analysis (4D-var). ERA-I uses the Integrated Forecast System (IFS) "Cy31r2" 4D-Var, and ERA5 applies the newer IFS "Cy41r2" 4D-Var". ERA5 includes various newly reprocessed datasets, for example, OSI-SAFr, the reprocessed version

of the Ocean and Sea Ice Satellite Application Facilities (OSI-SAF) sea ice concentration is used (Hersbach and Dee, 2016), and recent instruments that could not be ingested in ERA-I. Many new parameters, such as 100 m wind vector, are available as part of the ERA5 output. For comparison and evaluation against buoy observations, ERA5 is bilinearly interpolated to the buoy positions, and ERA-I is first linearly interpolated to hourly data, and then bilinearly interpolated to the buoy positions. For comparison between ERA-I and ERA5 over the Arctic sea ice, the ERA-I data are first bilinearly interpolated to the grid

of ERA5, and then T2M is averaged by season, and total precipitation and snowfall are integrated over the season.

## 3 Comparison of reanalysis and buoys' near surface air temperature, snowfall and precipitation over Arctic sea ice

### 3.1 Spatial distribution of seasonal differences of ERA5 and ERA-I near surface temperature, snowfall and precipitation

Figure 2 shows the seasonal mean differences of T2M, total precipitation (TP) and snowfall (SF) between ERA5 and ERA-I

over Arctic sea ice during 2010-2015. We classify spring as March, April and May, summer as June, July and August, autumn as September, October and November, and winter as December, January and February. The seasonal mean ice extent is obtained from the monthly sea ice concentration from NOAA/NSIDC during 2010-2015 (Meier et al., 2017).

The difference in T2M between ERA5 and ERA-I varies with season (Fig. 2a-d). ERA5 is generally warmer (0-1.2 °C) than ERA-I in spring and winter, and colder (0-0.6 °C) than ERA-I during summer and autumn over Arctic sea ice. These

temperature differences are smaller during summer, but substantial during the other seasons. Near the North Pole, ERA5 is warmer than ERA-I in summer, but colder than ERA-I in winter. Whether warmer or colder, the differences between ERA5 and ERA-I are small (±0.4 °C) in this region.

ERA-I is known to be a relatively "dry" global reanalysis product in the Arctic compared with most other modern reanalyses (e.g. MERRA-2, CFSR, and JRA-55) (Lindsay et al., 2014; Merkouriadi et al., 2017; Boisvert et al., 2018). The TP

in ERA5 is typically less than 10 mm water equivalent higher than for ERA-I in all seasons over Arctic sea ice, with exception of the Atlantic sector in autumn, winter and spring where TP in ERA5 can be up to 30 mm water equivalent larger (Fig 2e, g, h). The patterns of seasonal TP over sea ice are very similar in ERA5 and ERA-I (Fig. S1), and with distinctly highest annual TP in the Atlantic sector (Fig. S2).

Snowfall is substantially higher in ERA5 than in ERA-I in all seasons (Fig. 2i-l), particularly in the Atlantic sector, where

SF is up to 50 mm SWE higher in spring, autumn and winter seasons (Fig. 2i-l). In summer snowfall is much larger in ERA5 in the central and eastern Arctic (30-50 mm SWE higher) (Fig. 2j). Thus the differences in the snowfall between ERA5 and ERA-I are much larger than for TP in all seasons except winter (Fig. 2 i-l vs. Fig. 2e-h, see also Fig. S2). The patterns of

seasonal snowfall over sea ice are very similar in ERA5 and ERA-I (Fig. S1). Annual SF has increased all across the Arctic, more than TP, especially in the Atlantic sector (>100 mm) and eastern Arctic (Fig. S2).

## 3.2 Comparison of reanalysis near surface temperature, snowfall and precipitation against buoy observations

Both ERA-I and ERA5 accurately capture the observed evolution of MSLP measured by each of the buoys (not shown). The
hourly difference between the reanalysis MSLP and observations is no more than a few hPa. Excellent agreements between observed MSLP in the Arctic and earlier reanalyses have been shown in previous studies (e.g, Makshtas et al., 2007), demonstrating that MSLP is well simulated in reanalyses. In the following, we will focus on near surface temperature, snowfall and total precipitation.

### 3.2.1 Evaluation of near surface temperature in ERA5 and ERA-I using buoy observations

Figure 3-4 and Figures S3-5 show time series of T2M from different buoys, and the corresponding T2M difference between ERA5 and ERA-I, and T2M differences between reanalyses and observations at the buoys' positions. The observed T2M reveals the pronounced seasonal cycle in the Arctic. Low temperatures persist through winter and spring, before approaching near 0ºC in late May or early June. Temperatures near 0 ºC, or occasionally over 0 ºC, continue during summer, before lower temperatures return in late August or early September and decrease further in autumn.

The T2M in ERA5 and ERA-I generally agree well, both with each other and the observations (Figs. 3-4 & S3-5). The reanalyses perform best for buoys 2013E (Fig. 3b), and 2012J (Fig. S5a), which were both deployed in the central Arctic, the former near the North Pole and the latter closer to the Laptev Sea (Fig. 1). On occasions, hourly differences of T2M between ERA5 and ERA-I can exceed 4 ºC (e.g., Fig. 4). The largest hourly T2M differences between the two reanalyses and between the reanalyses and observations (Fig. 3-4 & S3-5), are found during the coldest months (November–May). Specifically, both
reanalyses have a warm bias during these months. Previous studies have shown that warm biases in the Arctic are prevalent among most reanalysis products, particularly during the winter season (Beesley et al., 2000; Tjernstöm and Graversen, 2009; Lüpkes et al., 2010; Jacobson et al., 2012; Lindsay et al., 2014; Wesslén et al., 2014; Graham et al., 2017b). This is because weather forecast models and climate models struggle to accurately simulate strong stable boundary layers (Beesley et al., 2000; Tjernstöm and Graversen, 2009; Sotiropoulou et al., 2015; Graham et al., 2017b; Kayser et al., 2017; Biosvert et al., 2018).
Interestingly, we find a larger warm bias in the new ERA5 compared with ERA-I (Fig. 3-4 & S3-5, Table 2), despite the higher vertical resolution in ERA5. The root-mean-square deviation (RMSE) values are higher for ERA5 than for ERA-I (see Fig. 3-4 and Fig. S3-5), in the range of 1.1-3.7 °C for ERA-I and 1.7-4.6 °C for ERA5.

We note that the near surface air temperature in both reanalyses corresponds to a height of 2 m, while it is likely often measured by buoys at a lower height. The initial observation height might also decrease further as snow accumulates. During
winter, the lowest temperatures in the Arctic occur under stable conditions with a strong surface-based inversion, meaning that the temperature increases with height from the surface. Hence, the near surface warm bias in reanalyses may partly be attributed

to the difference in height with the observations (Vihma et al., 2014). A prescribed ice thickness of 1.5 m and no snow accumulation on top of sea ice were applied both in ERA5 and ERA-I. The miss representation of snow may affect the surface energy budget, physically leading to, for example, overestimated conductive heat flux from the ocean to the surface.

A scatterplot of ERA5/ERA-I vs. buoy T2M clearly reveals the temperature dependence of the warm bias in both reanalyses
(Fig. 5a). The data crowd together near the 1:1 line when the air temperature is near 0°C, but spread further above the 1:1 line when the air temperature is low, especially at air temperatures below -25°C. The temperature dependence of the warm bias is also demonstrated in Fig. 5b, which shows the relationship between the daily mean T2M differences with the temperature bins of 5 °C from -45 – +5 °C. When the T2M is below -25 °C, the daily mean difference between reanalysis and observation is higher than 2 °C, with ERA5 3.1 – 8.0 °C warmer than in buoys, and ERA-I 2.4 – 4.4 °C warmer than in buoys (Fig. 5b). For
air temperatures above -25 °C, the bias between reanalysis and buoys is smaller, with ERA5 and ERA-I both 0.75 °C warmer than the observations on average.

Figure 5c shows the bias and standard deviation (std) for the reanalyses for each month, based on the buoy observations, and the temperature difference between the reanalyses. The smallest biases, and the smallest T2M differences between ERA5 and ERA-I are found in the months between July and October (also refer to Fig. 3-4 & S3-5). ERA5 is typically warmer than
ERA-I (and has a larger warm bias) throughout the winter and spring, including June. However, ERA5 is colder than ERA-I (0.01-0.6 °C) and has a smaller bias from July to October (Fig. 5c). Hence, the warm bias in ERA5 is smaller than ERA-I in the warm season (July-October). ERA-I has a warm bias in the warm season, but the magnitude is smaller (< 0.8 °C) than the warm bias in the cold season (Fig. 5c). Similarly, ERA5 has a small warm bias during July and August (<1 °C), and a likely insignificant cold bias (< 0.2 °C) in September and October (Fig. 5c).
The performance of reanalysis near surface temperature varies with region over Arctic sea ice (Fig. 6, also refer to Fig. 2). According to the buoys' positions (Fig. 1), we define four regions in the Arctic: the Central Arctic (north of 86° N), and the Pacific sector (90° W – 150° E), the Atlantic sector (30° W – 60° E), and the Laptev Sea (60° E – 150° E). The later three sectors are south of 86° N. The ERA5/ERA-I near surface temperature performs best in the Central Arctic (Fig. 6a), and well in the Pacific sector (Fig. 6c). It performs well in the Atlantic sector when the T2M is above -25 °C, but poorly when the T2M
is below -25 °C (Fig. 6b). The performance of reanalysis near surface temperature in the Laptev Sea needs to be further investigated due to small number of observations in this region (Fig. 6d & 6h). However, there is also some seasonal bias in the availability of data from buoys in the different regions, largely due to when buoys are deployed in different regions of the Arctic and subsequent ice drift patterns.

### 3.2.2 Comparison of precipitation and snowfall from ERA5 and ERA-I along buoy drift trajectories

We next compare the cumulative total precipitation and snowfall in ERA5 and ERA-I in autumn and winter, along the drift trajectories of the buoys. We begin accumulation from 15 August onwards if the buoy was deployed before this date, or from

October if the buoy was installed after 15 August but before 1 October. We accumulate the precipitation until 30 April, or until the buoy stopped working if this occurred before 30 April (Table 1).

The accumulated total precipitation (TP) in ERA5 is higher than ERA-I for all buoys (Figs. 7 & S6-7, and Table 1), which is consistent with the seasonal difference in TP documented in section 3.1 (Fig. 2e-h). On average, the accumulated TP in ERA5 is 13.8 mm water equivalent larger than in ERA-I, with differences for the individual buoys ranging from 0.4 (buoy 2013E; Fig. 7b) to 31.9 mm water equivalent (buoy 2012D; Fig. 7a). This is in agreement with the seasonal differences between the reanalyses (Fig. 2e-h).

Similar to the accumulated TP, the accumulated snowfall (SF) in ERA5 is larger than in ERA-I (Figs. 7 & S6-7; Table 1). For buoys deployed near the North Pole that started accumulating on 15 August, the accumulated SF in ERA5 is typically much larger than for ERA-I (Fig. 7a-b & S6, S7a). In contrast, for buoys deployed in other regions, which started accumulating on 1 October, the accumulated SF in ERA5 is typically slightly higher than ERA-I (Fig. 7c-d & Figs. S7b-f).

The ratio of snowfall to total precipitation (SF/TP) in ERA5 and ERA-I along the buoy trajectories is shown in Fig. 7 and Fig. S4-5. A higher ratio means that more precipitation falls as snow. The ratio of SF/TP for the buoy trajectories ranges from 0.31 to 0.94 in ERA-I, and from 0.62 to 1.0 in ERA5, with consistently more precipitation falling as snow in ERA5. The SF/TP ratio for ERA5 increases on average by 0.28 for the buoys that started accumulating on 15 August compared with that in ERA-I. In contrast, the ratio of SF/TP usually is 0.1 higher in ERA5 than in ERA-I for buoys that started accumulating on 1 October. This means that a substantial fraction of precipitation falls as rain in ERA-I during autumn (August-September), but the same precipitation events in ERA5 are classified as snowfall. This difference in SF/TP ratio can help to explain why the accumulated SF in ERA5 is much greater than ERA-I for buoys deployed in August, but only slightly higher than ERA-I for buoys starting in October. The higher ratio of SF/TP in ERA5 than in ERA-I takes place in all seasons over the Arctic sea ice (Fig. 8a-d vs. Fig. 8e-h). The increase of SF/TP ratio in ERA5 is more significant in autumn (~0.2) and summer (~0.3-0.4) as we found along the buoy trajectories, and relatively small in winter (~0.1) and spring (~0.1-0.2). This indicates more precipitation falls as snow in ERA5 not only in autumn, but also in summer.

The low SF/TP ratio and thus larger fraction of rainfall in ERA-I is known to be anomalous, and is likely due to the cloud physics scheme used (e.g., Dutra et al., 2011; Leeuw et al., 2015). In ERA-I, the split between liquid and ice in clouds is determined diagnostically as a function of temperature from -23 to 0 °C, with ice-only only below -23 °C and liquid-only above 0 °C. In contrast, the IFS Cy41r2 used in ERA5 includes a prognostic microphysics scheme, with separate cloud liquid, cloud ice, rain and snow prognostic variables (Sotiropoulou et al., 2015; see also ECMWF IFS documentation – Cy41r2; https://www.ecmwf.int/sites/default/files/elibrary/2016/16648-part-iv-physical-processes.pdf). Our findings indicate that ERA5 has significantly less Arctic rainfall than ERA-I, particularly in autumn (Fig. 7, Figs. S6-7) and summer (Fig. 8b and Fig. 8f).

Evaluating the performance of precipitation products over the Arctic Ocean is a major challenge due to the lack of observations, and difficulty accurately measuring snowfall (e.g. Lindsay et al., 2014; Rasmussen et al., 2012; Sato et al., 2017;

Blanchard-Wrigglesworth et al., 2018; Boisvert et al., 2018; Webster et al., 2018). Here we compare the precipitation from ERA-I and ERA5 with snow depth measurements from the buoys (Table 1). For this comparison, snow depth from the buoys is converted to snow water equivalent (SWE) using a climatological monthly mean snow densities of 220-380 kg m$^{-3}$ (Warren et al., 1999). Caution must be taken here, as the buoys reflect point observations, while the reanalyses provide a grid cell average. Snow depth is known to have large variability even over relatively small spatial scales (Warren et al., 1999; Sturm et al., 2002; Liston et al., 2018). An unknown fraction of the true snow fall will also be lost through blowing snow into leads, which is not accounted for in our calculation.

The accumulated TP and SF from ERA-I and ERA5 are generally comparable with the observed SWE from buoys in most cases during the accumulation period (refer to Fig. 7 & Figs. 4-5), such as for buoy 2012H deployed in the Beaufort Sea (Fig. 7d). However, in several cases the accumulated TP and SF from ERA-I and ERA5 are considerably lower than the observed SWE from buoys, such as for buoy 2012D from mid-September (Fig. 7a). This may be caused by snow drifting up against the buoy structure, or reflect anomalously low precipitation in the reanalyses. In other cases, the accumulated TP and SF from reanalysis is higher the observed SWE from buoys during some periods (buoys 2013B (Fig. S6c) and s20 (Fig. S7e) or for the whole accumulation period (buoys 2011M (Fig. 7c), 2012L (Fig. 7c) and S29 (Fig. S7f). This could be caused by snow erosion/sublimation around the buoy, or reflect anomalously high precipitation in the reanalyses. By the end of the accumulation period, the accumulated TP/SF is larger on average 55.4/41.9 mm SWE for ERA-I and 66.5/62.8 mm SWE for ERA5 than the observed SWE of the snow pack along the buoy trajectories (see Table 1).

## 4 Influence of air temperature and precipitation on sea ice evolution during the freezing season

In this section, we evaluate the impact of different forcing products (ERA-I, ERA5, and the buoys) on sea ice evolution. We focus on the freezing/growth season, from 1 October to 30 April, when sea ice generally starts to grow after summer. This period corresponds to the time when the largest differences of T2M between ERA5 and ERA-I were found (Figs. 2-4). For this exercise, we focus on buoys 2011M, 2012H, 2012L, and 2012J that were deployed in late August/early September and operated for more than one year, covering a complete freezing season (Table 1). These buoys were installed on MYI or FYI in the central Arctic (buoy 2011M), the Beaufort Sea (buoy 2012H, buoy 2012L), or the Laptev Sea (buoy 2012J). When these buoys were installed, sea ice thickness was between 1-2 m for buoys 2011M, 2012H, and 2012J, while buoy 2012L had an ice thickness of 3.35 m (Table 1). Snow depth was typically a few centimetres of snow at deployment. We use these buoys to assess the impact of different forcing data on sea ice evolution. For our simple approach we apply the empirical cumulative freezing degree day (FDD) model, which accounts for differences in T2M, and a 1D sea ice model that also account for effects of precipitation/snowfall.

## 4.1 Assessing the sea ice evolution with freezing degree days (FDD): impact of temperature bias

The cumulative freezing degree days (FDD) model only needs air temperature as input and is often used to estimate sea ice growth ($\Delta h$) from zero (e.g., Huntemann et al., 2014; Lei et al., 2017). The sea ice growth is estimated based on Lebedev (Maykut, 1986), $\Delta h = 1.33 \sum (FDD)^{0.58}$, where $\sum FDD$ is daily average temperature below the freezing point of sea water (-1.8 °C), integrated over the time period from 1 October to 30 April.

The positive near surface air temperature bias in ERA5 and ERA-I results in a negative ice thickness bias at the end of the growth season. The cumulative FDD is smallest for ERA5 (Fig. S8, Table 2), corresponding to the largest warm bias in ERA5 during the freezing season. The differences in FDD between ERA5, ERA-I and buoys are large for buoys 2011M, 2012H and 2012L, but negligible for buoy 2012J. The ice growth is 0.08-0.12 m less, with a mean of -0.09 m for ERA-I T2M, and 0.13-0.20 m less, with a mean of -0.16 m for ERA5 T2M compared to when using near surface buoy temperatures (Table 2).

## 4.2 Assessing sea ice evolution with a 1D sea ice model HIGHTSI: impact of T2M and precipitation

HIGHTSI is a 1D high-resolution thermodynamic snow and ice model designed for process studies to resolve the evolution of snow/ice thickness and temperature profile. The snow and ice temperature regimes are solved by the partial differential heat conduction equations applied for snow and ice layers, respectively. The turbulent surface fluxes are parameterized taking the thermal stratification of the atmosphere surface layer into account. Downward short- and longwave radiative fluxes are parameterized based on the total cloud cover. The model has been extensively used in Arctic studies (e.g., Cheng et al., 2008; Cheng et al., 2013; Wang et al., 2015; Merkouriadi et al., 2017).

In this section we perform six sensitivity simulations on each of the four buoys to explore the impact of temperature and precipitation on snow and sea ice evolution (Table 3). In the first two simulations, SFI_T2MI and SF5_T2M5, we force HIGHTSI with the T2M, 10 m wind speed (V), relative humidity (Rh), total cloud cover (CN) and snowfall, from ERA-I and ERA5 (Fig. S9), respectively. In the next two simulations, TPI_T2MI and TP5_T2M5, we force the model with the total precipitation from the reanalyses, rather than the snowfall, and treat precipitation as snow only when T2M is below 0 °C. In the final two simulations, we evaluate the influences of T2M and precipitation on the sea ice evolution individually. Specifically, we replace the T2M from ERA-I in the TPI_T2MI run with the T2M from ERA5, and name this run TPI_T2M5. Similarly, we replace the TP from ERA-I, in the run of TPI_T2MI, with the TP from ERA5 for the TP5_T2MI run (see Table 3). For all of the simulations we apply a seasonally variant ocean heat flux according to McPhee et al. (2003), which is large in October (10-20 Wm$^{-2}$), and decreases to nearly zero from mid-November (see Fig S9). Snow-ice, an ice type formed at ice surface (e.g., Leppäranta, 1983), was recently found to significantly contribute to the Arctic sea ice mass balance in a region with thick snowpack on relatively thin ice (Granskog et al., 2017; Merkouriadi et al., 2017). A few (1.5-3) millimetres snow-ice formed only in the TPI_T2MI and TPI_T2M5 runs for buoy 2012J (with the lowest initial ice thickness of all buoys examined, Table 1). This is negligible for the total ice mass balance. Thus, the effect we examine solely depends on the differences in T2M and precipitation/snowfall on thermodynamic ice growth.

The pattern of snow accumulation recorded by many buoys is consistent with observations by Warren et al. (1999). Namely, they record snow accumulation in late fall, followed by a relatively constant snow depth from December/January–March, and sometimes a late increase in snow depth in early spring (Fig. 9). For example, the observed snow depth at buoy 2012H increased to about 0.25 m in late December, and changed marginally thereafter (Fig. 9a). Similarly, the observed snow depth at buoy

2012L increased from 0.03 m to 0.13 m from early October to mid-November, and then remained around 0.10 m until the end of April (Fig. 9c). Most buoys recorded an increase in ice thickness from early December to the end of the freezing season. For example, the sea ice growth for buoy 2012H began in early December, at a rate of approximately 0.5 cm/d, until late March, and afterward the growth became sluggish at a rate of 0.16 cm/d until the end of April (Fig. 9a). However, buoy 2012L, which had an initial ice thickness of ~ 3.3 m, showed no significant growth until early February, before undergoing a slight

increase from around 3.3 m to 3.42 m by the end of the freezing season (Fig. 9c). Sea ice growth for buoy 2011M (Fig. 9b) and 2012J (Fig. 9d) showed a staircase pattern since the ice thickness was derived from measured temperature profile due to the failure of acoustic sounders as mentioned in section 2.1.

We first compare the simulations TPI_T2MI and TP5_T2M5. Differences in the ice thickness at the end of the growth season for these simulations are relatively small, despite the larger warm bias in ERA5 (Fig. 9). Sea ice was marginally thinner

(0.006-0.02 m) in TP5_T2M5 compared with TPI_T2MI for all the buoys. The major differences we see between these simulations is in the snow depth (Fig. 9). TPI_T2MI has a thinner snow pack than TP5_T2M5 for all four buoys, by 0.02-0.06 m. This is due to the higher total precipitation in ERA5, compared with ERA-I (See section 3.2).

In contrast, when HIGHTSI is forced with the reanalysis' snowfall product (SFI_ERAI and SF5_ERA5) the differences in snow depth are comparable with the simulations forced by the total precipitation (TPI_T2MI and TP5_T2M5). The SFI_T2MI

runs typically have a thinner snowpack (0.01-0.06 m) and a greater ice thickness (0.04-0.09 m) than SF5_T2M5. The snow depth in SFI_T2MI is thinner (by 0.01-0.04 m) and ice thickness is greater (0.01-0.06 m) than the TPI_T2MI runs (Fig. 9). This is because there is substantial rain at sub-zero temperatures in the SFI_T2MI runs that is classified as snow in the TPI_T2MI runs. There are no large differences between the snow depth and sea ice thickness at the end of the growth season for the SF5_T2M5 and TP5_T2M5 runs because, unlike in ERA-I, there is little rain at sub-zero temperatures for SF5_T2M5.

We now look at the effect of T2M differences between ERA5 and ERA-I, and compare the TPI_T2M5 runs vs. TPI_T2MI runs (Fig. 10). When using the T2M from ERA5 and not altering the precipitation forcing, the snowpack remains unchanged from the TPI_T2MI run. However, we find a slightly thinner ice at the end of freezing season, compared with TPI_T2MI runs (0-0.04 m thinner), as a result of the larger warm bias in ERA5 which slows down the growth of sea ice. This is consistent with our results from the FDD model in Section 4.1.

Finally, we look at the effect of precipitation by comparing the TP5_T2MI and TPI_T2MI runs. The snowpack in TP5_T2MI is thicker (0.006-0.02 m), while the ice thickness is thinner (0.003-0.02 m) than in the TPI_T2MI runs (Fig. 10). The thicker snowpack, is due to the higher precipitation in ERA5 compared with ERA-I. This thicker snowpack allows less heat loss to the atmosphere, which results in less ice growth.

Overall the difference of using different T2M and TP forcing are very moderate and equal in magnitude during the freezing period. Obviously using the ERA-I SF will result in larger differences, due to the low SF in ERA-I.

In general, HIGHTSI reproduces the evolution of snow and sea ice observed by the buoys well during the freezing season (Fig. 9-10) although there are some differences. For the snowpack, there was a 10 cm increase in snow depth for IMB_2012H during late December, which seems not well captured by any of the reanalyses and therefore by any the simulations (Fig. 9a & 10a). The simulations for IMB_2012H show an increase in snow depth at the end of April, indicating a snowfall event in the reanalysis. However, this was not recorded by the buoy. Thus, not only the magnitude but also the frequency of the precipitation in the reanalysis data is crucial for the snow evolution in the simulation. The representation of snow in the model may further influence the simulated ice thickness (e.g., Fig. 9a). Evaluating precipitation in the Arctic is however challenging as mentioned previously due to the large local variability and lack of representative in-situ observations (e.g., Liston et al., 2018). Differences in the modelled sea ice thickness from the buoy observations in part arise from not knowing the local ocean heat flux at each individual buoy, however, our approach is here to look at the sensitivity relative to the differences in T2M and precipitation/snowfall in the reanalyses.

## 5. Conclusions

Atmospheric reanalysis are often used to force snow and sea ice models. The accuracy of these forcing products is paramount for the reproduction of the sea ice evolution in the model. ERA5 is a new global reanalysis product from ECMWF and will replace the widely used ERA-I. Here we compare the 2 m air temperature (T2M), snowfall and total precipitation in ERA5 and ERA-I, and evaluate these products against in-situ observations from drifting buoys (IMBs and Snow Buoys) over Arctic sea ice.

Overall, we find a warm bias in ERA-I and ERA5, when compared with the buoys. In both reanalysis, the bias is smallest in summer months, and larger in autumn, winter and spring. The warm bias in ERA5 is smaller than ERA-I in summer. However, we find a larger warm bias in ERA5 than in ERA-I during the cold season, especially when the observed T2M was lower than -25 °C in the Atlantic sector and Pacific Sector. For days when the observed T2M was <-25 °C, the daily mean difference between the reanalyses and buoys was, on average, +5.4 °C for ERA5 and +3.4 °C for ERA-I. The near surface warm bias in ERA5 and ERA-I may partly be attributed to the difference in height with observations. The larger warm bias in ERA5 during cold periods suggests this reanalysis also struggles to accurately simulate strong stable boundary layers, which frequently appear in winter and early spring, despite the higher vertical resolution compared with ERA-I (e.g., Beesley et al., 2000). It may also be also partly attributed to the simplified representation of snow and ice thickness in the reanalyses.

The total precipitation over Arctic sea ice in ERA5 was higher than in ERA-I in all seasons, amounting to an additional 20-40 mm more in most of the Arctic over a full year. Annual precipitation is higher in ERA5 especially in the Atlantic sector (by 40-100 mm). This is promising, as ERA-I is known to be drier in the Arctic compared with some other recent reanalyses (Lindsay et al., 2014; Merkouriadi et al., 2017; Boisvert et al., 2018). More critically, the snowfall is substantially higher in

ERA5 than in ERA-I in all seasons, especially during summer and autumn and especially in the Atlantic sector of the Arctic. In the Atlantic sector the annual snowfall in ERA5 is 80-200 mm water equivalent higher than in ERA-I. ERA5 has a higher snowfall to precipitation ratio than ERA-I, in particular during summer and autumn. ERA-I is known to have an anomalously large fraction of liquid precipitation (rain) and thus low snowfall to precipitation ratio in the Arctic, especially during August-September (Dutra et al., 2011; Leeuw et al., 2015). The total precipitation accumulated along the buoys drift trajectories, during the cold season (from 15 August/1 October until a buoy fails or until 30 April), was higher in ERA5 than in ERA-I for every buoy examined. The snowfall to precipitation ratio is on average 0.6 for ERA-I and 0.8 for ERA5 along buoy trajectories. This ratio in ERA5 is somewhat higher than in ERA-I for all buoys with an accumulation date starting from 1 October, and much higher than in ERA-I for buoys with accumulation starting from 15 August, likely due to anomalous autumn rainfall in ERA-I being now snowfall in ERA5. The total precipitation in ERA5 and ERA-I and the snowfall in ERA5 are closer to the SWE content of buoy measured snow pack, compared with the snowfall in ERA-I which is often much less, suggesting the total precipitation and snowfall in ERA5 are better represented. Nonetheless, the lack of representative in-situ observations and difficulty in measuring snow accumulation on sea ice in the Arctic makes it a challenge to accurately evaluate precipitation products over sea ice (e.g. Rasmussen et al., 2012; Lindsay et al., 2014; Sato et al., 2017; Blanchard-Wrigglesworth et al., 2018; Boisvert et al., 2018).

The larger warm bias during the ice growth season in ERA5, compared with ERA-I, can result in a lower ice thickness when using this as a forcing product for an ice model or a cumulative FDD model. The higher precipitation and snowfall in ERA5 results in a thicker snow pack that allows less heat loss to the atmosphere. Overall, using a 1D thermodynamic sea ice model simulations with ERA5 had a thinner ice thickness compared with ERA-I at the end of the growth season with a combined effect of higher T2M and more snow. However, the effects on ice growth are very small, order of centimeters, during the freezing period. Given snow on sea ice is such a critical factor in sea ice evolution, more representative observations are therefore needed (e.g. Merkouriadi et al., 2017; Webster et al., 2018).

**Authors contributions**

CW, KW and MAG initiated the study. CW and MAG retrieved the buoy data. CW and KW downloaded and analysed the reanalysis data and performed the 1D model simulations. All authors contributed to writing and to revisions of the manuscript.

**Acknowledgements**

The authors would like to thank Alek Petty and one anonymous reviewer, whose comments significantly improved this manuscript. This study was supported by the Research Council of Norway through project SPARSE (project no 254765), the Fram Centre "Arctic Ocean" flagship program through the SOLICE project, and by the Centre for Ice, Climate and Ecosystems (ICE) at the Norwegian Polar Institute through the N-ICE project. We wish to acknowledge ECMWF for ERA-Interim and

ERA5 data, the International Arctic Buoy Program (IABP) for the IMB data (http://imb-crrel-dartmouth.org/results/), the Alfred Wegener Institute for Snow Buoy data (http://data.seaiceportal.de/gallery/index_new.php).

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

**Table 1. Summary of deployment locations and initial conditions for the buoys. The accumulated snow water equivalent (SWE) is given based on ERA-I, ERA5 and buoy data. The cumulative SWE TP is based on total precipitation assuming precipitation falls as snow when T2M is <0 °C. The cumulative snowfall (SF) is calculated in the same period as what did for the cumulative TP. The accumulated SWE measured by the buoy is estimated using a climatological monthly mean snow density based on Warren et al. (1999).**

| Buoy | Deployment location | Period of operation | Ice type | Initial thickness (m) | | Accumulated SWE (mm water equivalent) | | | | |
|---|---|---|---|---|---|---|---|---|---|---|
| | | | | | | ERA-I | | ERA5 | | Buoy |
| | | | | ice | snow | TP | SF | TP | SF | SWE |
| 2010A | Central Arctic | 20 Apr 2010 – 2 Dec 2010 | FYI | 1.67 | 0.24 | 77.5[B] | 51.8 [B] | 80.9 [B] | 78.5[B] | 67.2[B] |
| 2011M | Central Arctic | 29 Sept 2011 – 22 Apr 2013 | MYI | 1.67 | 0.07 | 94.6[A] | 89.2 [A] | 99.8 [A] | 99.8[A] | 19.2[A] |
| 2012C | Central Arctic | 13 Apr 2012 - 4 Oct 2012 | FYI | 1.24 | 0.43 | 56.2[B] | 21.1 [B] | 65.1 [B] | 48.3[B] | NA |
| 2012D | Central Arctic | 4 May 2012 - 2 Nov 2012 | FYI | 1.67 | 0.47 | 89.9[B] | 47.1 [B] | 100.9 [B] | 91.8[B] | 124.2 [A] |
| 2012H | Beaufort Sea | 10 Sept 2012 - 16 Jan 2014 | FYI | 1.50 | 0.02 | 75.8[A] | 68.1 [A] | 83.7 [A] | 83.4[A] | 63.0[A] |
| 2012L | Beaufort Sea | 27 Aug 2012 - 25 Sept 2013 | MYI | 3.35 | 0.02 | 76.9[A] | 69.3 [A] | 90.4[A] | 90.4[A] | 12.8[A] |
| 2012I | Chukchi Sea | 14 Aug 2012 - 21 Dec 2012 | MYI | 1.09 | 0.10 | 94.8[B] | 71.1 [B] | 120.2 [B] | 111.4[B] | 98.0[B] |
| 2012J | Laptev Sea | 25 Aug 2012 – 11 Jan 2014 | MYI | 1.09 | 0 | 80.3[A] | 71.2 [A] | 94.4 [A] | 94.4[A] | 41.6[A] |
| 2013B | Central Arctic | 10 Apr 2013 - 19 Dec 2013 | NA | 2.00 | 0.02 | 151.3[B] | 104.0 [B] | 168.0 [B] | 146.8[B] | 36.4[B] |
| 2013E | Central Arctic | 11 Apr 2013 – 4 Oct 2013 | FYI | 1.40 | 0.05 | 57.4[B] | 17.8 [B] | 57.8[B] | 35.3[B] | NA |
| 2013H | Central Arctic | 3 Sept 2013 - 29 Dec 2013 | NA | 1.30 | 0.05 | 42.3[C] | 38.3 [C] | 61.7 [C] | 61.7[C] | 20.3[C] |
| 2014E | Central Arctic | 11 Apr 2014 - 18 Feb 2015 | NA | 1.73 | 0.19 | 182.6[B] | 122.9 [B] | 203.4 [B] | 192.4[B] | 103.6[B] |
| 2015D | Central Arctic | 10 Apr 2015 - 1 Feb 2016 | NA | 1.96 | 0.05 | 144.4[C] | 110.7 [C] | 176.3 [C] | 163.7[C] | 354.0[C] |
| s16 | Laptev Sea | 19 Sept 2015 - 20 Dec 2016 | FYI | NA | 0.07 | 123.6[A] | 107.6 [A] | 144.7 [A] | 144.7[A] | 80.0[A] |
| s20 | Central Arctic | 14 Sept 2015 – 19 Apr 2016 | FYI | 1.50 | 0.05 | 84.0[C] | 76.8 [C] | 89.6 [C] | 89.6[C] | ~6.0[C] |
| s29 | Laptev Sea | 10 Sept 2015 - 16 Oct 2016 | FYI | 1.20 | 0.01 | 108.5[A] | 95.9 [A] | 124.8 [A] | 124.8[A] | 20.0[A] |

NA: no data

[A]: from 1 October to 30 April.

[B]: from 15 August until the IMB fails or there is no snow data.

[C]: from 1 October until the buoy fails or there is no longer snow data during the first freezing season

**Table 2. The mean T2M, accumulated FDD, and estimated ice growth with FDD model**

| Buoy | T2M mean (°C) | | | FDD (K·d)[A] /ice growth (m)[B] | | |
|---|---|---|---|---|---|---|
| | ERA5 | ERA-I | Buoy | ERA5 | ERA-I | Buoy |
| 2011M | -22.5 | -24.2 | -26.6 | 4295/1.70 | 4662/1.78 | 5174/1.90 |
| 2012H | -22.5 | -24.1 | -25.8 | 4276/1.70 | 4624/1.78 | 4978/1.85 |
| 2012L | -22.1 | -23.1 | -24.9 | 4198/1.68 | 4402/1.73 | 4788/1.81 |
| 2012J | -20.8 | -20.8 | NA | 3902/1.61 | 3921/1.61 | NA |

NA: no data

[A]: from 1 October to 30 April.

[B]: ice growth estimation by the end of freezing season with the Lebedev FDD model (Maykut, 1986).

Table 3. Model runs and atmospheric forcing data in model simulations, where TP is total precipitation, SF is snowfall, V is wind at 10 m height, Rh is relative humidity, and CN is total cloud cover.

| Model runs | T2M | Precipitation | V, Rh, CN |
|---|---|---|---|
| TPI_T2MI | ERA-I | TP from ERA-I | ERA-I |
| TP5_T2M5 | ERA5 | TP from ERA5 | ERA5 |
| SFI_T2MI | ERA-I | SF from ERA-I | ERA-I |
| SF5_T2M5 | ERA5 | SF from ERA5 | ERA5 |
| TPI_T2M5 | ERA5 | TP from ERA-I | ERA-I |
| TP5_T2MI | ERA-I | TP from ERA5 | ERA-I |

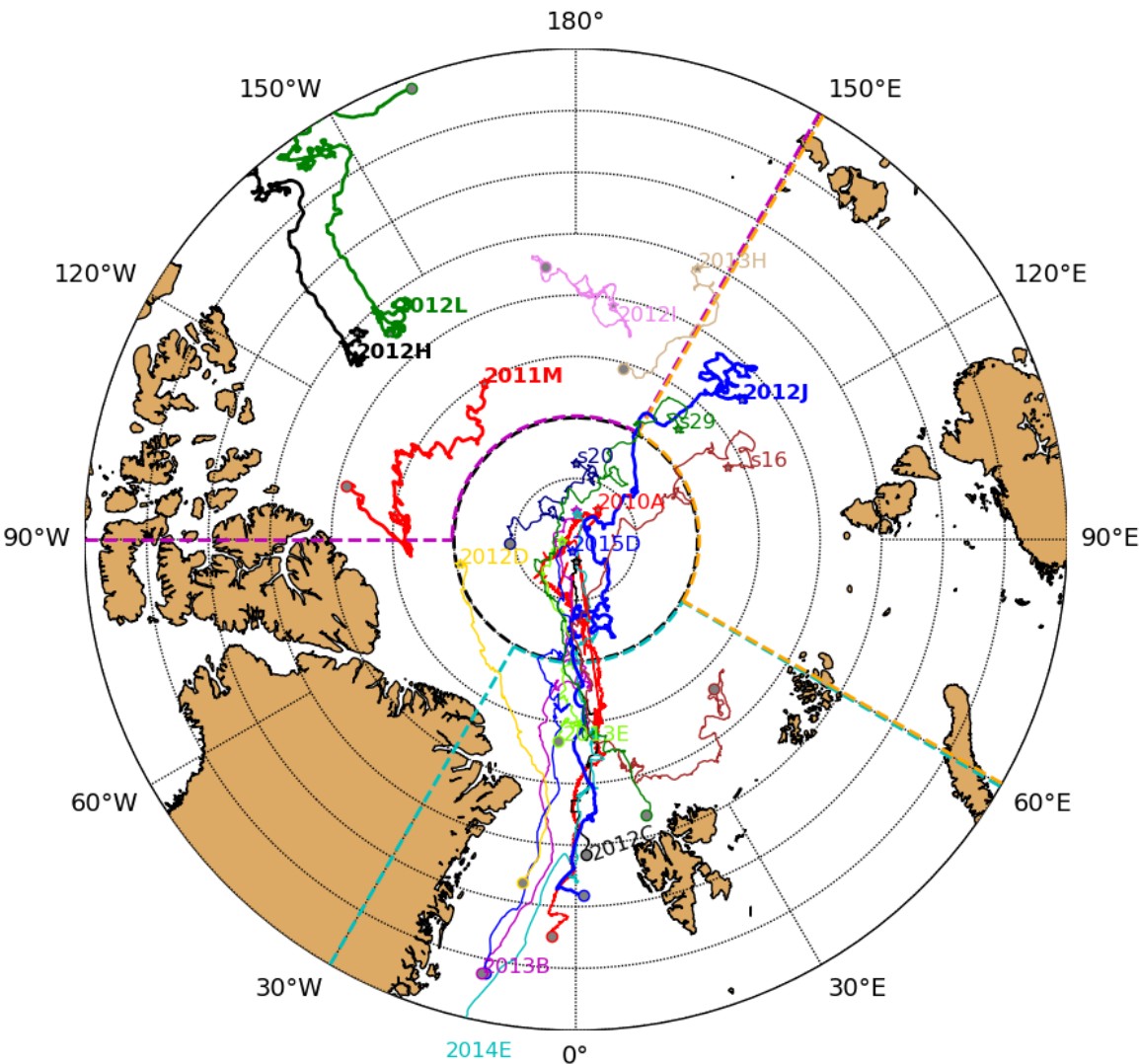

**Figure 1. Drift trajectories of all selected buoys (IMBs and snow buoys) in 2010 to 2015. Symbol "⋆" indicates the start of the drift and "o" signals the end of the drift. Buoys are labelled at the beginning or the end of the drift using same colour as trajectories. Buoys used for model simulations are highlighted with solid thick line and bold font. Dashed thick lines illustrate our definition for sectors: Central Arctic (black; north of 86° N), and south of 86°N: Pacific sector (magenta; 90° W-150° E), Atlantic sector (cyan; 30° W–60° E) and Laptev Sea (orange; 60° E – 150° E) used in Figure 6.**

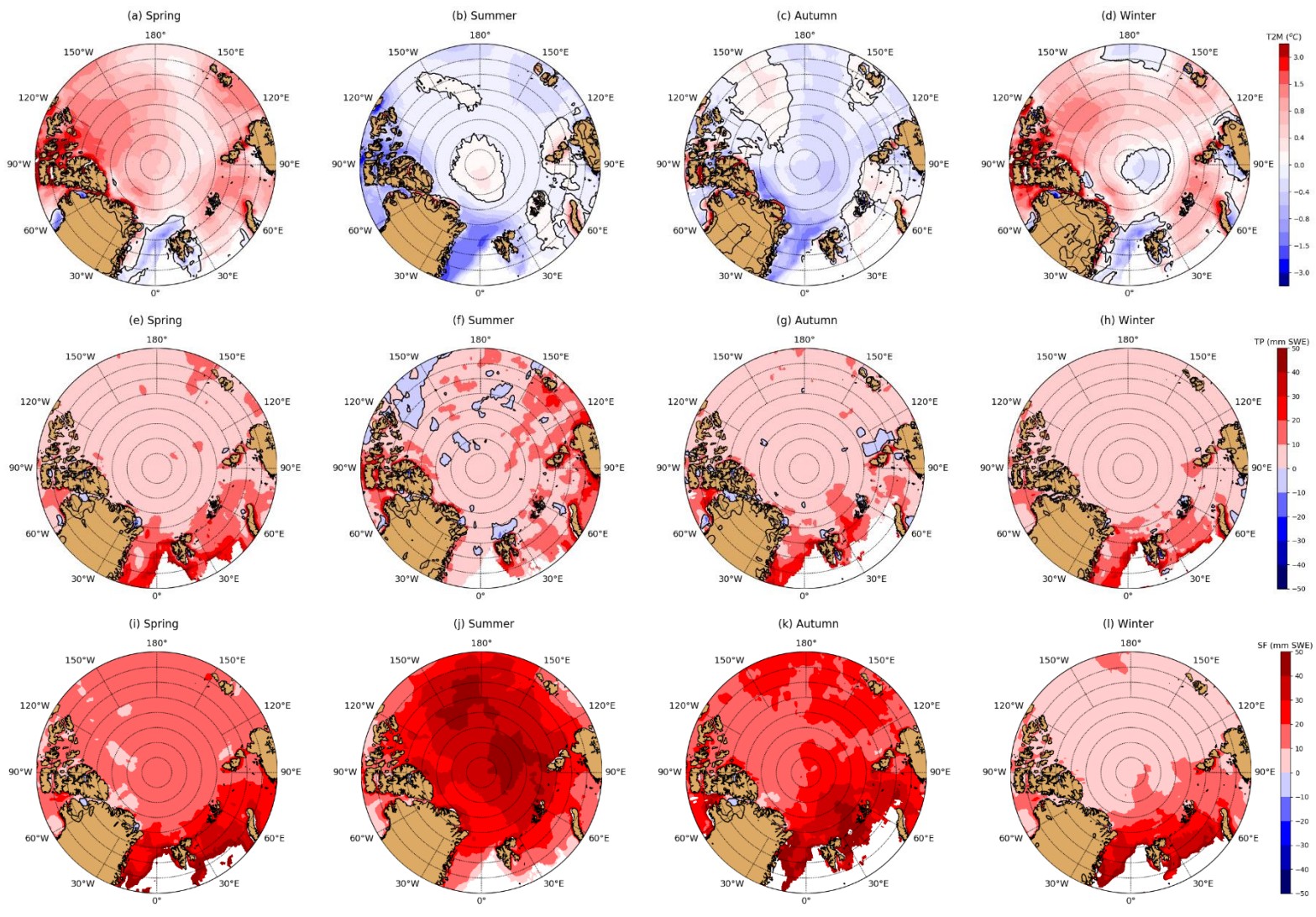

**Figure 2. Seasonal mean difference between ERA5 and ERA-I (ERA5-ERA-I) for T2M (a-d), total precipitation (e-h), and snowfall (i-l) in spring (a, e, i),**
5   **summer (b, f, j), autumn (c, g, k) and winter (d, h, l) over Arctic sea ice during 2010-2015..**

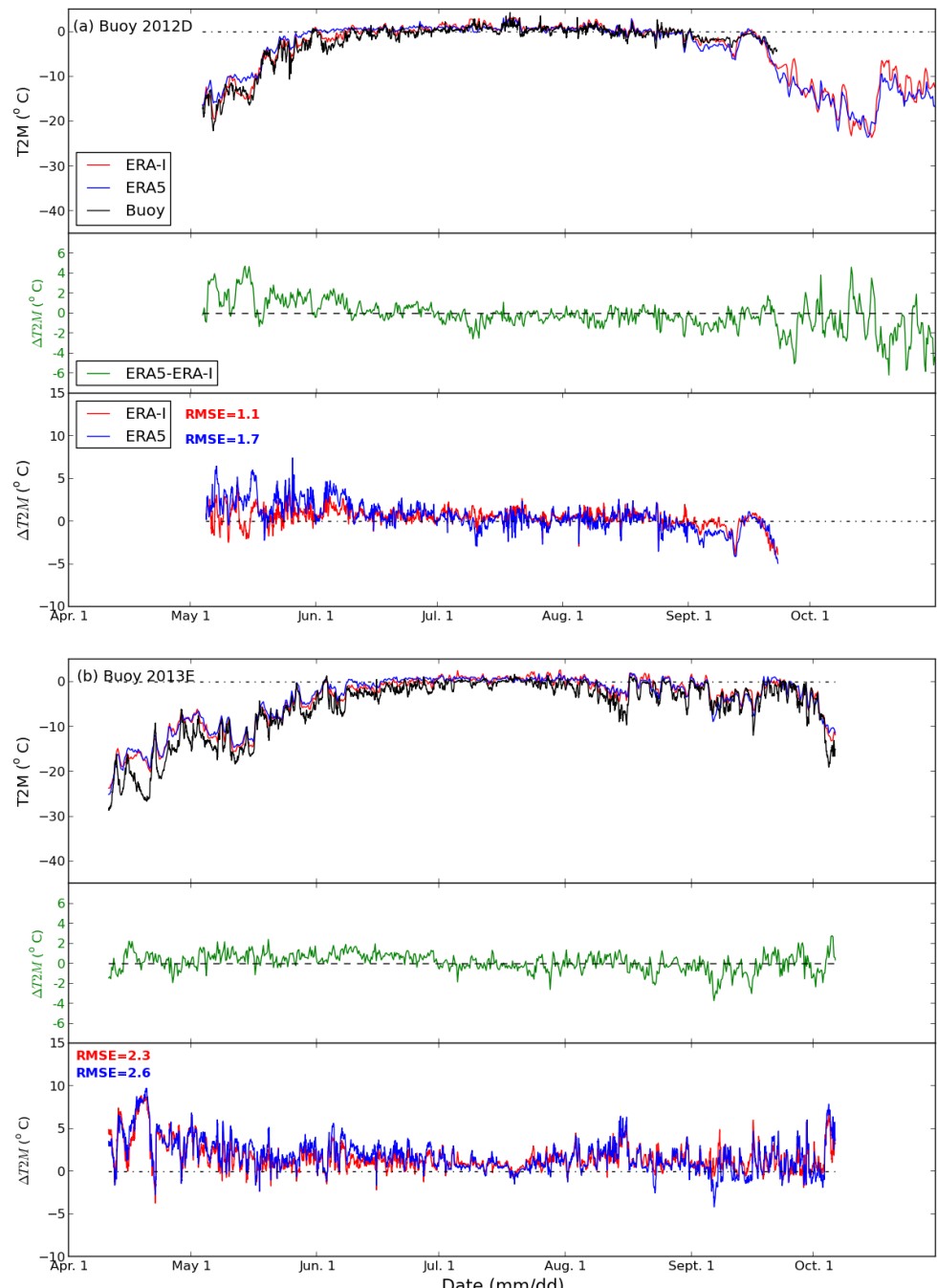

**Figure 3. Variation of 2 m air temperature (T2M) in ERA5, ERA-I and the buoys (upper panel) and the differences of T2M between ERA5 and ERA-I (mid-panel; green color) and comparisons for ERA5 and ERA-I with buoys (ERA5 minus buoy; ERA-I minus buoy) for buoys (a) 2012D and (b) 2013E.    RMSE values for the comparison between ERA products and buoys are shown as text, blue for ERA5-buoy, red for ERA-I-buoy.**

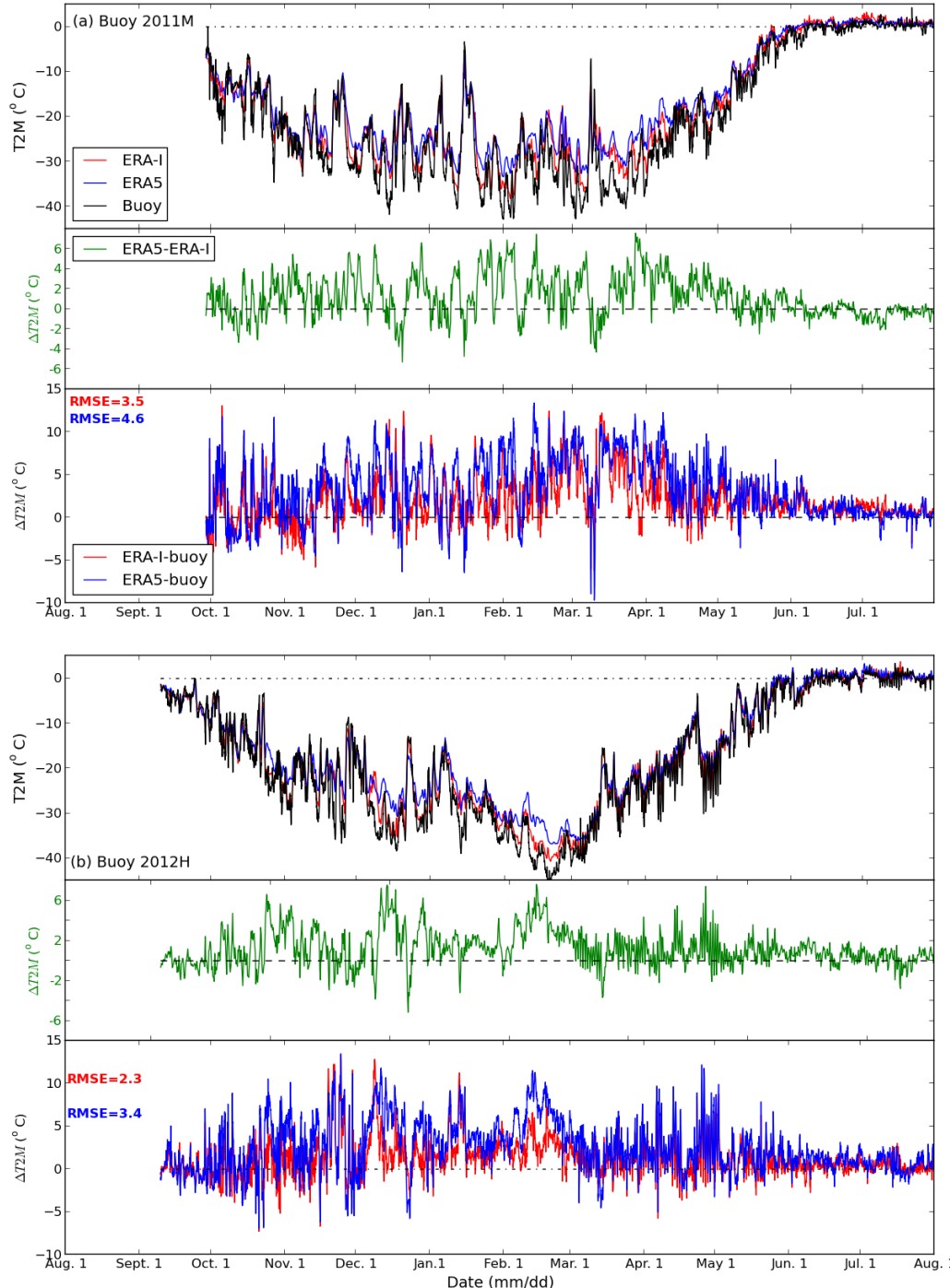

**Figure 4. Same as Fig. 3, but for (a) buoy 2011M and (b) buoy 2012H.**

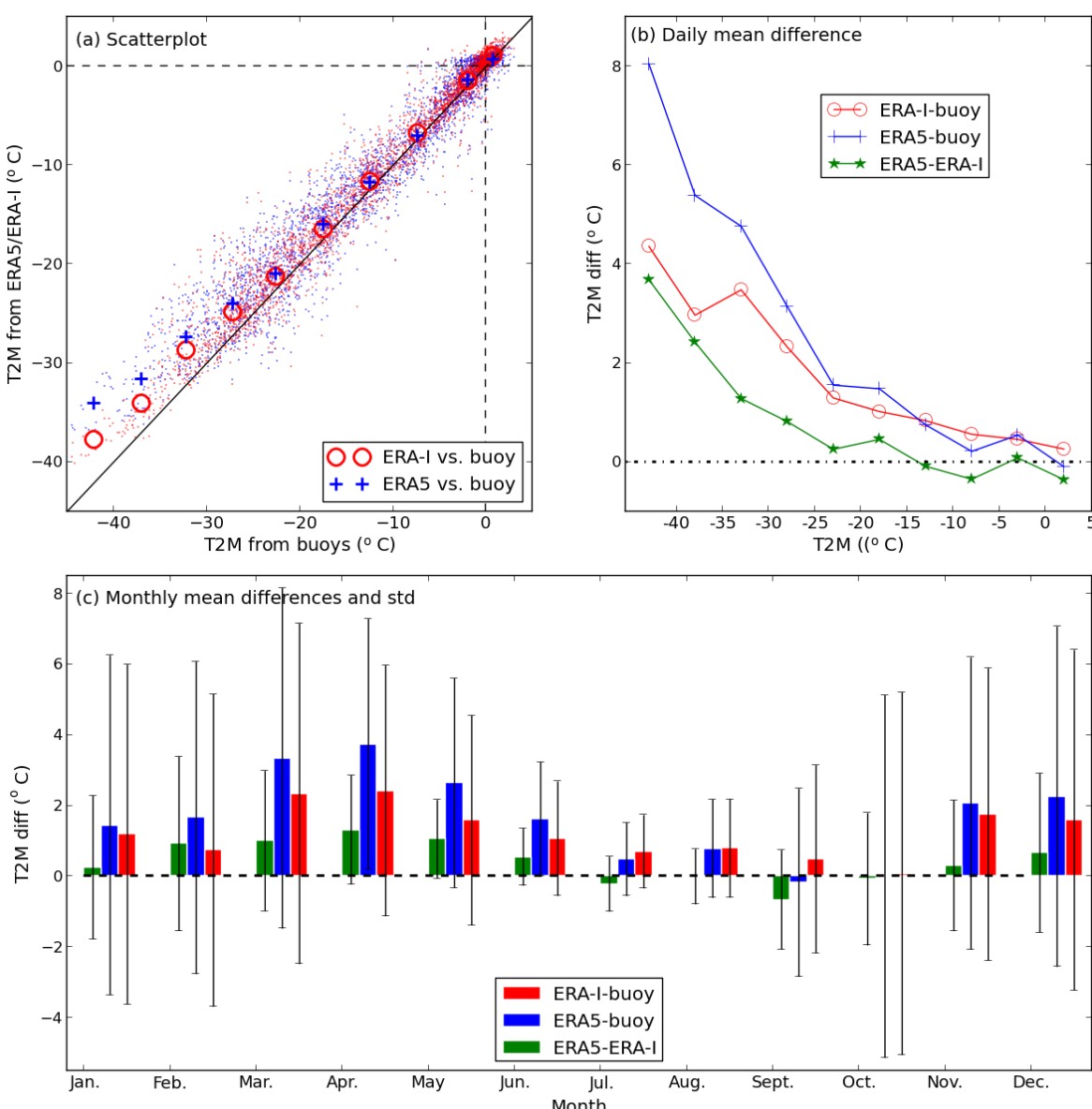

**Fig. 5. Statistics of T2M from ERA5, ERA-I and all the buoys. (a) Scatter plot for all data (small dots) and average T2M at 5 degree bins between -45 °C and +5 °C, (b) Daily temperature differences between the reanalysis and between the reanalysis and the buoys corresponding to 5 degree bins between -45 °C and +5 °C, and (c) monthly mean differences and standard deviation (std). In panel a, the black solid line is for 1:1.**

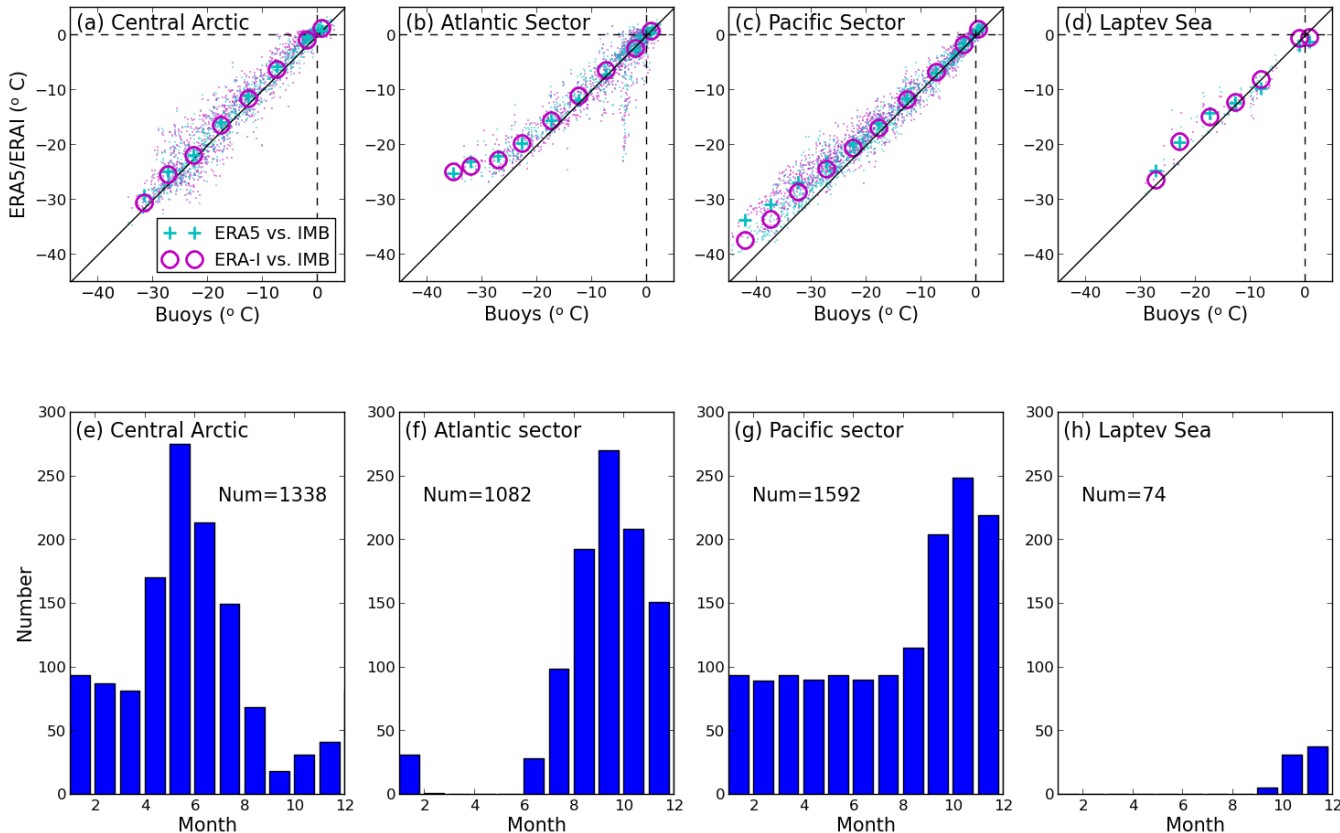

**Figure 6. Scatter plot of T2M from ERA5 and ERA-I vs. from buoys for (a) Central Arctic, (b) Atlantic sector, (c) Pacific sector, and (d) Laptev Sea, and number of buoy data (daily) per month for (e) Central Arctic (f) Atlantic sector, (g) Pacific sector, and (h) Laptev sea. The definition of sectors are shown in Figure 1.**

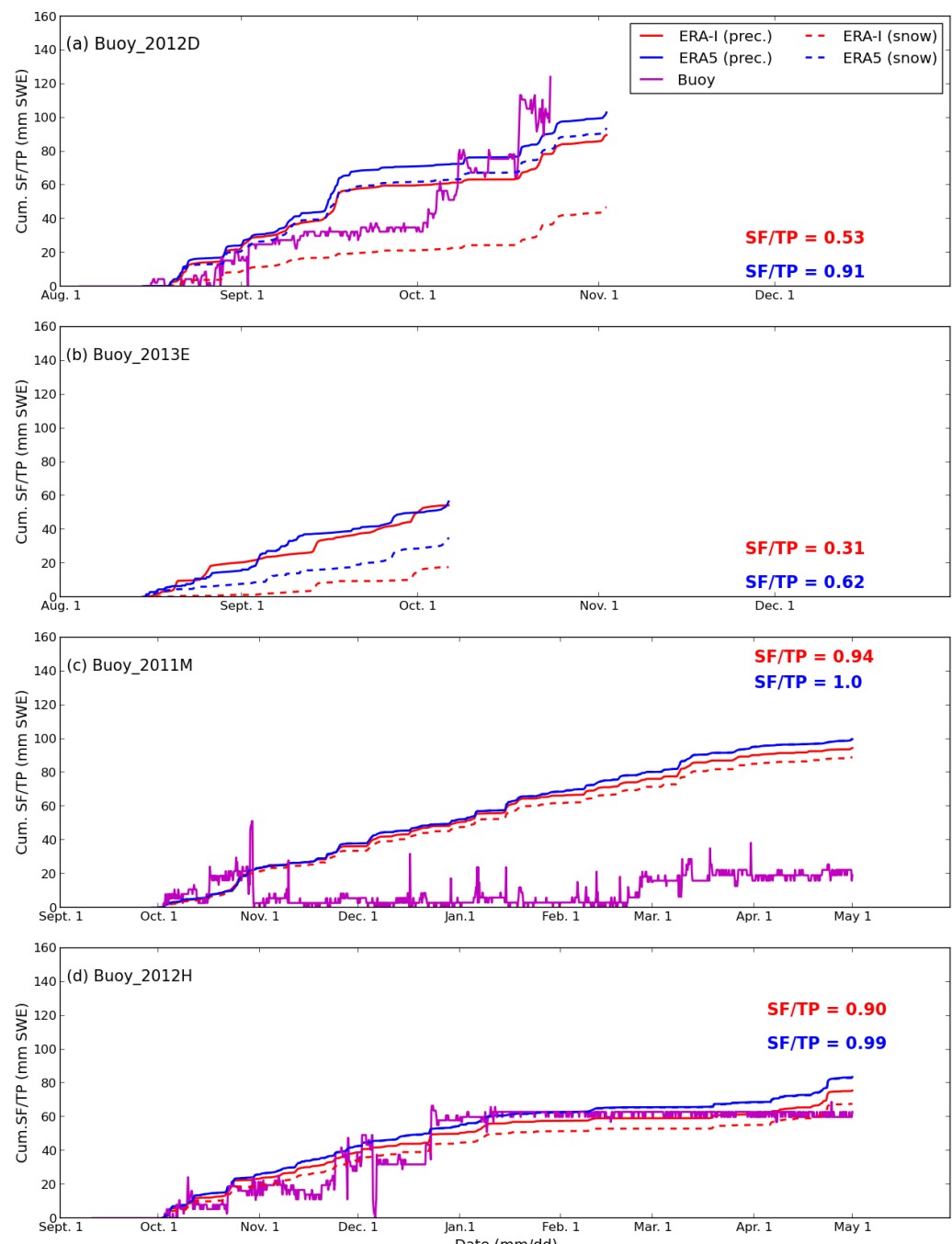

**Figure 7. Cumulative total precipitation (TP) and snowfall (SF) for ERA5 and ERA-I and snow depth for buoys (a) 2012D, (b) 2013E, (c) 2011M, (d) 2012H. Accumulation starts from 15 August for panels (a) and (b), and from 1 October for panels (c)-(d). The ratio of snowfall to total precipitation (SF/TP) in ERA5 (blue text) and ERA-I (red text) is also shown in the figure. Note there was no**

5  **snow depth data for buoy 2013E during the accumulation period.**

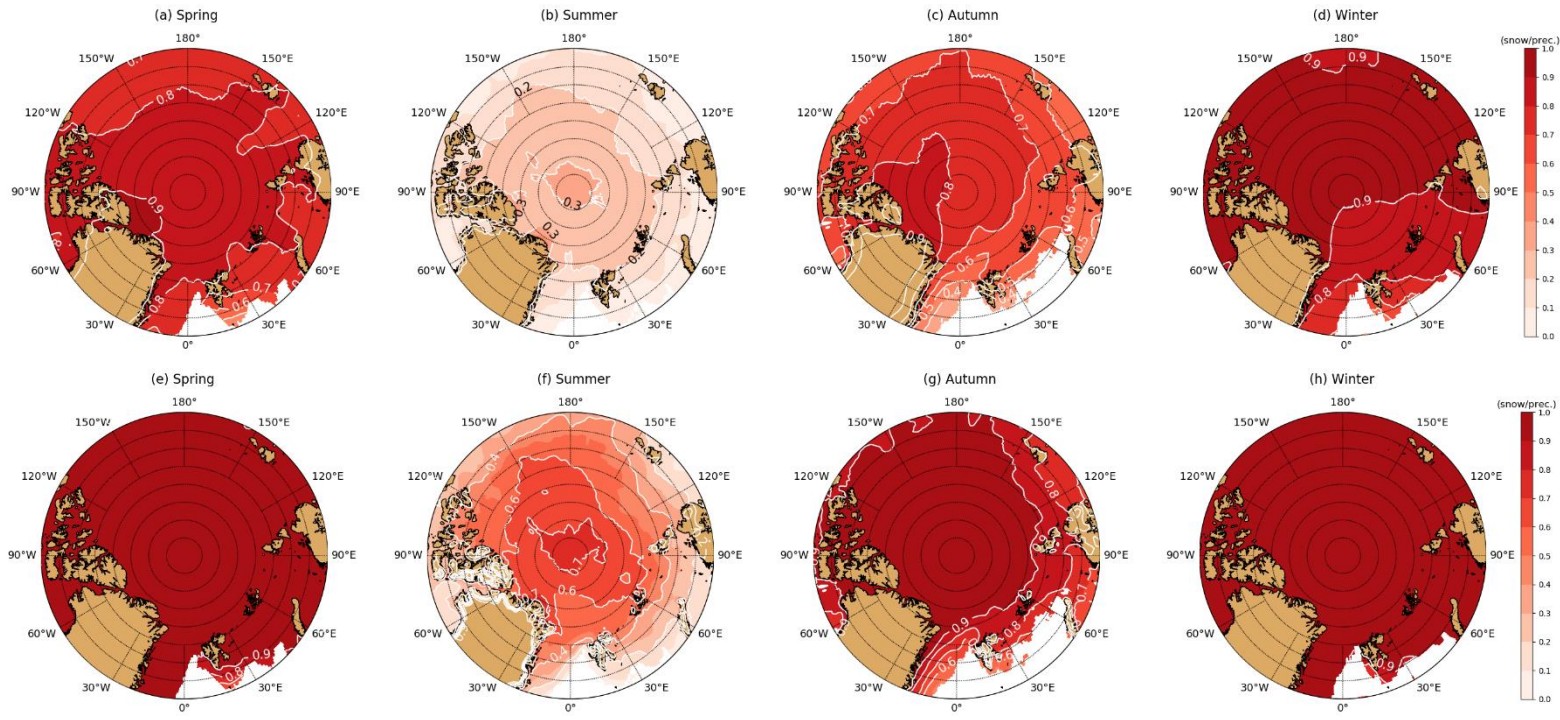

**Fig. 8 The ratio of snowfall to total precipitation (SF/TP) in ERA-I (a-d) and ERA5 (e-h) in spring (a, e), summer (b, f), autumn (c, g), and winter (d, h)**

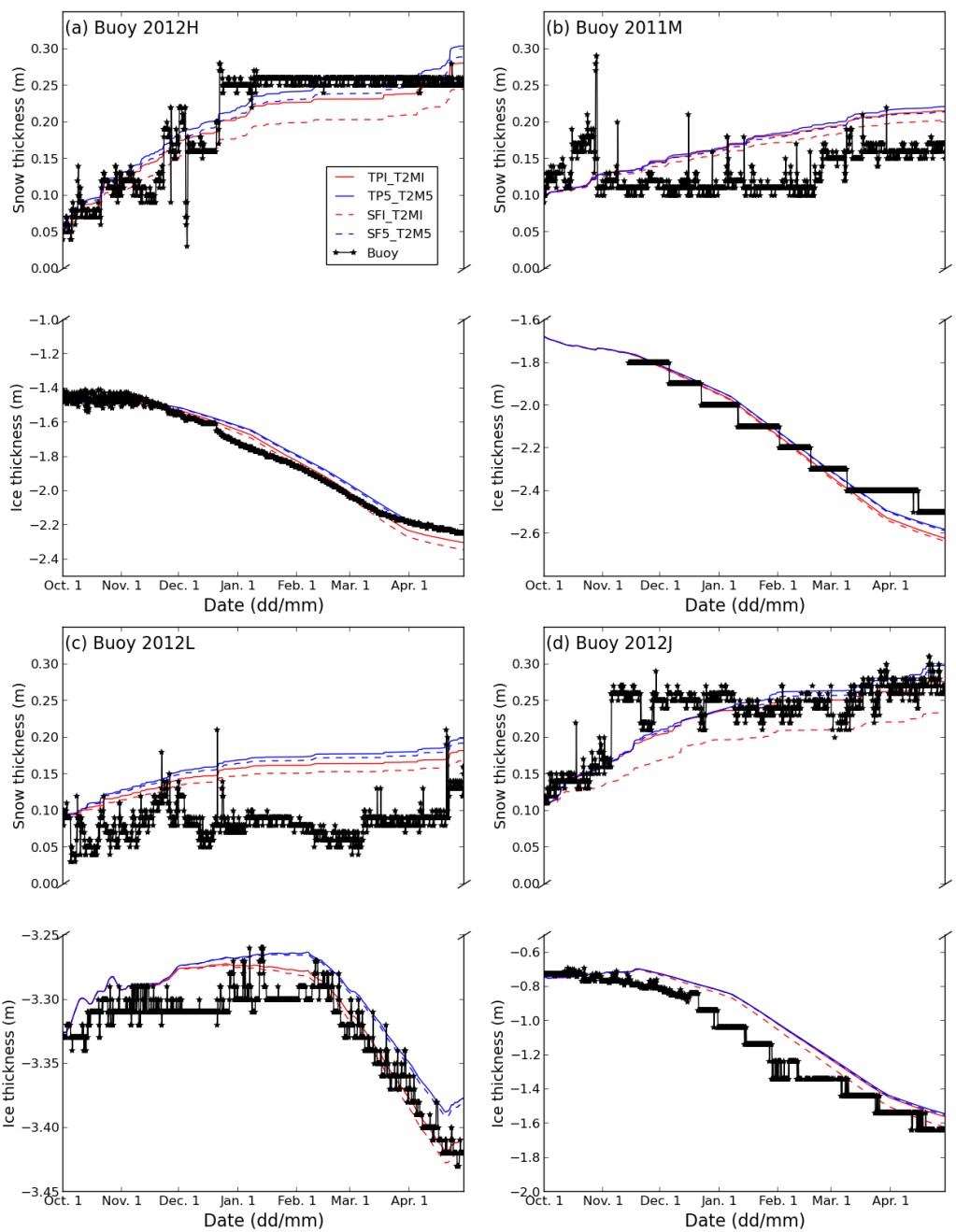

**Fig. 9. Evolution of snow and sea ice thickness during freezing season based on simulations with HIGHTSI for (a) buoy 2012H, (b) buoy 2011M, (c) buoy 2012L, and (d) buoy 2012J for runs TPI_T2MI, TP5_T2M5, SFI_T2MI, and SF5_T2M5.**

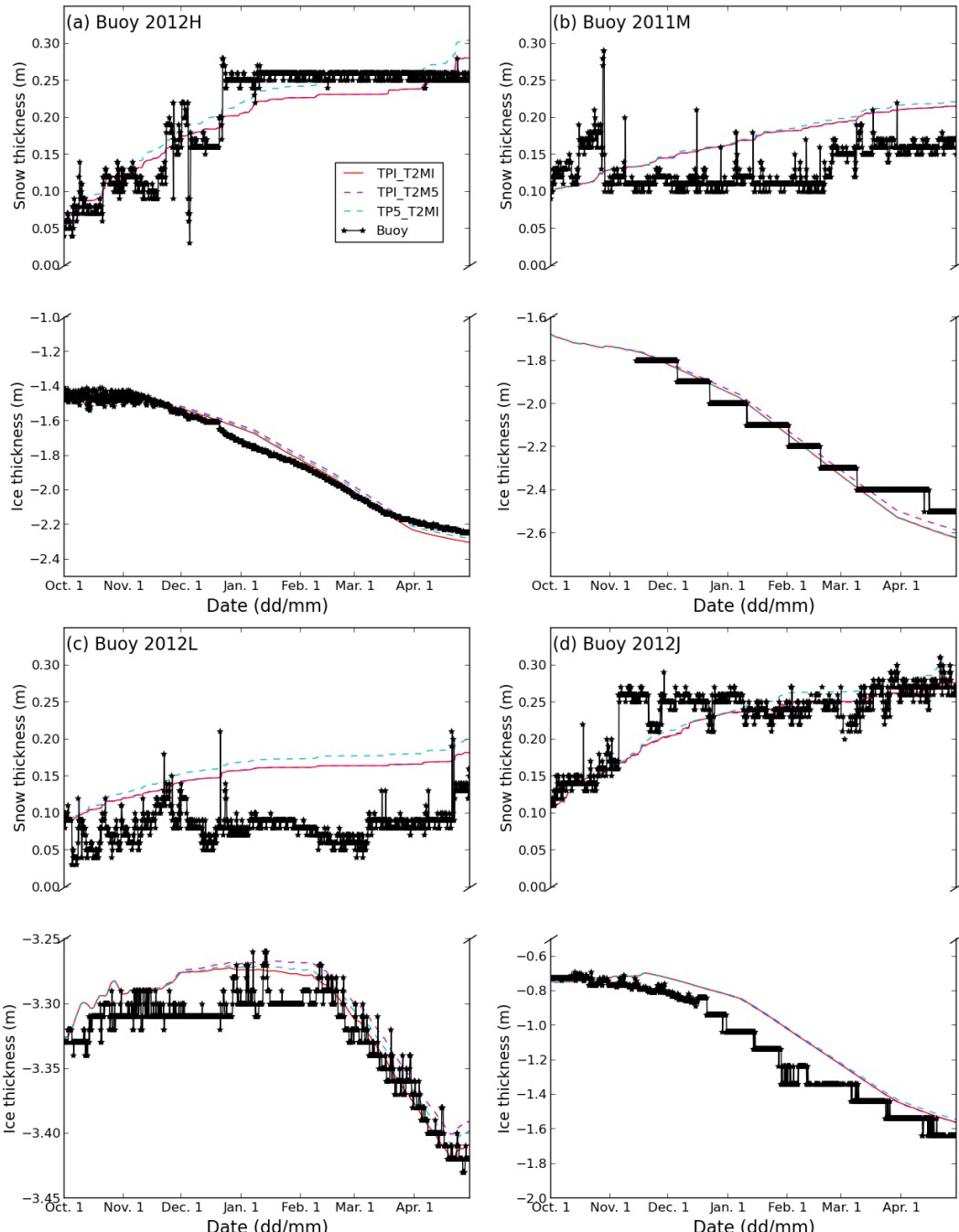

**Fig. 10. Same as Fig. 9, but for model runs TPI_T2M5 and TP5_T2MI.**

# Comparison of ERA5 and ERA-Interim near surface air temperature, snowfall and precipitation over Arctic sea ice: Effects on sea ice thermodynamics and evolution

Caixin Wang[1, 2], Robert M. Graham[2], Keguang Wang[1], Sebastian Gerland[2], Mats A. Granskog[2]

[1]Norwegian Meteorological Institute, 9293 Tromsø, Norway

[2]Norwegian Polar Institute, Fram Centre, P.O.Box 6606 Langnes, 9296 Tromsø, Norway

*Correspondence to*: Caixin Wang (caixin.wang@npolar.no)

**Abstract.** Rapid changes are occurring in the Arctic, including a reduction in sea ice thickness and coverage and a shift towards younger and thinner sea ice. Snow and sea ice models are often used to study these ongoing changes in the Arctic, and are typically forced by atmospheric reanalyses in absence of observations. ERA5 is a new global reanalysis that will replace the widely used ERA-Interim (ERA-I). In this study, we compare the 2 m air temperature (T2M), snowfall (SF) and total precipitation (TP) from ERA-I and ERA5, and evaluate these products using buoy observations from Arctic sea ice for years 2010 to 2016. We further assess how biases in reanalyses can influence the snow and sea ice evolution in the Arctic, when used to force a thermodynamic sea ice model. We find that ERA5 is generally warmer than ERA-I in winter and spring (0-1.2 °C), but colder than ERA-I in summer and autumn (0-0.6 °C) over Arctic sea ice. Both reanalyses have a warm bias over Arctic sea ice relative to buoy observations. The warm bias is smaller in the warm season, and larger in the cold season, especially when the T2M is below -25°C in the Atlantic and Pacific sectors. Interestingly, the warm bias for ERA-I and new ERA5 is on average 3.4 °C and 5.4 °C (daily mean), respectively, when T2M is lower than -25 °C. The TP and SF along the buoy trajectories and over Arctic sea ice is consistently higher in ERA5 than in ERA-I. Over Arctic sea ice, the TP in ERA5 is typically less than 10 mm SWE greater than in ERA-I in any of the seasons, while the SF in ERA5 can be 50 mm SWE higher than in ERA-I in a season. The largest increase in annual TP (40-100 mm) and SF (100-200 mm) in ERA5 occurs in the Atlantic sector. The SF to TP ratio is larger in ERA5 than in ERA-I, on average 0.6 for ERA-I and 0.8 for ERA5 along the buoy trajectories. Thus, the substantial anomalous Arctic rainfall in ERA-I is reduced in ERA5, especially in summer and autumn. Simulations with a 1D thermodynamic sea ice model demonstrate that the warm bias in ERA5 acts to reduce thermodynamic ice growth. The higher precipitation and snowfall in ERA5 results in a thicker snow pack that allows less heat loss to the atmosphere. Thus, the larger winter warm bias and higher precipitation in ERA5, compared with ERA-I, on ice growth result in thinner ice thickness at the end of growth season when using ERA5, however the effect is small during the freezing period.

# 1 Introduction

The Arctic has been undergoing substantial changes in the recent decades. The decline of Arctic sea ice is seen as one of the most prominent indicators of Arctic climate change (Stroeve et al., 2012). The extent and area of the Arctic sea ice has decreased (Comiso et al., 2008), the length of the sea ice melt season is increasing (Markus et al., 2009; Mortin et al., 2014;

Stroeve et al., 2014; Mortin et al., 2016; Stroeve and Notz, 2018), and large areas of thick multi-year ice (MYI) have been replaced by thinner and more dynamic first-year ice (FYI) (Maslanik et al., 2011; Lindsay and Schweiger, 2015; King et al., 2017). The Arctic is warming more than twice as fast as the global average temperature over the past 50 years (Bekryaev et al., 2010; AMAP, 2017). The fastest warming in the Arctic occurs during the fall and winter season (Graversen et al., 2008; Boisvert and Stroeve, 2015), and is driven in part by an increased number of storms that bring warm winds from the south

(Woods and Caballero, 2016; Dahlke and Maturilli, 2017; Graham et al., 2017a, 2017b; Rinke et al., 2017). The additional heat and moisture carried by these storms could contribute to a reduction in the winter ice growth (Woods and Caballero, 2016; Alexeev et al., 2017; Stroeve et al., 2018).

Despite the rapid ongoing changes in the Arctic, there are relatively few direct observations of the atmosphere, sea ice and ocean conditions, especially during winter. Due to the lack of in-situ observations, most studies documenting changes in the

Arctic rely heavily on atmospheric reanalyses (Screen and Simmonds, 2010; Kapsch et al., 2014; Woods and Caballero, 2016; Sato and Inoue, 2017). In addition, reanalyses are also frequently used to force snow and sea ice models (Schweiger et al., 2011; Merkouriadi et al., 2017; Stroeve et al., 2018). However, there are inherent biases and uncertainties within these reanalyses, and large differences can exist among the different products (Tjernström and Graversen, 2009; Decker, et al., 2012; Jakobson et al., 2012; Lindsay et al., 2014; Wesslén et al., 2014; Graham et al., 2017b). Thus the choice of reanalysis, and

inherent biases within that product, will ultimately influence the simulation of Arctic sea ice mass balance (Cheng et al., 2008; Wang et al., 2015).

The European Centre for Medium-range Weather Forecasts (ECMWF) reanalysis product, ERA-Interim (ERA-I, Dee et al., 2011), has been widely used for studying changes in the Arctic and forcing ocean and sea ice models (e.g., Cheng et al., 2008; Maksimovich and Vihma, 2012; Kapsch et al., 2014; Woods and Caballero, 2016; Graham et al., 2017b). In 2017, the ECMWF

released a new reanalysis ERA5 (Hersbach and Dee, 2016). There are several major improvements in ERA5 compared with ERA-I, including much higher spatial and temporal resolutions, and more consistent sea surface temperature and sea ice concentration (Hersbach and Dee, 2016). Evaluations of the performance of ERA5 have been conducted over the land and revealed a higher performance of ERA5 than ERA-I (Albergel et al., 2018; Urraca et al., 2018), and other commonly used reanalysis, such as, MERRA-2 (the second version of the Modern-Era Retrospective Analysis for Research and Applications)

(Olausen, 2018; Urraca et al., 2018). However, the performance of ERA5 over Arctic sea ice is yet to be fully investigated.

In this study, we compare and evaluate the performance of ERA-I and ERA5 over Arctic sea ice. For this, we use data from Ice Mass Balance buoys (IMB) (Perovich et al., 2018) and Snow Buoys (Grosfeld et al., 2016; Nicolaus et al., 2017) deployed in 2010 to 2015. The buoys record position, the 2 m air temperature (T2M), mean sea level pressure (MSLP), and snow depth

at regular intervals. Hence, these observations can be used to evaluate the variables of T2M, precipitation and MSLP in the reanalyses. The former two variables are critical parameters for sea ice simulation (Cheng et al., 2008; Wang et al. 2015), and form the focus of our study. We use the T2M and snow depth observations from these buoys to assess the performance of ERA5 and ERA-I over Arctic sea ice. We further use the reanalyses to force a 1-D thermodynamic sea ice model. The simulations are compared with snow and ice thickness observations from the buoys to evaluate how differences in the T2M and precipitation influence the evolution of sea ice in the model.

## 2 Materials and Methods

### 2.1 Buoy data

IMBs autonomously measure thermodynamic changes in sea ice mass balance (Richter-Menge et al., 2006; Polashenski et al., 2011). They are part of a network of drifting buoys over the Arctic Ocean that provide meteorological and oceanographic data for real-time operational requirements and research purposes (Rigor et al., 2000). These instruments typically record GPS position, T2M and mean sea level pressure (MSLP) at hourly intervals, as well as temperature profiles through the air, snow, ice, and upper-ocean, and distances to snow/ice surface and ice bottom at four hour intervals. Snow depth and ice thickness can be estimated from the distances measured by acoustic sounders, if the initial thickness of snow and ice are known when the IMB is deployed (Wang et al., 2013). If the acoustic sounders fail but the temperature string works, the positions of the ice surface and bottom can be determined from the temperature readings. Similar to IMBs, Snow Buoys also record GPS position, T2M, MSLP, and snow depth at hourly intervals (Grosfeld et al., 2016; Nicolaus et al., 2017). However, Snow Buoys do not measure temperature profiles, and provide no information on ice thickness.

Since 2000, a large number of IMBs have been deployed across the Arctic, in regions such as the Central Arctic, the Beaufort Sea, the Chukchi Sea, the Laptev Sea, the North Pole, Canadian Islands and Svalbard (Perovich et al., 2018) (http://imb-crrel-dartmouth.org/archived-data/). In this study, we use data from 13 IMBs deployed in these different regions between 2010-2015 (Fig. 1, Table 1). The IMBs were typically deployed in the Central Arctic during April/May, while deployments in the Beaufort, the Laptev, and Chukchi Seas generally took place in August/September (Fig. 1, Table 1). For additional coverage, we also use observations from 3 Snow Buoys deployed in 2015, two of which in the Laptev Sea and one in the Central Arctic (Table 1; Fig. 1) (http://www.meereisportal.de/en). For simplicity, hereafter we refer to IMBs and Snow Buoys as buoys.

### 2.2 ERA5 and ERA-I reanalysis data

ERA5 is the ECMWF's latest reanalysis product, and will replace the widely used ERA-I. The first batch of ERA5, covering the period 2010-2016, was released in July 2017. The entire ERA5 dataset, extending back to 1950 will be available for use in late 2019. ERA5 and ERA-I both have global coverage, with a horizontal spatial resolution of 80 km for ERA-I, and 31 km for ERA5. In the vertical, ERA5 resolves the atmosphere using 137 levels from the surface up to a height equalling 0.01 hPa,

and ERA-I uses 60 levels from the surface up to an equivalent height of 0.1 hPa. ERA5 provides hourly analysis and forecast fields, while ERA-I provides 6-hourly analysis and 3-hourly forecast fields. For the data assimilation, both apply 4-dimensional variational analysis (4D-var). ERA-I uses the Integrated Forecast System (IFS) "Cy31r2" 4D-Var, and ERA5 applies the newer IFS "Cy41r2" 4D-Var". ERA5 includes various newly reprocessed datasets, for example, OSI-SAFr, the reprocessed version

of the Ocean and Sea Ice Satellite Application Facilities (OSI-SAF) sea ice concentration is used (Hersbach and Dee, 2016), and recent instruments that could not be ingested in ERA-I. Many new parameters, such as 100 m wind vector, are available as part of the ERA5 output. For comparison and evaluation against buoy observations, ERA5 is bilinearly interpolated to the buoy positions, and ERA-I is first linearly interpolated to hourly data, and then bilinearly interpolated to the buoy positions. For comparison between ERA-I and ERA5 over the Arctic sea ice, the ERA-I data are first bilinearly interpolated to the grid

of ERA5, and then T2M is averaged by season, and total precipitation and snowfall are integrated over the season.

**3 Comparison of reanalysis and buoys' near surface air temperature, snowfall and precipitation over Arctic sea ice**

**3.1 Spatial distribution of seasonal differences of ERA5 and ERA-I near surface temperature, snowfall and precipitation**

Figure 2 shows the seasonal mean differences of T2M, total precipitation (TP) and snowfall (SF) between ERA5 and ERA-I

over Arctic sea ice during 2010-2015. We classify spring as March, April and May, summer as June, July and August, autumn as September, October and November, and winter as December, January and February. The seasonal mean ice extent is obtained from the monthly sea ice concentration from NOAA/NSIDC during 2010-2015 (Meier et al., 2017).

The difference in T2M between ERA5 and ERA-I varies with season (Fig. 2a-d). ERA5 is generally warmer (0-1.2 °C) than ERA-I in spring and winter, and colder (0-0.6 °C) than ERA-I during summer and autumn over Arctic sea ice. These

temperature differences are smaller during summer, but substantial during the other seasons. Near the North Pole, ERA5 is warmer than ERA-I in summer, but colder than ERA-I in winter. Whether warmer or colder, the differences between ERA5 and ERA-I are small (±0.4 °C) in this region.

ERA-I is known to be a relatively "dry" global reanalysis product in the Arctic compared with most other modern reanalyses (e.g. MERRA-2, CFSR, and JRA-55) (Lindsay et al., 2014; Merkouriadi et al., 2017; Boisvert et al., 2018). The TP

in ERA5 is typically less than 10 mm water equivalent higher than for ERA-I in all seasons over Arctic sea ice, with exception of the Atlantic sector in autumn, winter and spring where TP in ERA5 can be up to 30 mm water equivalent larger (Fig 2e, g, h). The patterns of seasonal TP over sea ice are very similar in ERA5 and ERA-I (Fig. S1), and with distinctly highest annual TP in the Atlantic sector (Fig. S2).

Snowfall is substantially higher in ERA5 than in ERA-I in all seasons (Fig. 2i-l), particularly in the Atlantic sector, where

SF is up to 50 mm SWE higher in spring, autumn and winter seasons (Fig. 2i-l). In summer snowfall is much larger in ERA5 in the central and eastern Arctic (30-50 mm SWE higher) (Fig. 2j). Thus the differences in the snowfall between ERA5 and ERA-I are much larger than for TP in all seasons except winter (Fig. 2 i-l vs. Fig. 2e-h, see also Fig. S2). The patterns of

seasonal snowfall over sea ice are very similar in ERA5 and ERA-I (Fig. S1). Annual SF has increased all across the Arctic, more than TP, especially in the Atlantic sector (>100 mm) and eastern Arctic (Fig. S2).

## 3.2 Comparison of reanalysis near surface temperature, snowfall and precipitation against buoy observations

Both ERA-I and ERA5 accurately capture the observed evolution of MSLP measured by each of the buoys (not shown). The

hourly difference between the reanalysis MSLP and observations is no more than a few hPa. Excellent agreements between observed MSLP in the Arctic and earlier reanalyses have been shown in previous studies (e.g, Makshtas et al., 2007), demonstrating that MSLP is well simulated in reanalyses. In the following, we will focus on near surface temperature, snowfall and total precipitation.

### 3.2.1 Evaluation of near surface temperature in ERA5 and ERA-I using buoy observations

Figure 3-4 and Figures S3-5 show time series of T2M from different buoys, and the corresponding T2M difference between ERA5 and ERA-I, and T2M differences between reanalyses and observations at the buoys' positions. The observed T2M reveals the pronounced seasonal cycle in the Arctic. Low temperatures persist through winter and spring, before approaching near 0°C in late May or early June. Temperatures near 0 °C, or occasionally over 0 °C, continue during summer, before lower temperatures return in late August or early September and decrease further in autumn.

The T2M in ERA5 and ERA-I generally agree well, both with each other and the observations (Figs. 3-4 & S3-5). The reanalyses perform best for buoys 2013E (Fig. 3b), and 2012J (Fig. S5a), which were both deployed in the central Arctic, the former near the North Pole and the latter closer to the Laptev Sea (Fig. 1). On occasions, hourly differences of T2M between ERA5 and ERA-I can exceed 4 °C (e.g., Fig. 4). The largest hourly T2M differences between the two reanalyses and between the reanalyses and observations (Fig. 3-4 & S3-5), are found during the coldest months (November–May). Specifically, both

reanalyses have a warm bias during these months. Previous studies have shown that warm biases in the Arctic are prevalent among most reanalysis products, particularly during the winter season (Beesley et al., 2000; Tjernstöm and Graversen, 2009; Lüpkes et al., 2010; Jacobson et al., 2012; Lindsay et al., 2014; Wesslén et al., 2014; Graham et al., 2017b). This is because weather forecast models and climate models struggle to accurately simulate strong stable boundary layers (Beesley et al., 2000; Tjernstöm and Graversen, 2009; Sotiropoulou et al., 2015; Graham et al., 2017b; Kayser et al., 2017; Biosvert et al., 2018).

Interestingly, we find a larger warm bias in the new ERA5 compared with ERA-I (Fig. 3-4 & S3-5, Table 2), despite the higher vertical resolution in ERA5. The root-mean-square deviation (RMSE) values are higher for ERA5 than for ERA-I (see Fig. 3-4 and Fig. S3-5), in the range of 1.1-3.7 °C for ERA-I and 1.7-4.6 °C for ERA5.

We note that the near surface air temperature in both reanalyses corresponds to a height of 2 m, while it is likely often measured by buoys at a lower height. The initial observation height might also decrease further as snow accumulates. During

winter, the lowest temperatures in the Arctic occur under stable conditions with a strong surface-based inversion, meaning that the temperature increases with height from the surface. Hence, the near surface warm bias in reanalyses may partly be attributed

to the difference in height with the observations (Vihma et al., 2014). A prescribed ice thickness of 1.5 m and no snow accumulation on top of sea ice were applied both in ERA5 and ERA-I. The miss representation of snow may affect the surface energy budget, physically leading to, for example, overestimated conductive heat flux from the ocean to the surface.

A scatterplot of ERA5/ERA-I vs. buoy T2M clearly reveals the temperature dependence of the warm bias in both reanalyses (Fig. 5a). The data crowd together near the 1:1 line when the air temperature is near 0°C, but spread further above the 1:1 line when the air temperature is low, especially at air temperatures below -25°C. The temperature dependence of the warm bias is also demonstrated in Fig. 5b, which shows the relationship between the daily mean T2M differences with the temperature bins of 5 °C from -45 – +5 °C. When the T2M is below -25 °C, the daily mean difference between reanalysis and observation is

higher than 2 °C, with ERA5 3.1 – 8.0 °C warmer than in buoys, and ERA-I 2.4 – 4.4 °C warmer than in buoys (Fig. 5b). For air temperatures above -25 °C, the bias between reanalysis and buoys is smaller, with ERA5 and ERA-I both 0.75 °C warmer than the observations on average.

Figure 5c shows the bias and standard deviation (std) for the reanalyses for each month, based on the buoy observations, and the temperature difference between the reanalyses. The smallest biases, and the smallest T2M differences between ERA5

and ERA-I are found in the months between July and October (also refer to Fig. 3-4 & S3-5). ERA5 is typically warmer than ERA-I (and has a larger warm bias) throughout the winter and spring, including June. However, ERA5 is colder than ERA-I (0.01-0.6 °C) and has a smaller bias from July to October (Fig. 5c). Hence, the warm bias in ERA5 is smaller than ERA-I in the warm season (July-October). ERA-I has a warm bias in the warm season, but the magnitude is smaller (< 0.8 °C) than the warm bias in the cold season (Fig. 5c). Similarly, ERA5 has a small warm bias during July and August (<1 °C), and a likely

insignificant cold bias (< 0.2 °C) in September and October (Fig. 5c).

The performance of reanalysis near surface temperature varies with region over Arctic sea ice (Fig. 6, also refer to Fig. 2). According to the buoys' positions (Fig. 1), we define four regions in the Arctic: the Central Arctic (north of 86° N), and the Pacific sector (90° W – 150° E), the Atlantic sector (30° W – 60° E), and the Laptev Sea (60° E – 150° E). The later three sectors are south of 86° N. The ERA5/ERA-I near surface temperature performs best in the Central Arctic (Fig. 6a), and well

in the Pacific sector (Fig. 6c). It performs well in the Atlantic sector when the T2M is above -25 °C, but poorly when the T2M is below -25 °C (Fig. 6b). The performance of reanalysis near surface temperature in the Laptev Sea needs to be further investigated due to small number of observations in this region (Fig. 6d & 6h). However, there is also some seasonal bias in the availability of data from buoys in the different regions, largely due to when buoys are deployed in different regions of the Arctic and subsequent ice drift patterns.

**3.2.2 Comparison of precipitation and snowfall from ERA5 and ERA-I along buoy drift trajectories**

We next compare the cumulative total precipitation and snowfall in ERA5 and ERA-I in autumn and winter, along the drift trajectories of the buoys. We begin accumulation from 15 August onwards if the buoy was deployed before this date, or from

October if the buoy was installed after 15 August but before 1 October. We accumulate the precipitation until 30 April, or until the buoy stopped working if this occurred before 30 April (Table 1).

The accumulated total precipitation (TP) in ERA5 is higher than ERA-I for all buoys (Figs. 7 & S6-7, and Table 1), which is consistent with the seasonal difference in TP documented in section 3.1 (Fig. 2e-h). On average, the accumulated TP in ERA5 is 13.8 mm water equivalent larger than in ERA-I, with differences for the individual buoys ranging from 0.4 (buoy 2013E; Fig. 7b) to 31.9 mm water equivalent (buoy 2012D; Fig. 7a). This is in agreement with the seasonal differences between the reanalyses (Fig. 2e-h).

Similar to the accumulated TP, the accumulated snowfall (SF) in ERA5 is larger than in ERA-I (Figs. 7 & S6-7; Table 1). For buoys deployed near the North Pole that started accumulating on 15 August, the accumulated SF in ERA5 is typically much larger than for ERA-I (Fig. 7a-b & S6, S7a). In contrast, for buoys deployed in other regions, which started accumulating on 1 October, the accumulated SF in ERA5 is typically slightly higher than ERA-I (Fig. 7c-d & Figs. S7b-f).

The ratio of snowfall to total precipitation (SF/TP) in ERA5 and ERA-I along the buoy trajectories is shown in Fig. 7 and Fig. S4-5. A higher ratio means that more precipitation falls as snow. The ratio of SF/TP for the buoy trajectories ranges from 0.31 to 0.94 in ERA-I, and from 0.62 to 1.0 in ERA5, with consistently more precipitation falling as snow in ERA5. The SF/TP ratio for ERA5 increases on average by 0.28 for the buoys that started accumulating on 15 August compared with that in ERA-I. In contrast, the ratio of SF/TP usually is 0.1 higher in ERA5 than in ERA-I for buoys that started accumulating on 1 October. This means that a substantial fraction of precipitation falls as rain in ERA-I during autumn (August-September), but the same precipitation events in ERA5 are classified as snowfall. This difference in SF/TP ratio can help to explain why the accumulated SF in ERA5 is much greater than ERA-I for buoys deployed in August, but only slightly higher than ERA-I for buoys starting in October. The higher ratio of SF/TP in ERA5 than in ERA-I takes place in all seasons over the Arctic sea ice (Fig. 8a-d vs. Fig. 8e-h). The increase of SF/TP ratio in ERA5 is more significant in autumn (~0.2) and summer (~0.3-0.4) as we found along the buoy trajectories, and relatively small in winter (~0.1) and spring (~0.1-0.2). This indicates more precipitation falls as snow in ERA5 not only in autumn, but also in summer.

The low SF/TP ratio and thus larger fraction of rainfall in ERA-I is known to be anomalous, and is likely due to the cloud physics scheme used (e.g., Dutra et al., 2011; Leeuw et al., 2015). In ERA-I, the split between liquid and ice in clouds is determined diagnostically as a function of temperature from -23 to 0 °C, with ice-only only below -23 °C and liquid-only above 0 °C. In contrast, the IFS Cy41r2 used in ERA5 includes a prognostic microphysics scheme, with separate cloud liquid, cloud ice, rain and snow prognostic variables (Sotiropoulou et al., 2015; see also ECMWF IFS documentation – Cy41r2; https://www.ecmwf.int/sites/default/files/elibrary/2016/16648-part-iv-physical-processes.pdf). Our findings indicate that ERA5 has significantly less Arctic rainfall than ERA-I, particularly in autumn (Fig. 7, Figs. S6-7) and summer (Fig. 8b and Fig. 8f).

Evaluating the performance of precipitation products over the Arctic Ocean is a major challenge due to the lack of observations, and difficulty accurately measuring snowfall (e.g. Lindsay et al., 2014; Rasmussen et al., 2012; Sato et al., 2017;

Blanchard-Wrigglesworth et al., 2018; Boisvert et al., 2018; Webster et al., 2018). Here we compare the precipitation from ERA-I and ERA5 with snow depth measurements from the buoys (Table 1). For this comparison, snow depth from the buoys is converted to snow water equivalent (SWE) using a climatological monthly mean snow densities of 220-380 kg m$^{-3}$ (Warren et al., 1999). Caution must be taken here, as the buoys reflect point observations, while the reanalyses provide a grid cell average. Snow depth is known to have large variability even over relatively small spatial scales (Warren et al., 1999; Sturm et al., 2002; Liston et al., 2018). An unknown fraction of the true snow fall will also be lost through blowing snow into leads, which is not accounted for in our calculation.

The accumulated TP and SF from ERA-I and ERA5 are generally comparable with the observed SWE from buoys in most cases during the accumulation period (refer to Fig. 7 & Figs. 4-5), such as for buoy 2012H deployed in the Beaufort Sea (Fig. 7d) (refer to Fig. 7 & Figs. 4-5). However, in several cases the accumulated TP and SF from ERA-I and ERA5 are considerably lower than the observed SWE from buoys, such as for buoy 2012D from mid-September (Fig. 7a). This may be caused by snow drifting up against the buoy structure, or reflect anomalously low precipitation in the reanalyses. In other cases, the accumulated TP and SF from reanalysis is higher the observed SWE from buoys during some periods (buoys 2013B (Fig. S6c) and s20 (Fig. S7e) or for the whole accumulation period (buoys 2011M (Fig. 7c), 2012L (Fig. 7c) and S29 (Fig. S7f). This could be caused by snow erosion/sublimation around the buoy, or reflect anomalously high precipitation in the reanalyses. By the end of the accumulation period, the accumulated TP/SF is larger on average 55.4/41.9 mm SWE for ERA-I and 66.5/62.8 mm SWE for ERA5 than the observed SWE of the snow pack along the buoy trajectories (see Table 1).

## 4 Influence of air temperature and precipitation on sea ice evolution during the freezing season

In this section, we evaluate the impact of different forcing products (ERA-I, ERA5, and the buoys) on sea ice evolution. We focus on the freezing/growth season, from 1 October to 30 April, when sea ice generally starts to grow after summer. This period corresponds to the time when the largest differences of T2M between ERA5 and ERA-I were found (Figs. 2-4). For this exercise, we focus on buoys 2011M, 2012H, 2012L, and 2012J that were deployed in late August/early September and operated for more than one year, covering a complete freezing season (Table 1). These buoys were installed on MYI or FYI in the central Arctic (buoy 2011M), the Beaufort Sea (buoy 2012H, buoy 2012L), or the Laptev Sea (buoy 2012J). When these buoys were installed, sea ice thickness was between 1-2 m for buoys 2011M, 2012H, and 2012J, while buoy 2012L had an ice thickness of 3.35 m (Table 1). Snow depth was typically a few centimetres of snow at deployment. We use these buoys to assess the impact of different forcing data on sea ice evolution. For our simple approach we apply the empirical cumulative freezing degree day (FDD) model, which accounts for differences in T2M, and a 1D sea ice model that also account for effects of precipitation/snowfall.

## 4.1 Assessing the sea ice evolution with freezing degree days (FDD): impact of temperature bias

The cumulative freezing degree days (FDD) model only needs air temperature as input and is often used to estimate sea ice growth ($\Delta h$) from zero (e.g., Huntemann et al., 2014; Lei et al., 2017). The sea ice growth is estimated based on Lebedev (Maykut, 1986), $\Delta h = 1.33 \sum (FDD)^{0.58}$, where $\sum FDD$ is daily average temperature below the freezing point of sea water (-
1.8 °C), integrated over the time period from 1 October to 30 April.

    The positive near surface air temperature bias in ERA5 and ERA-I results in a negative ice thickness bias at the end of the growth season. The cumulative FDD is smallest for ERA5 (Fig. S8, Table 2), corresponding to the largest warm bias in ERA5 during the freezing season. The differences in FDD between ERA5, ERA-I and buoys are large for buoys 2011M, 2012H and 2012L, but negligible for buoy 2012J. The ice growth is 0.08-0.12 m less, with a mean of -0.09 m for ERA-I T2M, and 0.13-
0.20 m less, with a mean of -0.16 m for ERA5 T2M compared to when using near surface buoy temperatures (Table 2).

## 4.2 Assessing sea ice evolution with a 1D sea ice model HIGHTSI: impact of T2M and precipitation

HIGHTSI is a 1D high-resolution thermodynamic snow and ice model designed for process studies to resolve the evolution of snow/ice thickness and temperature profile. The snow and ice temperature regimes are solved by the partial differential heat conduction equations applied for snow and ice layers, respectively. The turbulent surface fluxes are parameterized taking the
thermal stratification of the atmosphere surface layer into account. Downward short- and longwave radiative fluxes are parameterized based on the total cloud cover. The model has been extensively used in Arctic studies (e.g., Cheng et al., 2008; Cheng et al., 2013; Wang et al., 2015; Merkouriadi et al., 2017).

    In this section we perform six sensitivity simulations on each of the four buoys to explore the impact of temperature and precipitation on snow and sea ice evolution (Table 3). In the first two simulations, SFI_T2MI and SF5_T2M5, we force
HIGHTSI with the T2M, 10 m wind speed (V), relative humidity (Rh), total cloud cover (CN) and snowfall, from ERA-I and ERA5 (Fig. S9), respectively. In the next two simulations, TPI_T2MI and TP5_T2M5, we force the model with the total precipitation from the reanalyses, rather than the snowfall, and treat precipitation as snow only when T2M is below 0 °C. In the final two simulations, we evaluate the influences of T2M and precipitation on the sea ice evolution individually. Specifically, we replace the T2M from ERA-I in the TPI_T2MI run with the T2M from ERA5, and name this run TPI_T2M5.
Similarly, we replace the TP from ERA-I, in the run of TPI_T2MI, with the TP from ERA5 for the TP5_T2MI run (see Table 3). For all of the simulations we apply a seasonally variant ocean heat flux according to McPhee et al. (2003), which is large in October (10-20 Wm$^{-2}$), and decreases to nearly zero from mid-November (see Fig S9). Snow-ice, an ice type formed at ice surface (e.g., Leppäranta, 1983), was recently found to significantly contribute to the Arctic sea ice mass balance in a region with thick snowpack on relatively thin ice (Granskog et al., 2017; Merkouriadi et al., 2017). A few (1.5-3) millimetres snow-
ice formed only in the TPI_T2MI and TPI_T2M5 runs for buoy 2012J (with the lowest initial ice thickness of all buoys examined, Table 1). This is negligible for the total ice mass balance. Thus, the effect we examine solely depends on the differences in T2M and precipitation/snowfall on thermodynamic ice growth.

The pattern of snow accumulation recorded by many buoys is consistent with observations by Warren et al. (1999). Namely, they record snow accumulation in late fall, followed by a relatively constant snow depth from December/January–March, and sometimes a late increase in snow depth in early spring (Fig. 9). For example, the observed snow depth at buoy 2012H increased to about 0.25 m in late December, and changed marginally thereafter (Fig. 9a). Similarly, the observed snow depth at buoy
2012L increased from 0.03 m to 0.13 m from early October to mid-November, and then remained around 0.10 m until the end of April (Fig. 9c). Most buoys recorded an increase in ice thickness from early December to the end of the freezing season. For example, the sea ice growth for buoy 2012H began in early December, at a rate of approximately 0.5 cm/d, until late March, and afterward the growth became sluggish at a rate of 0.16 cm/d until the end of April (Fig. 9a). However, buoy 2012L, which had an initial ice thickness of ~ 3.3 m, showed no significant growth until early February, before undergoing a slight
increase from around 3.3 m to 3.42 m by the end of the freezing season (Fig. 9c). Sea ice growth for buoy 2011M (Fig. 9b) and 2012J (Fig. 9d) showed a staircase pattern since the ice thickness was derived from measured temperature profile due to the failure of acoustic sounders as mentioned in section 2.1.

We first compare the simulations TPI_T2MI and TP5_T2M5. Differences in the ice thickness at the end of the growth season for these simulations are relatively small, despite the larger warm bias in ERA5 (Fig. 9). Sea ice was marginally thinner
(0.006-0.02 m) in TP5_T2M5 compared with TPI_T2MI for all the buoys. The major differences we see between these simulations is in the snow depth (Fig. 9). TPI_T2MI has a thinner snow pack than TP5_T2M5 for all four buoys, by 0.02-0.06 m. This is due to the higher total precipitation in ERA5, compared with ERA-I (See section 3.2).

In contrast, when HIGHTSI is forced with the reanalysis' snowfall product (SFI_ERAI and SF5_ERA5) the differences in snow depth are comparable with the simulations forced by the total precipitation (TPI_T2MI and TP5_T2M5). The SFI_T2MI
runs typically have a thinner snowpack (0.01-0.06 m) and a greater ice thickness (0.04-0.09 m) than SF5_T2M5. The snow depth in SFI_T2MI is thinner (by 0.01-0.04 m) and ice thickness is greater (0.01-0.06 m) than the TPI_T2MI runs (Fig. 9). This is because there is substantial rain at sub-zero temperatures in the SFI_T2MI runs that is classified as snow in the TPI_T2MI runs. There are no large differences between the snow depth and sea ice thickness at the end of the growth season for the SF5_T2M5 and TP5_T2M5 runs because, unlike in ERA-I, there is little rain at sub-zero temperatures for SF5_T2M5.
We now look at the effect of T2M differences between ERA5 and ERA-I, and compare the TPI_T2M5 runs vs. TPI_T2MI runs (Fig. 10). When using the T2M from ERA5 and not altering the precipitation forcing, the snowpack remains unchanged from the TPI_T2MI run. However, we find a slightly thinner ice at the end of freezing season, compared with TPI_T2MI runs (0-0.04 m thinner), as a result of the larger warm bias in ERA5 which slows down the growth of sea ice. This is consistent with our results from the FDD model in Section 4.1.
Finally, we look at the effect of precipitation by comparing the TP5_T2MI and TPI_T2MI runs. The snowpack in TP5_T2MI is thicker (0.006-0.02 m), while the ice thickness is thinner (0.003-0.02 m) than in the TPI_T2MI runs (Fig. 10). The thicker snowpack, is due to the higher precipitation in ERA5 compared with ERA-I. This thicker snowpack allows less heat loss to the atmosphere, which results in less ice growth.

Overall the difference of using different T2M and TP forcing are very moderate and equal in magnitude during the freezing period. Obviously using the ERA-I SF will result in larger differences, due to the low SF in ERA-I.

In general, HIGHTSI reproduces the evolution of snow and sea ice observed by the buoys well during the freezing season (Fig. 9-10) although there are some differences. For the snowpack, there was a 10 cm increase in snow depth for IMB_2012H during late December, which seems not well captured by any of the reanalyses and therefore by any the simulations (Fig. 9a & 10a). The simulations for IMB_2012H show an increase in snow depth at the end of April, indicating a snowfall event in the reanalysis. However, this was not recorded by the buoy. Thus, not only the magnitude but also the frequency of the precipitation in the reanalysis data is crucial for the snow evolution in the simulation. The representation of snow in the model may further influence the simulated ice thickness (e.g., Fig. 9a). Evaluating precipitation in the Arctic is however challenging as mentioned previously due to the large local variability and lack of representative in-situ observations (e.g., Liston et al., 2018). Differences in the modelled sea ice thickness from the buoy observations in part arise from not knowing the local ocean heat flux at each individual buoy, however, our approach is here to look at the sensitivity relative to the differences in T2M and precipitation/snowfall in the reanalyses.

## 5. Conclusions

Atmospheric reanalysis are often used to force snow and sea ice models. The accuracy of these forcing products is paramount for the reproduction of the sea ice evolution in the model. ERA5 is a new global reanalysis product from ECMWF and will replace the widely used ERA-I. Here we compare the 2 m air temperature (T2M), snowfall and total precipitation in ERA5 and ERA-I, and evaluate these products against in-situ observations from drifting buoys (IMBs and Snow Buoys) over Arctic sea ice.

Overall, we find a warm bias in ERA-I and ERA5, when compared with the buoys. In both reanalysis, the bias is smallest in summer months, and larger in autumn, winter and spring. The warm bias in ERA5 is smaller than ERA-I in summer. However, we find a larger warm bias in ERA5 than in ERA-I during the cold season, especially when the observed T2M was lower than -25 °C in the Atlantic sector and Pacific Sector. For days when the observed T2M was <-25 °C, the daily mean difference between the reanalyses and buoys was, on average, +5.4 °C for ERA5 and +3.4 °C for ERA-I. The near surface warm bias in ERA5 and ERA-I may partly be attributed to the difference in height with observations. The larger warm bias in ERA5 during cold periods suggests this reanalysis also struggles to accurately simulate strong stable boundary layers, which frequently appear in winter and early spring, despite the higher vertical resolution compared with ERA-I (e.g., Beesley et al., 2000). It may also be also partly attributed to the simplified representation of snow and ice thickness in the reanalyses.

The total precipitation over Arctic sea ice in ERA5 was higher than in ERA-I in all seasons, amounting to an additional 20-40 mm more in most of the Arctic over a full year. Annual precipitation is higher in ERA5 especially in the Atlantic sector (by 40-100 mm). This is promising, as ERA-I is known to be drier in the Arctic compared with some other recent reanalyses (Lindsay et al., 2014; Merkouriadi et al., 2017; Boisvert et al., 2018). More critically, the snowfall is substantially higher in

ERA5 than in ERA-I in all seasons, especially during summer and autumn and especially in the Atlantic sector of the Arctic. In the Atlantic sector the annual snowfall in ERA5 is 80-200 mm water equivalent higher than in ERA-I. ERA5 has a higher snowfall to precipitation ratio than ERA-I, in particular during summer and autumn. ERA-I is known to have an anomalously large fraction of liquid precipitation (rain) and thus low snowfall to precipitation ratio in the Arctic, especially during August-September (Dutra et al., 2011; Leeuw et al., 2015). The total precipitation accumulated along the buoys drift trajectories, during the cold season (from 15 August/1 October until a buoy fails or until 30 April), was higher in ERA5 than in ERA-I for every buoy examined. The snowfall to precipitation ratio is on average 0.6 for ERA-I and 0.8 for ERA5 along buoy trajectories. This ratio in ERA5 is somewhat higher than in ERA-I for all buoys with an accumulation date starting from 1 October, and much higher than in ERA-I for buoys with accumulation starting from 15 August, likely due to anomalous autumn rainfall in ERA-I being now snowfall in ERA5. The total precipitation in ERA5 and ERA-I and the snowfall in ERA5 are closer to the SWE content of buoy measured snow pack, compared with the snowfall in ERA-I which is often much less, suggesting the total precipitation and snowfall in ERA5 are better represented. Nonetheless, the lack of representative in-situ observations and difficulty in measuring snow accumulation on sea ice in the Arctic makes it a challenge to accurately evaluate precipitation products over sea ice (e.g. Rasmussen et al., 2012; Lindsay et al., 2014; Sato et al., 2017; Blanchard-Wrigglesworth et al., 2018; Boisvert et al., 2018). ~~Given snow is such a critical factor in sea ice evolution, more representative observations are therefore needed (e.g. Merkouriadi et al., 2017; Webster et al., 2018).~~

The larger warm bias during the ice growth season in ERA5, compared with ERA-I, can result in a lower ice thickness when using this as a forcing product for an ice model or a cumulative FDD model. The higher precipitation and snowfall in ERA5 results in a thicker snow pack that allows less heat loss to the atmosphere. Overall, using a 1D thermodynamic sea ice model simulations with ERA5 had a thinner ice thickness compared with ERA-I at the end of the growth season with a combined effect of higher T2M and more snow. However, the effects on ice growth are very small, order of centimeters, during the freezing period. Given snow on sea ice is such a critical factor in sea ice evolution, more representative observations are therefore needed (e.g. Merkouriadi et al., 2017; Webster et al., 2018).

**Authors contributions**

CW, KW and MAG initiated the study. CW and MAG retrieved the buoy data. CW and KW downloaded and analysed the reanalysis data and performed the 1D model simulations. All authors contributed to writing and to revisions of the manuscript.

**Acknowledgements**

The authors would like to thank Alek Petty and one anonymous reviewer, whose comments significantly improved this manuscript. This study was supported by the Research Council of Norway through project SPARSE (project no 254765), the Fram Centre "Arctic Ocean" flagship program through the SOLICE project, and by the Centre for Ice, Climate and Ecosystems

(ICE) at the Norwegian Polar Institute through the N-ICE project. We wish to acknowledge ECMWF for ERA-Interim and ERA5 data, the International Arctic Buoy Program (IABP) for the IMB data (http://imb-crrel-dartmouth.org/results/), the Alfred Wegener Institute for Snow Buoy data (http://data.seaiceportal.de/gallery/index_new.php).

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

**Table 1. Summary of deployment locations and initial conditions for the buoys. The accumulated snow water equivalent (SWE) is given based on ERA-I, ERA5 and buoy data. The cumulative SWE TP is based on total precipitation assuming precipitation falls as snow when T2M is <0 °C. The cumulative snowfall (SF) is calculated in the same period as what did for the cumulative TP. The accumulated SWE measured by the buoy is estimated using a climatological monthly mean snow density based on Warren et al. (1999).**

| Buoy | Deployment location | Period of operation | Ice type | Initial thickness (m) | | Accumulated SWE (mm water equivalent) | | | | |
|---|---|---|---|---|---|---|---|---|---|---|
| | | | | | | ERA-I | | ERA5 | | Buoy |
| | | | | ice | snow | TP | SF | TP | SF | SWE |
| 2010A | Central Arctic | 20 Apr 2010 – 2 Dec 2010 | FYI | 1.67 | 0.24 | 77.5[B] | 51.8 [B] | 80.9 [B] | 78.5 [B] | 67.2[B] |
| 2011M | Central Arctic | 29 Sept 2011 – 22 Apr 2013 | MYI | 1.67 | 0.07 | 94.6[A] | 89.2 [A] | 99.8 [A] | 99.8[A] | 19.2[A] |
| 2012C | Central Arctic | 13 Apr 2012 - 4 Oct 2012 | FYI | 1.24 | 0.43 | 56.2[B] | 21.1 [B] | 65.1 [B] | 48.3[B] | NA |
| 2012D | Central Arctic | 4 May 2012 - 2 Nov 2012 | FYI | 1.67 | 0.47 | 89.9[B] | 47.1 [B] | 100.9 [B] | 91.8[B] | 124.2 [A] |
| 2012H | Beaufort Sea | 10 Sept 2012 - 16 Jan 2014 | FYI | 1.50 | 0.02 | 75.8[A] | 68.1 [A] | 83.7 [A] | 83.4[A] | 63.0[A] |
| 2012L | Beaufort Sea | 27 Aug 2012 - 25 Sept 2013 | MYI | 3.35 | 0.02 | 76.9[A] | 69.3 [A] | 90.4[A] | 90.4[A] | 12.8[A] |
| 2012I | Chukchi Sea | 14 Aug 2012 - 21 Dec 2012 | MYI | 1.09 | 0.10 | 94.8[B] | 71.1 [B] | 120.2 [B] | 111.4[B] | 98.0[B] |
| 2012J | Laptev Sea | 25 Aug 2012 – 11 Jan 2014 | MYI | 1.09 | 0 | 80.3[A] | 71.2 [A] | 94.4 [A] | 94.4[A] | 41.6[A] |
| 2013B | Central Arctic | 10 Apr 2013 - 19 Dec 2013 | NA | 2.00 | 0.02 | 151.3[B] | 104.0 [B] | 168.0 [B] | 146.8[B] | 36.4[B] |
| 2013E | Central Arctic | 11 Apr 2013 – 4 Oct 2013 | FYI | 1.40 | 0.05 | 57.4[B] | 17.8 [B] | 57.8[B] | 35.3[B] | NA |
| 2013H | Central Arctic | 3 Sept 2013 - 29 Dec 2013 | NA | 1.30 | 0.05 | 42.3[C] | 38.3 [C] | 61.7 [C] | 61.7[C] | 20.3[C] |
| 2014E | Central Arctic | 11 Apr 2014 - 18 Feb 2015 | NA | 1.73 | 0.19 | 182.6[B] | 122.9 [B] | 203.4 [B] | 192.4[B] | 103.6[B] |
| 2015D | Central Arctic | 10 Apr 2015 - 1 Feb 2016 | NA | 1.96 | 0.05 | 144.4[C] | 110.7 [C] | 176.3 [C] | 163.7[C] | 354.0[C] |
| s16 | Laptev Sea | 19 Sept 2015 - 20 Dec 2016 | FYI | NA | 0.07 | 123.6[A] | 107.6 [A] | 144.7 [A] | 144.7[A] | 80.0[A] |
| s20 | Central Arctic | 14 Sept 2015 – 19 Apr 2016 | FYI | 1.50 | 0.05 | 84.0[C] | 76.8 [C] | 89.6 [C] | 89.6[C] | ~6.0[C] |
| s29 | Laptev Sea | 10 Sept 2015 - 16 Oct 2016 | FYI | 1.20 | 0.01 | 108.5[A] | 95.9 [A] | 124.8 [A] | 124.8[A] | 20.0[A] |

NA: no data

[A]: from 1 October to 30 April.

[B]: from 15 August until the IMB fails or there is no snow data.

[C]: from 1 October until the buoy fails or there is no longer snow data during the first freezing season

**Table 2. The mean T2M, accumulated FDD, and estimated ice growth with FDD model**

| Buoy | T2M mean (°C) | | | FDD (K·d)[A] /ice growth (m)[B] | | |
|------|------|------|------|------|------|------|
| | ERA5 | ERA-I | Buoy | ERA5 | ERA-I | Buoy |
| 2011M | -22.5 | -24.2 | -26.6 | 4295/1.70 | 4662/1.78 | 5174/1.90 |
| 2012H | -22.5 | -24.1 | -25.8 | 4276/1.70 | 4624/1.78 | 4978/1.85 |
| 2012L | -22.1 | -23.1 | -24.9 | 4198/1.68 | 4402/1.73 | 4788/1.81 |
| 2012J | -20.8 | -20.8 | NA | 3902/1.61 | 3921/1.61 | NA |

NA: no data

[A]: from 1 October to 30 April.

[B]: ice growth estimation by the end of freezing season with the Lebedev FDD model (Maykut, 1986).

Table 3. Model runs and atmospheric forcing data in model simulations, where TP is total precipitation, SF is snowfall, V is wind at 10 m height, Rh is relative humidity, and CN is total cloud cover.

| Model runs | T2M | Precipitation | V, Rh, CN |
|---|---|---|---|
| TPI_T2MI | ERA-I | TP from ERA-I | ERA-I |
| TP5_T2M5 | ERA5 | TP from ERA5 | ERA5 |
| SFI_T2MI | ERA-I | SF from ERA-I | ERA-I |
| SF5_T2M5 | ERA5 | SF from ERA5 | ERA5 |
| TPI_T2M5 | ERA5 | TP from ERA-I | ERA-I |
| TP5_T2MI | ERA-I | TP from ERA5 | ERA-I |

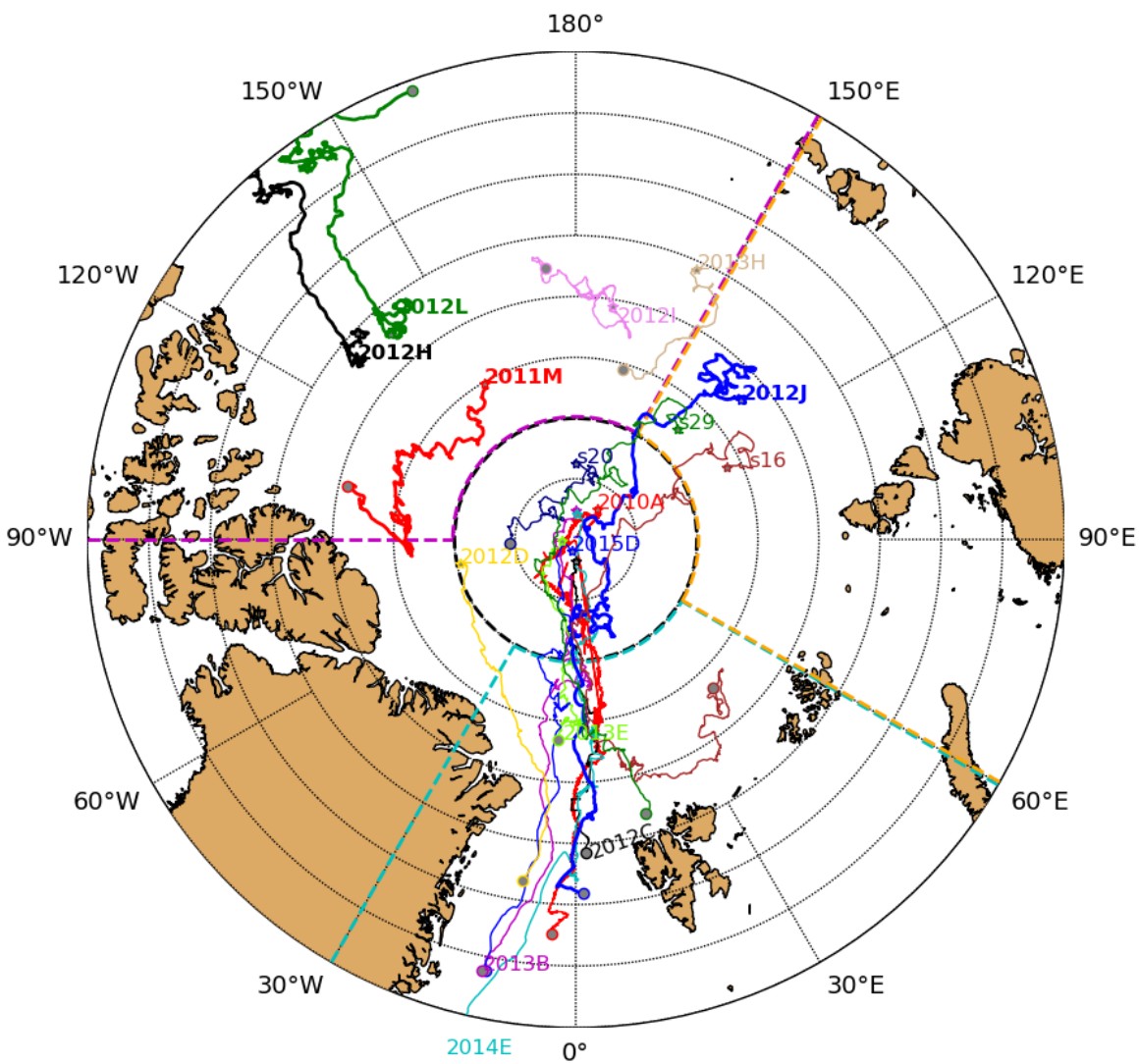

**Figure 1. Drift trajectories of all selected buoys (IMBs and snow buoys) in 2010 to 2015. Symbol "⋆" indicates the start of the drift and "o" signals the end of the drift. Buoys are labelled at the beginning or the end of the drift using same colour as trajectories. Buoys used for model simulations are highlighted with solid thick line and bold font. Dashed thick lines illustrate our definition for sectors: Central Arctic (black; north of 86° N), and south of 86°N: Pacific sector (magenta; 90° W-150° E), Atlantic sector (cyan; 30° W–60° E) and Laptev Sea (orange; 60° E – 150° E) used in Figure 6.**

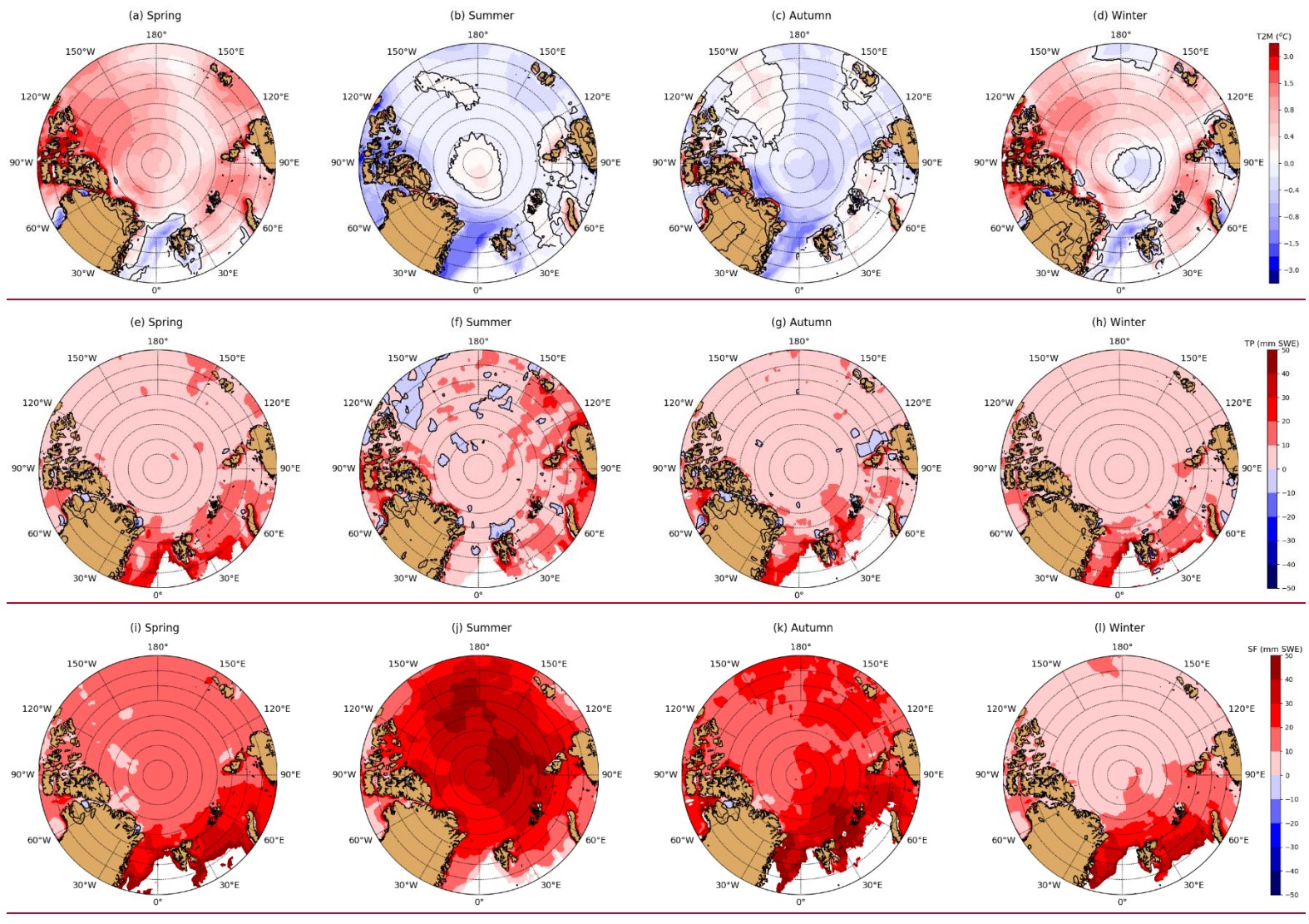

**Figure 2. Seasonal mean difference between ERA5 and ERA-I (ERA5-ERA-I) for T2M (a-d), total precipitation (e-h), and snowfall (i-l) in spring (a, e, i), summer (b, f, j), autumn (c, g, k) and winter (d, h, l) over Arctic sea ice during 2010-2015..**

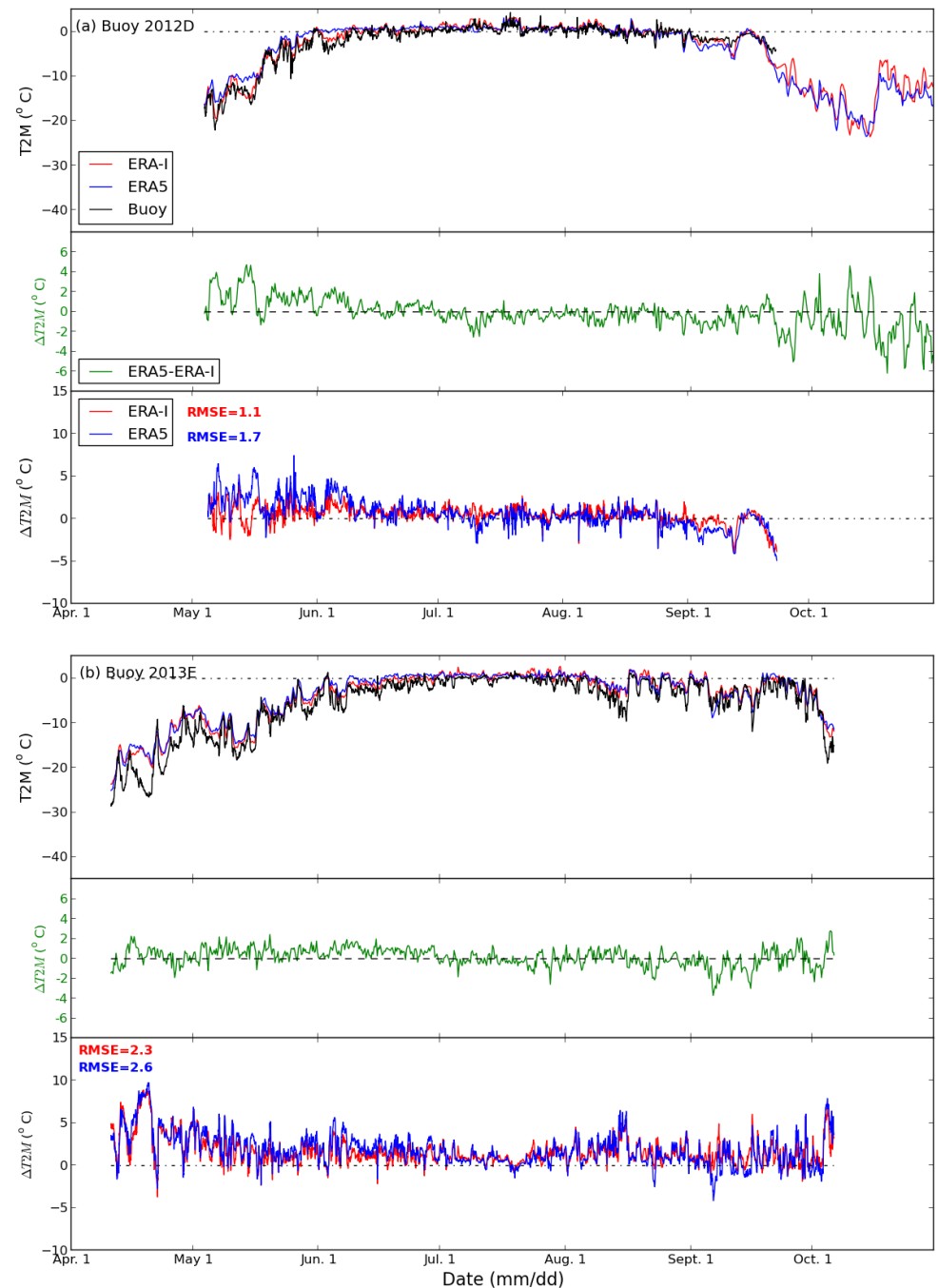

**Figure 3. Variation of 2 m air temperature (T2M) in ERA5, ERA-I and the buoys (upper panel) and the differences of T2M between ERA5 and ERA-I (mid-panel; green color) and comparisons for ERA5 and ERA-I with buoys (ERA5 minus buoy; ERA-I minus buoy) for buoys (a) 2012D and (b) 2013E.    RMSE values for the comparison between ERA products and buoys are shown as text, blue for ERA5-buoy, red for ERA-I-buoy.**

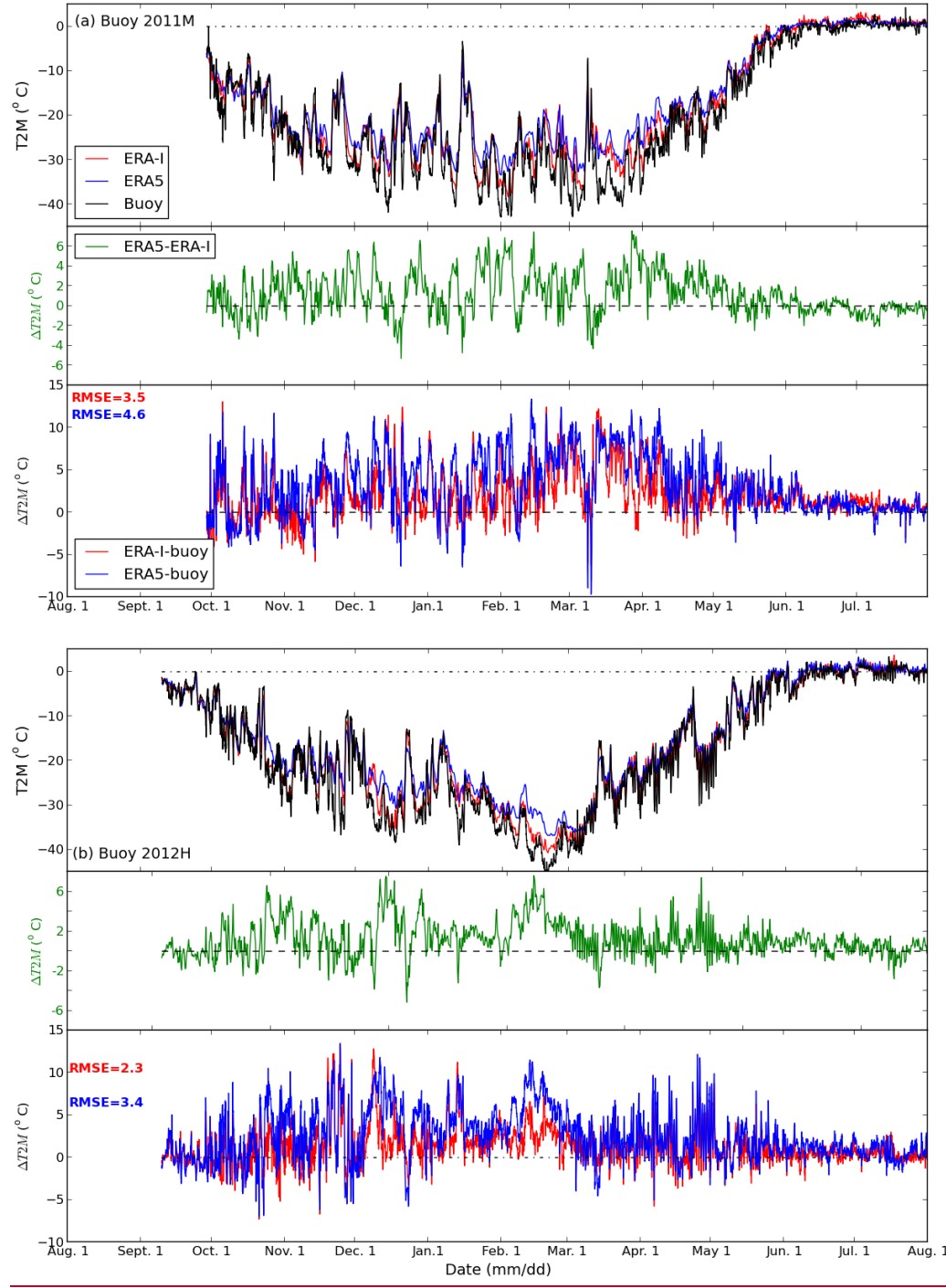

**Figure 4.** Same as Fig. 3, but for (a) buoy 2011M and (b) buoy 2012H.

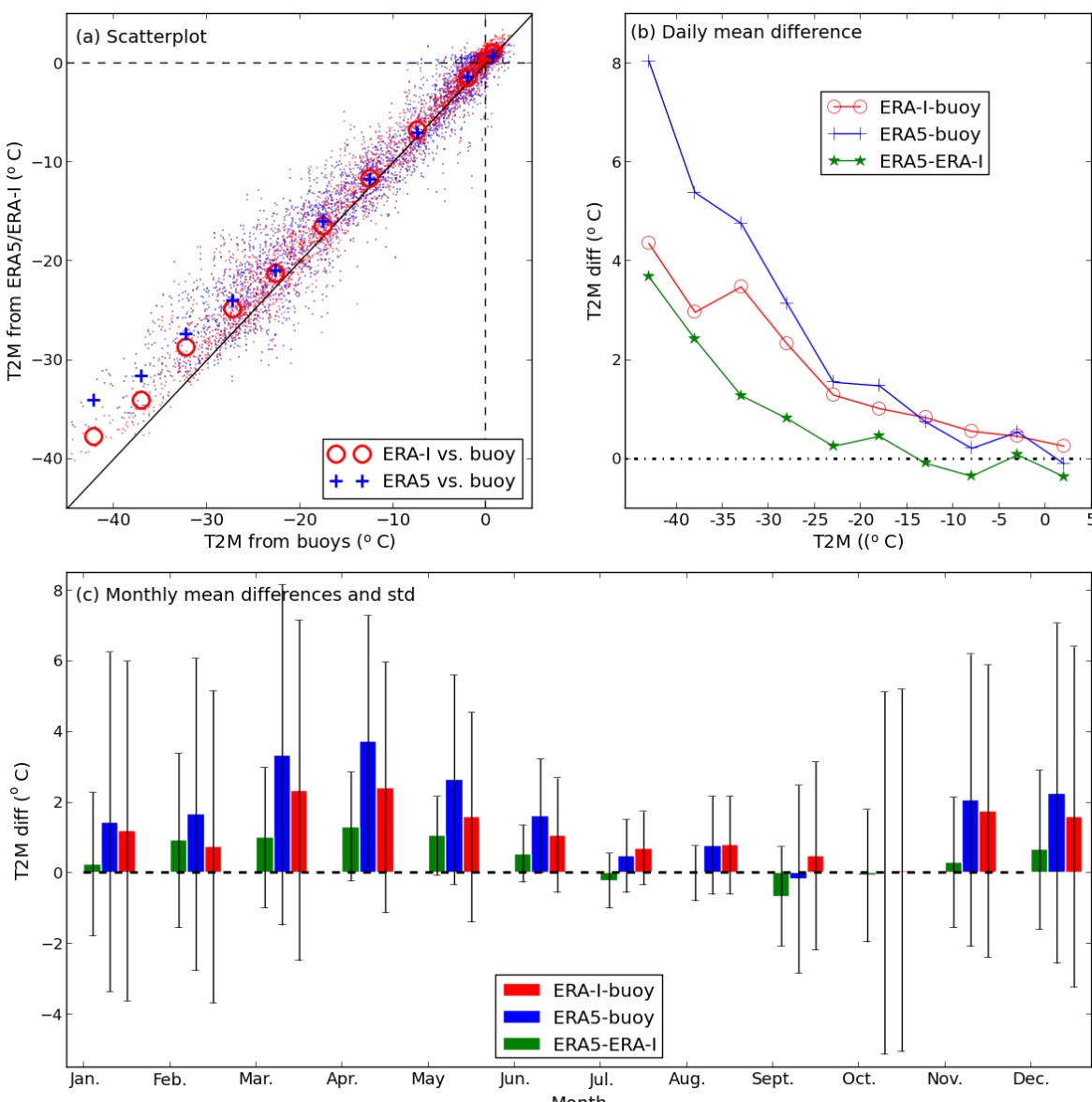

Fig. 5. Statistics of T2M from ERA5, ERA-I and all the buoys. (a) Scatter plot for all data (small dots) and average T2M at 5 degree bins between -45 °C and +5 °C, (b) Daily temperature differences between the reanalysis and between the reanalysis and the buoys corresponding to 5 degree bins between -45 °C and +5 °C, and (c) monthly mean differences and standard deviation (std). In panel a, the black solid line is for 1:1.

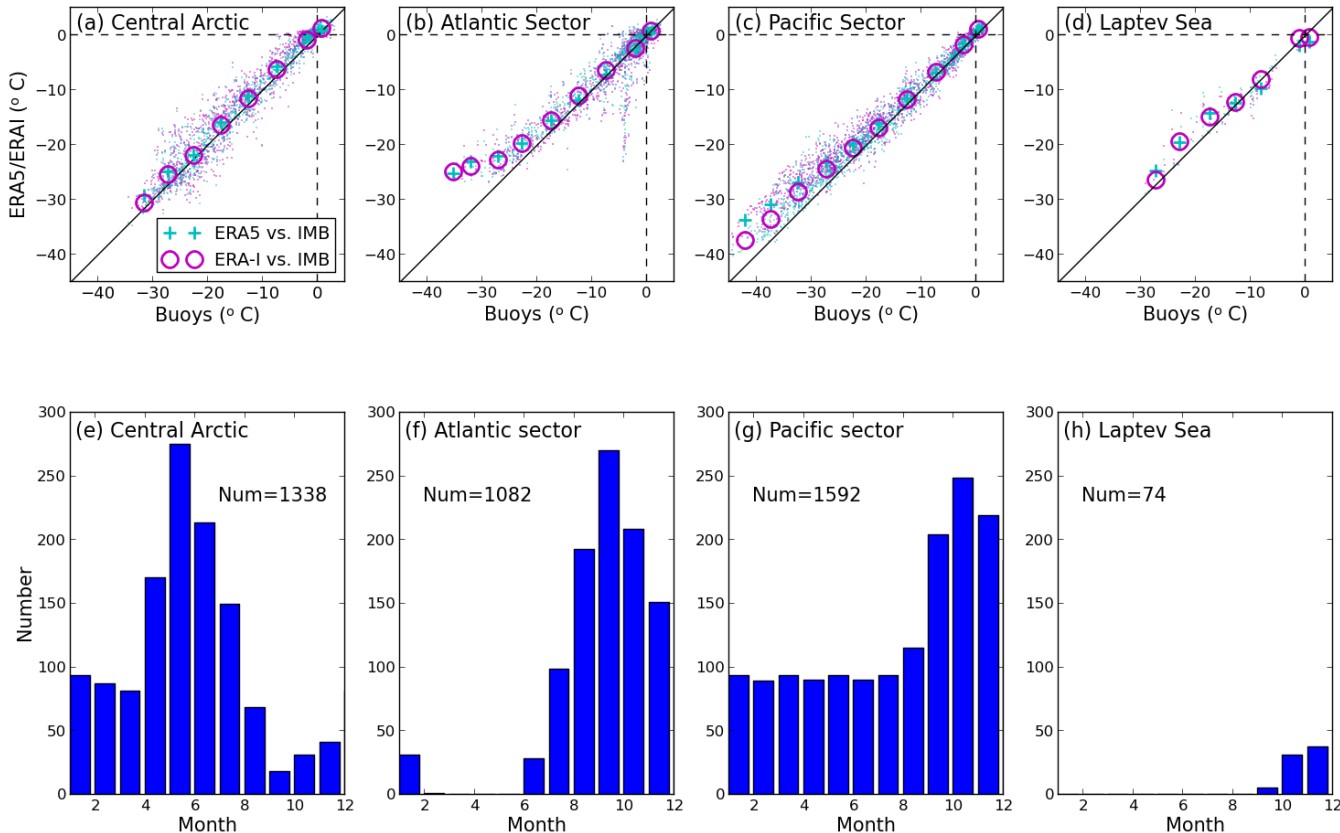

**Figure 6. Scatter plot of T2M from ERA5 and ERA-I vs. from buoys for (a) Central Arctic, (b) Atlantic sector, (c) Pacific sector, and (d) Laptev Sea, and number of buoy data (daily) per month for (e) Central Arctic (f) Atlantic sector, (g) Pacific sector, and (h) Laptev sea. The definition of sectors are shown in Figure 1.**

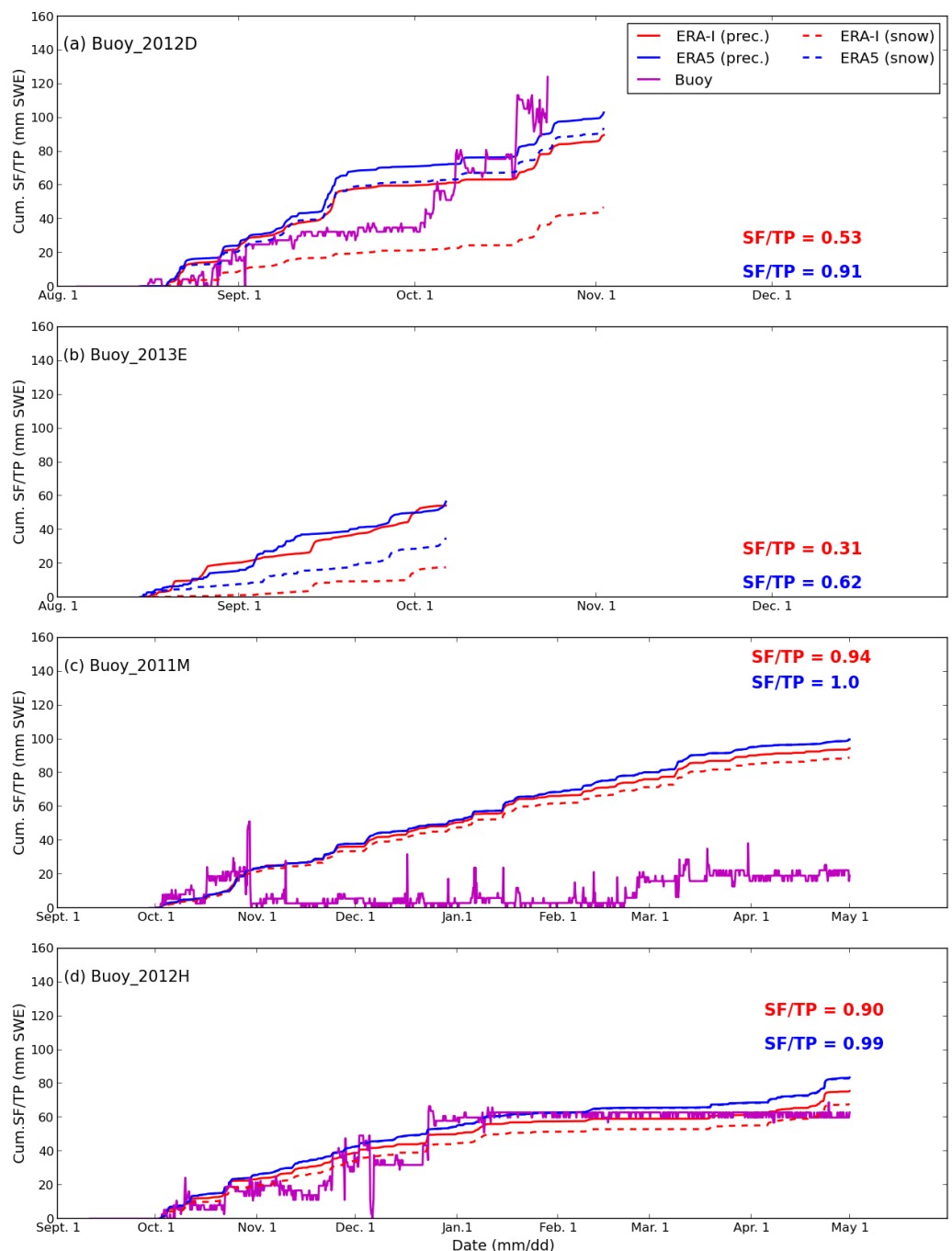

**Figure 7. Cumulative total precipitation (TP) and snowfall (SF) for ERA5 and ERA-I and snow depth for buoys (a) 2012D, (b) 2013E, (c) 2011M, (d) 2012H. Accumulation starts from 15 August for panels (a) and (b), and from 1 October for panels (c)-(d). The ratio of snowfall to total precipitation (SF/TP) in ERA5 (blue text) and ERA-I (red text) is also shown in the figure.** Note there was no
5    snow depth data for buoy 2013E during the accumulation period.

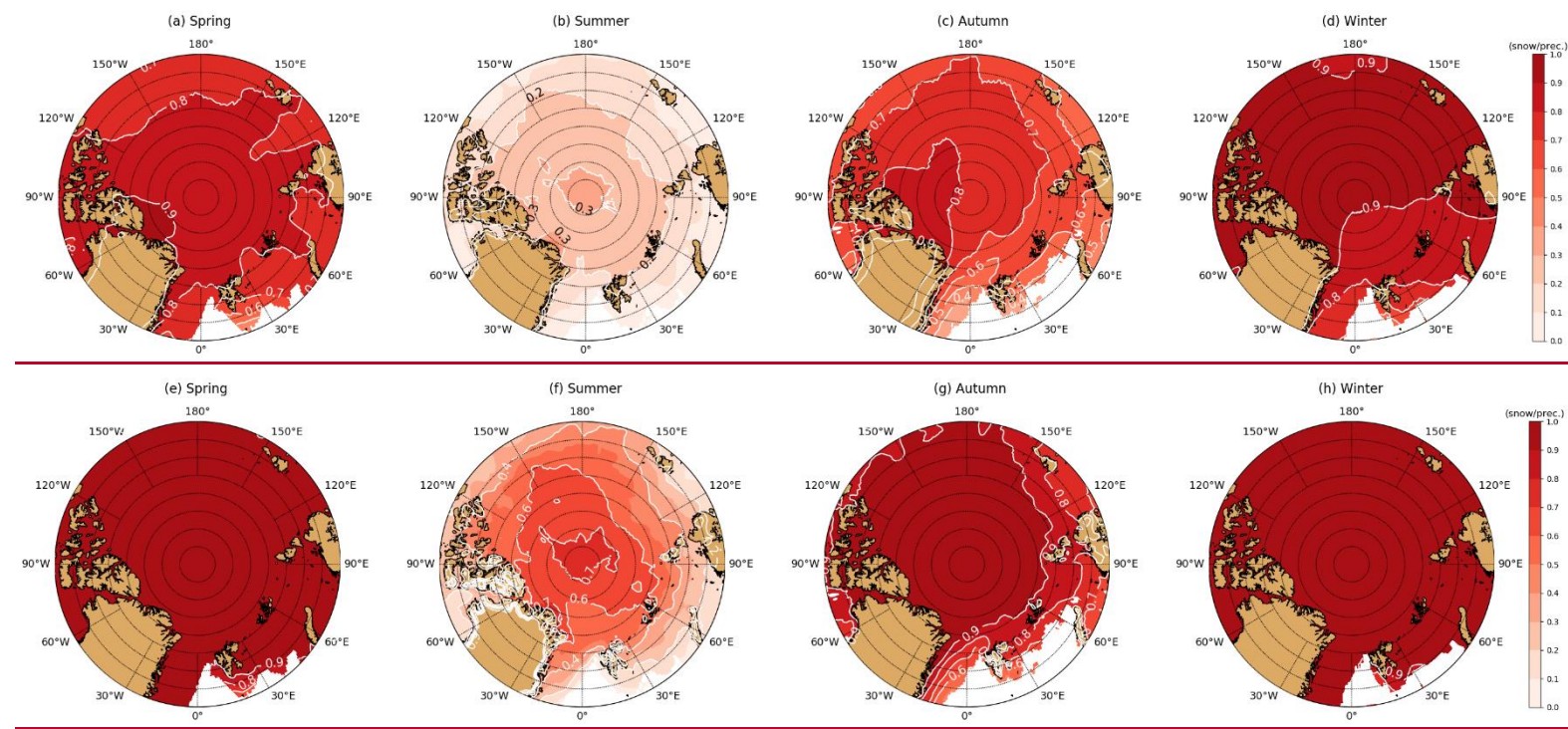

**Fig. 8 The ratio of snowfall to total precipitation (SF/TP) in ERA-I (a-d) and ERA5 (e-h) in spring (a, e), summer (b, f), autumn (c, g), and winter (d, h)**

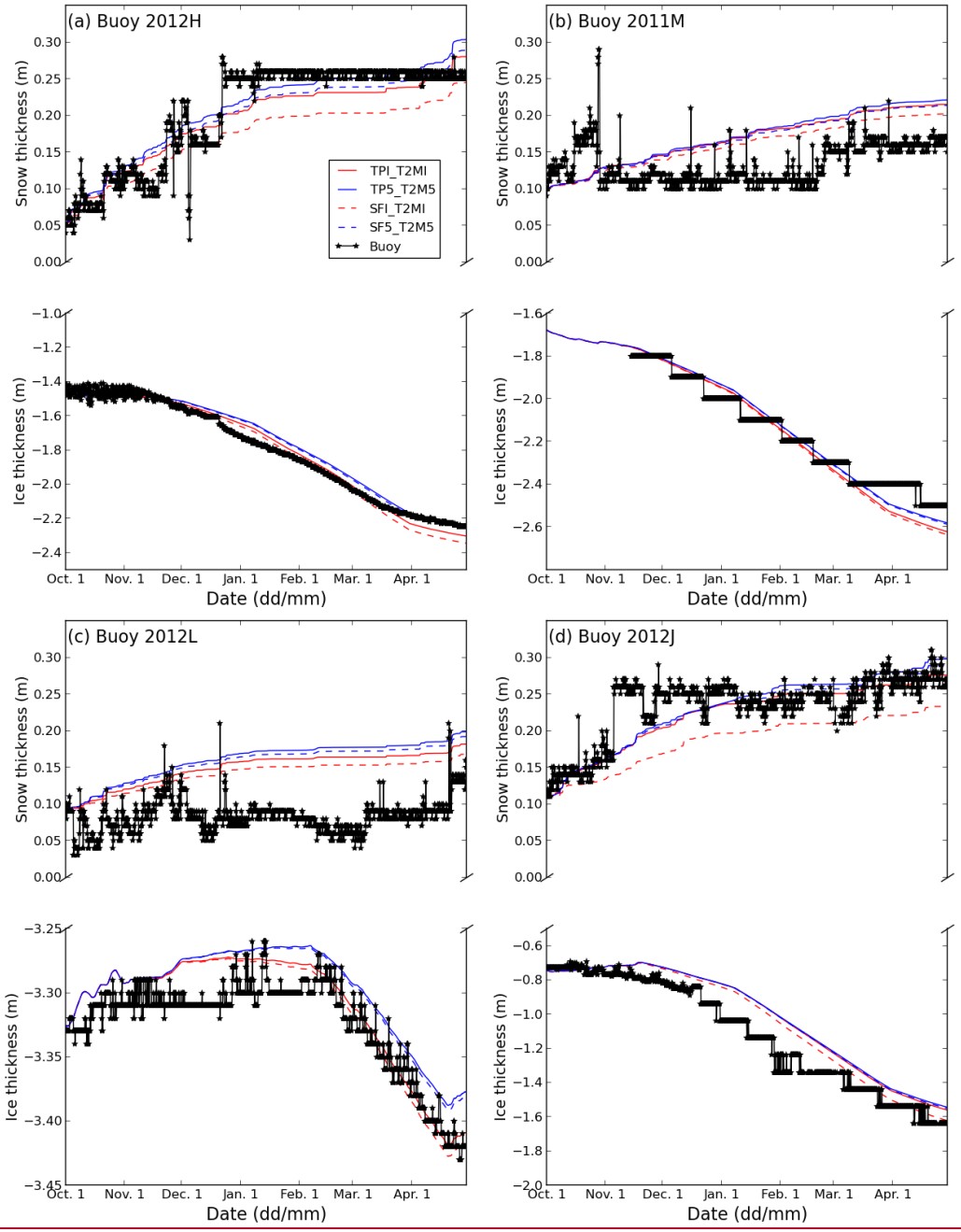

**Fig. 9.** Evolution of snow and sea ice thickness during freezing season based on simulations with HIGHTSI for (a) buoy 2012H, (b) buoy 2011M, (c) buoy 2012L, and (d) buoy 2012J for runs TPI_T2MI, TP5_T2M5, SFI_T2MI, and SF5_T2M5.

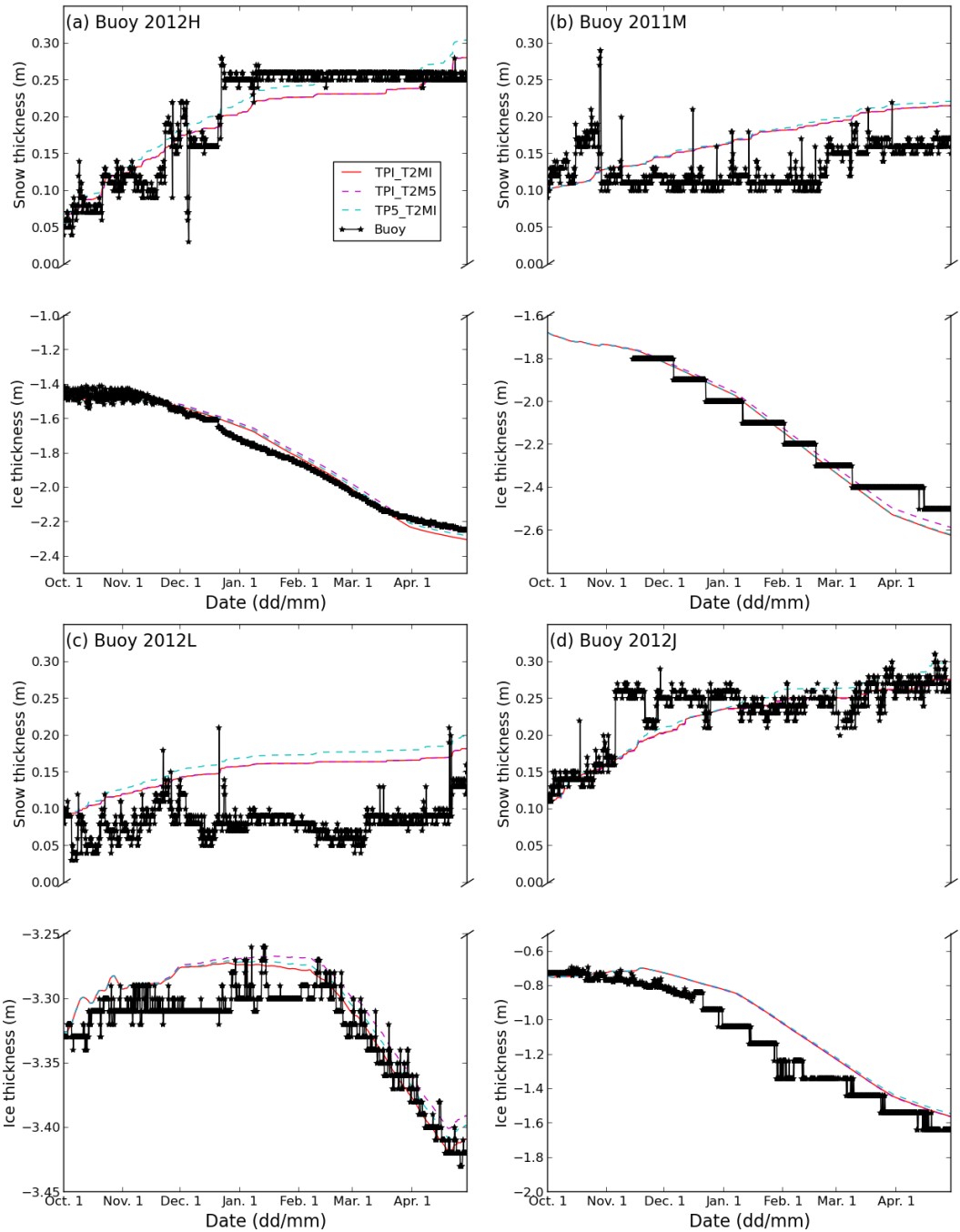

**Fig. 10.** Same as Fig. 9, but for model runs TPI_T2M5 and TP5_T2MI.