# Peer review of "Comparison of ERA5 and ERA-Interim near surface air temperature, snowfall and precipitation over Arctic sea ice: Effects on sea ice thermodynamics and evolution"

_The Cryosphere, 2018_

## Referee Comment (RC1) · Anonymous Referee #1 · 4 Jan 2019

This paper described a comparison of two reanalyses ERA-Interim and the newly released ERA5, which is supposed to replace ERA-Interim. Two meter temperatures and precipitation (both total and snowfall) were compared with multiple IMB and snow buoys between 2010-2015 in the Arctic Ocean. Both reanalyses produced too warm air temperatures when conditions were below -25C compared to the buoys. Although the accuracy of precipitation, especially snowfall is difficult to assess in the Arctic due to the lack of in situ data available, a method of snow water equivalent was used to compare the buoy SWE with those from reanalyses. In some cases the reanalyses

produced too much SWE and some cases produced too little. ERA5 appeared to produce less precipitation compared to ERA-interim overall. A freezing degree day, and a simple 1-d thermodynamic model were also used to assess how the errors in the air temperatures and snowfall would affect sea ice growth in models because models are forced with reanalysis data. Precipitation was found to matter more in the sea ice pack growth compared to the 2-m air temperatures.

This paper was well written, and provided a good background. The results are also of importance because ERA5 is a new reanalysis and snow on sea ice in the Arctic is such an uncertainty. I did not see any major issues with this paper, just a few minor comments below.

In the abstract, I would not say that ERA-I is drier than 'most' reanalyses, I will say ERA-I is drier than 'some' reanalyses - see Boisvert et al., 2018 Journal of Climate

Figure 3 caption. Do you mean panel (D), not (K)? Because there is no panel K in the figure.

It would be great to see a little more conversation dealing with the differences in Temp and Precipitation compared to the buoys and to themselves. It seemed like some regions where the buoys were/times of the year produce larger differences between the buoys and the reanalyses. For example, there appeared to be larger differences between realanyses and the buoys in the Beaufort sea areas.

Figures 2 and 3. It would be beneficial to also have the differences between ERA5 and ERA-I and the buoy temperatures perhaps in a different figure? Because it is a little hard to see how well the reanalyses compare with the buoys the way it is now. Or perhaps provide a table with the differences and biases for each buoy.

Page 4, line 16: Might be best to say where these 2 buoys are located in the text. 2013 E and 2012 J? Perhaps the reanalyses are better at producing accurate temperatures in certain regions of the Arctic and perhaps this could be elaborated on more.

[Figure]

I know that snow depths are fairly uncertain, but perhaps instead of taking a constant snow density of 350 kg/m3, why not time vary it throughout the winter season and based on locations based on the Warren climatology. This might improve your results.

Line 18 page 6 should be ERA5

---

## Referee Comment (RC2) · Petty (Referee) · 15 Jan 2019

This paper presents an analysis of ERA-Interim and the newly released ERA5 reanalysis over Arctic sea ice. The study mainly involves comparing the near surface air temperature and precipitation fields against data collected by drifting buoys (IMBs and Snow Buoys). The warm bias in ERA-I for low near surface air temperatures was shown to be higher in ERA5 (unfortunately!). The precipitation results were more mixed, as ERA5 seemed to have lower total precipitation but higher snowfall compared to ERA-I (a big increase in the snowfall/total precip ratio). A simple Freezing Degree Day (FDD)

[Figure]

model and a more sophisticated 1D model (HIGHTSI) were used to assess the impact of these differences on simulations of Arctic sea ice mass balance.

In general, the paper was well written and included a clear motivation/set of objectives and data/model descriptions, along with some useful figures and discussion. The study is obviously timely considering the potential utility of ERA5 for Arctic studies in the coming months/years and the lack of current assessment efforts. The study was rather simple in its objectives, however (i.e. it wasn't exactly a complete inter-comparison of ERA5 and ERA-I over Arctic sea ice), and some of the results and surrounding discussion were not as insightful as they could have been. My view is that the paper should be published once some improvements have been made.

General comments

I think it would help to show some more general comparisons between ERA5 and ERAI over the Arctic Ocean/sea ice, e.g. raw and difference maps/time series of air temperature, snowfall, precip, pressure. These could be just annual means but seasonal means might be good to see too. This could be included in the SI but I think it will be valuable to include in the main paper to help motivate the study (are there any big/obvious differences from the off?!). This doesn't need to be too detailed.

I was a bit disappointed in the FDD analysis and am unsure of its value. The main conclusion seems to be that the warm bias introduces a negative thickness bias, which is pretty obvious without the need for an FDD model. . .I would be tempted to drop this section entirely unless you can make it seem more value-added compared to the 1D modelling study that follows this (and does seem valuable despite my concerns).

To me it's a shame you didn't show a complete regional Arctic sea ice model forced by both reanalyses as this could have been a useful way of showing regional biases in the reanalyses!

While I think the HIGHTSI section is useful and needed, I think it needs to be re-written

and potentially expanded on to improve clarity. e.g.: - I know you cite those model papers but I think you to include at least a brief description of the model here and how those forcings were used. Why no downwelling for example? This is calculated form the cloud cover? I think you can provide some general information then cite the papers for more information. - I got pretty confused by the use of Sp and Sf and think there could be some mistakes here. Why did you start by comparing the Sp runs and not the Sf runs? Sp is a bit of a confusing acronym so maybe you could try something like Tp2mt? - Can you not also show some idealized Arctic mean simulations instead of just the buoy track simulations? This might give us a better sense of what the potential impact of these differences might be when we want to consider the Arctic as a whole. - 'a good representation of precip seems crucial' seems like a pretty loose interpretation of the analysis you presented. I think this discussion needs to tie back better to what exactly your results demonstrated.

I think you need to better justify early on why you only look at these two variables. Maybe mention earlier that you also looked at MSLP in ERAI and ERA5 (reanalyses tend to agree more in this regard as expected) and that you're limited by what the buoys can provide? You later force the HIGHTSI model with other variables (e.g. cloud cover) so I think you should show these and their differences too, despite the lack of buoy obs to validate it against.

Comparing 2 m and 10 m air temps might be illuminating. Any change in how the 2 m air temps are calculated in ERA5 (still not an explicit model level, right?).

Confused why you need to interpolate the ERAI data to the ERA5 grid before interpolating to the buoy position. Guessing this won't be a big issue as ERA5 is of intermediate resolution but still seems odd to me.

How do you deal with the temporal differences between ERAI and ERA5? ERA5 is hourly and ERA-I is 6-hourly?

I'm confused why you don't show the actual ice thickness for the buoys (I think you just

show the estimated ice thickness change from the FDD model?). Also confused as to whether you initialize the FDD model with zero ice thickness or not, as you show ice growth, not ice thickness. Any reason for this? Again, I see little value in this analysis so suggest dropping this and improving the rest of the analysis presented in the paper.

Specific comments

P1 L3-4: 'The decline of Arctic sea ice has been attributed to various interrelated causes, including a general overall warming trend (Steel et al., 2008; Polyakov et al., 2010).' Seems pretty vague so would recommend you either improve this or drop it.

P1 L10: I would replace 'in-situ atmospheric observations' with something like 'direct observations of the atmosphere, sea ice and ocean conditions'?

P1 L15: I suggest you combine this line 'Atmospheric reanalyses etc..' with the one about their use earlier on L12-13.

P1 L22: I think you need to make the point here that the 1950 onwards data isn't yet available yet? Unless you're guessing it will be at the time of publication. . . I also think you need to provide a better discussion of these supposed improvements and how they might increase ERA5's utility for Arctic studies, e.g.: - What do you mean by improved representation of troposphere and global balance? - More consistent? How?

P3 L20: Where were these buoys deployed?

P4 L23: Boisvert not Biosvert

P6 L3-4: And any thoughts on how the cloud physics might have changed in ERA5 to cause this big change in snowfall/precip ratio? I think you should plot this ratio out as a separate figure as it seems like a crucial part of the story.

P6 L4-5: I'm not quite sure how Figure 5 shows it is less anomalous as these are just showing the reanalysis data not compared to anything. Think you need to plot the buoy results too despite the big issues of representation etc.

P6 L10: why not use a daily climatology of density? You cite Warren1999 for the 350 value but this seems overly simplistic considering the results presented in Warren1999.

P8, L4-5: can you briefly describe what this ocean heat flux is? E.g. 2 W/m2?

P9, L30: drop the warm summer bias comment here as you repeat it later.

P10, L1-3: think you should mention the caveat here that the buoy probably isn't giving the 2 m air temperatures. Can we be sure resolving the boundary layer is the actual problem here?

P10, L5-14: I think this needs a bit of rewording for clarity. Really worth stressing that the total precip is lower but the snowfall is higher, right?! I would start with that difference then explain what it means in terms of the comparisons with the buoys. - Think you also need to make the point later regarding which precip was used to force the 1D model and that care needs to be taken regarding how precip is used in the products perhaps. Any particular recommendations here? I.e. do you think we should be using the snowfall product or deriving this from the total precip? There are other ways of doing this also (using higher level temps).

Figure 2: why does the green line seem dashed? Can you move the difference line lower so it's easier to see? - I also think you should show not the difference between ERA5 and ERA-I but two lines representing the differences between the reanalyses and the buoys. Maybe just pick a couple as good examples of the seasonal cycles you mention in the text and make these bigger/clearer, then put the rest in the supplementary information? As it is, it's hard to really get a sense of what these figures show quantitatively.

Figure 4: this is a good figure!

Figure 5: you don't need the second y-axis, just state in the legend that the dashed lines are snow. You should add the units to the label and legend. Why does this not include the buoys change in snow depth? As a second y-axis?! Or converted to

cumulative precip and plotted on the same axis.

Figure 7: I don't think you need to show the FDD values as they don't mean much physically.

Figure 8: why the weird staircase in buoy 2011M? Lower temporal resolution for some reason?

---

## Author Comment (AC1) · 1 Mar 2019

**Supplementary Information**

Caixin Wang1, 2, Robert M. Graham2, Keguang Wang1, Sebastian Gerland2, Mats A. Granskog2

1Norwegian Meteorological Institute, 9293 Tromsø, Norway

2Norwegian Polar Institute, Fram Centre, P.O.Box 6606 Langnes, 9296 Tromsø, Norway

Correspondence to: Caixin Wang (caixin.wang@npolar.no)

The supplementary Information includes 7 figures. Figure captions are:

Figure S1. Variation of 2 m air temperature (T2M) in ERA5, ERA-I and the buoys and the difference of T2M between ERA5 and ERA-I for buoys (a) 2010A, (b) 2012C, (c) 2013B, (d) 2014E, and (e) 2015D.

Figure S2. Same as Figure S1, but for buoys (a) 2012I, (b) 2012J, (c) s16, (d) s20, and (e) s29. Note there is no buoy data in Figure (b) for buoy 2012J.

Figure S3. Variation of T2M differences between ERA5/ERA-I and buoys for buoys (a) 2010A, (b) 2012C, (c) 2012I, (d) 2013B, (e) 2014E, (f) 2015D, (g) s16, (h) s20, and (i) s29.

Figure S4. Cumulative total precipitation (prec.) and snowfall (snow) for ERA5 and ERA-I and snow depth for buoys (a) 2010A, (b) 2012C, (c) 2013B, (d) 2014E, and (e) 2015D. Accumulation starts from 15 August. Note that Buoy\_2012C does not have snow depth.

Figure S5. Same as Figure S4, but for buoys (a) 2012I, (b) 2012J, (c) s16, (d) s20, and (e) s29, and accumulation starts on 1 October.

Figure S6. Cumulative FDD and estimated ice growth using cumulative FDD model along the trajectories of (a) buoy 2011M, (b) buoy 2012H, (c) buoy 2012L, and (d) buoy 2012J for freeze-up from 1 October.

Figure S7. Forcing data of wind speed (V), relatively humidity (Rh), total cloud (CN) and ocean heat flux (fw) used in the model.

---

## Author Comment (AC2) · 1 Mar 2019

**We would like to thank the reviewer for the evaluation of our study and the constructive comments that helped us to improve the manuscript. Please find below the reviewer's comments in black font and the author's response in blue font.**

**Responses to Referee Alek Petty**

- In the revised manuscript, we added one new subsection 3.1 to show the seasonal difference of T2M, total precipitation and snowfall over the pan-Arctic sea ice and added one new figure (Figure 2). Accordingly, the original section 3.1 changed to 3.2, and the original subsections 3.1 and 3.2 changed to 3.2.1 and 3.2.2, respectively.
- We replotted the original Figure 2 and 3 for clarity, which are Figure 3 and 4 in the revised manuscript:
  - In the new Figure 3 we show the variation of T2M from ERA5, ERA-I and buoys and the differences of T2M between ERA5 and ERA-I for 5 buoys, plots for the other buoys provided in the Supplementary Information as Figure S1 and S2.
  - In the new Figure 4 we show the variation of T2M differences between ERA5/ERA-I and buoys for same 5 buoys as in the new Figure 2. Plots for the other buoys are provided in the Supplementary Information as Figure S3.
- We added a new Figure 6 to show the T2M difference between ERA5/ERA-I and buoys in four regions (Central Arctic, Atlantic sector, Pacific sector, and Laptev Sea) in the Arctic. Correspondingly, we added one paragraph at the end of the original section of 3.1 to describe the regional differences.
- We replotted the original Figure 5 and 6, the new figure named as Figure 7 and Figure S4-5. In these new figures, we show the accumulated total precipitation and snowfall from ERA5 and ERA-I and snow depth measured by buoys.
- The original Figure 7 (showing the FDD and sea ice growth) was moved to Supplementary Information as Figure S6 and the discussion on the FDD model was shortened.
- Forcing data of wind speed (V), relative humidity (Rh) and total cloud (CN) and ocean heat flux were plotted and provided in the Supplementary Information as Figure S7.

For our specific responses please see below.

General comments

I think it would help to show some more general comparisons between ERA5 and ERAI over the Arctic Ocean/sea ice, e.g. raw and difference maps/time series of air temperature, snowfall, precip, pressure. These could be just annual means but seasonal means might be good to see too. This could be included in the SI but I think it will be valuable to include in the main paper to help motivate the study (are there any big/obvious differences from the off?!). This doesn't need to be too detailed.

We now include a new figure showing seasonal mean differences of T2M, total precipitation and snowfall between ERA5 and ERAI over the Arctic (Figure 2). This figure is discussed in a new subsection 3.1 The new Figure 2 is shown below and the added subsection 3.1 reads as,

**"3.1 Spatial distribution of seasonal difference of reanalysis near surface temperature and precipitation**

Figure 2 shows the seasonal mean differences of T2M, total precipitation and snowfall between ERA5 and ERA-I over Arctic sea ice during 2010-2015. We classify spring as March, April and May, summer as June, July and August, autumn as September, October and November, and winter as December, January and February. The seasonal mean ice extent is obtained from the monthly sea ice concentration from NOAA/NSIDC during 2010-2015 (Meier et al., 2017).

The difference in T2M between ERA5 and ERA-I clearly varies with season (Fig. 2a-d). ERA5 is generally warner than ERA-I in spring and winter, and colder than ERA-I during summer and autumn over most regions of Arctic sea ice. These temperature differences are small during summer, but large during the other seasons. Near the North Pole, ERA5 is warmer than ERA-I in summer, but colder than ERA-I in winter. Whether warmer or colder, the differences between ERA5 and ERA-I are small (<±0.4 °C) in this region.

ERA-I is known to be a relatively "dry" global reanalysis product in the Arctic compared with most other modern reanalyses (e.g. MERRA-2, CFSR, and JRA-55) (Lindsay et al., 2014; Merkouriadi et al., 2017; Boisvert et al., 2018). However, the total precipitation in ERA5 is lower than in EAR-I over Arctic sea ice in all seasons (Fig. 2e-h). The lower precipitation in ERA5 is most pronounced in summer, and in the eastern Arctic. Differences in the snowfall between ERA5 and ERA-I are smaller than for total precipitation (Fig. 2 i-j vs. Fig. 2e-h). The snowfall in ERA5 is lower than in ERA-I in spring, autumn and winter, but larger than ERA-I in summer."

[Figure]

**Figure 2. Seasonal mean difference (ERA5-ERA-I) of T2M (a-d), total precipitation (e-h), and snowfall (i-l) in spring (a, e, i), summer (b, f, j), autumn (c, g, k) and winter (d, h, l) over Arctic sea ice. The mean ice extent in the season is used for classification of sea ice and open ocean.**

I was a bit disappointed in the FDD analysis and am unsure of its value. The main conclusion seems to be that the warm bias introduces a negative thickness bias, which is pretty obvious without the need for an FDD model. . .I would be tempted to drop this section entirely unless you can make it seem more value-added compared to the 1D modelling study that follows this (and does seem valuable despite my concerns).

We understand these concerns. We have rewritten this section to make it much more concise, and moved Figure 7 to Supplementary Information (now Figure S6). While it is intuitive that a warm bias introduces a negative thickness bias, we believe it is important to give an indication of how large we expect the magnitude of this bias to be based on a simple analytical model. Therefore we did not wish to entirely delete / move this section to supplementary material.

The revised text in this section reads as follows,
 "The cumulative freezing degree days (FDD) model only needs air temperature as input and is often used to estimate sea ice growth ($\Delta h$) from zero (e.g., Huntemann et al., 2014; Lei et al., 2017). The sea ice growth is estimated based on Lebedev (Maykut, 1986), $\Delta h = 1.33 \sum (FDD)^{0.58}$, where $\sum FDD$ is daily average temperature below the freezing point of sea water (-1.8 °C), integrated over the time period from 1 October to 30 April.

The positive near surface air temperature bias in ERA5 and ERA-I results in a negative ice thickness bias at the end of the growth season. The cumulative FDD is smallest for ERA5 (Fig. S6, Table 2), corresponding to the largest warm bias in ERA5 during the freezing season. The differences in FDD between ERA5, ERA-I and buoys are large for buoys 2011M, 2012H and 2012L, but negligible for buoy 2012J. The ice growth is 0.08-0.12 m less, with a mean of -0.09 m for ERA-I T2M, and 0.13-0.20 m less, with a mean of -0.16 m for ERA5 T2M compared to when using buoy temperatures (Table 2).".

To me it's a shame you didn't show a complete regional Arctic sea ice model forced by both reanalyses as this could have been a useful way of showing regional biases in the reanalyses!

We agree that a complete regional Arctic sea ice model forced by both reanalysis is a useful way to show regional biases in the reanalysis. The work is our next step. However, in this study, the focus is on comparing reanalysis with buoy observations.

While I think the HIGHTSI section is useful and needed, I think it needs to be re-written and potentially expanded on to improve clarity. e.g.: - I know you cite those model papers but I think you to include at least a brief description of the model here and how those forcings were used. Why no downwelling for example? This is calculated form the cloud cover? I think you can provide some general information then cite the papers for more information.

We now provide further details on the HIGHTSI model and how it was forced.

"The snow and ice temperature regimes are solved by the partial differential heat conduction equations applied for snow and ice layers, respectively. The turbulent surface fluxes are parameterized taking the thermal stratification of the atmosphere surface layer into account. Downward short- and longwave radiative fluxes are parameterized, based on the total cloud cover."

We also now show the forcing data in Figure S7.

[Figure]

**Figure S7.  Forcing data of wind speed (V), relatively humidity (Rh), total cloud (CN) and ocean heat flux (fw) used in the model.**

I got pretty confused by the use of Sp and Sf and think there could be some mistakes here. Why did you start by comparing the Sp runs and not the Sf runs? Sp is a bit of a confusing acronym so maybe you could try something like Tp2mt?

We apologize for this confusion. We have introduced a new naming system for the model runs. We now use TP to indicate total precipitation, and SF for snowfall. "I" indicates forcing is from ERAI, and "5" means it is from ERA5. Thus TPI_T2MI means total precipitation and T2M are both from ERAI. Further details are provided in Table 2, and the manuscript text.

Can you not also show some idealized Arctic mean simulations instead of just the buoy track simulations? This might give us a better sense of what the potential impact of these differences might be when we want to consider the Arctic as a whole.

Our simulations for the four buoys already cover FYI and MYI with a range of snow depths and ice thicknesses. We believe that these simulations already provide a realistic indication of the potential impacts of these differences over the wider Arctic, without the need for idealized simulations. For such idealized simulations we would neither have any direct observations to compare their validity, and such we do not believe they are of great value, as the focus in this study is to compare reanalysis to buoy observations.

'a good representation of precip seems crucial' seems like a pretty loose interpretation of the analysis you presented. I think this discussion needs to tie back better to what exactly your results demonstrated.

We have rewritten this sentence as

"Thus, not only the magnitude but also the frequency of the precipitation in the reanalysis data is crucial for the snow evolution in the simulation."

I think you need to better justify early on why you only look at these two variables. Maybe mention earlier that you also looked at MSLP in ERAI and ERA5 (reanalyses tend to agree more in this regard as expected) and that you're limited by what the buoys can provide?

We focus on these two variables because these are the observations we have from the buoys. The final paragraph of the introduction has been rewritten as follows:

"In this study, we compare and evaluate the performance of ERA-I and ERA5 over Arctic sea ice. For this, we use data from Ice Mass Balance buoys (IMB) (Perovich et al., 2018) and Snow Buoys (Grosfeld et al., 2016; Nicolaus et al., 2017) deployed in 2010-2015. The buoys record position, the 2 m air temperature (T2M), mean sea level pressure (MSLP), and snow depth at regular intervals. Hence, these observations can be used to evaluate the variables of T2M, precipitation and MSLP in the reanalyses. The former two variables are critical parameters for sea ice simulation (Cheng et al., 2008; Wang et al. 2015), and form the focus of our study. We use the T2M and snow depth observations from these buoys to assess the performance of ERA5 and ERA-I over Arctic sea ice. We further use the reanalyses to force a 1-D thermodynamic sea ice model. The simulations are compared with snow and ice thickness observations from the buoys to evaluate how differences in the T2M and precipitation influence the evolution of sea ice in the model."

You later force the HIGHTSI model with other variables (e.g. cloud cover) so I think you should show these and their differences too, despite the lack of buoy obs to validate it against. These variables are shown in Figures S7 in the Supplementary Information.

Comparing 2 m and 10 m air temps might be illuminating.

We agree that comparing 2 m and 10 m air temperature might be illuminating. However, we not have observations from 10 m height, and so feel this goes beyond the scope of this study as the focus is to be able to compare buoy observations and reanalysis directly.

Any change in how the 2 m air temps are calculated in ERA5 (still not an explicit model level, right?).

Computation of temperature at 2 m level is based on interpolation between the lowest model level and the surface making use of the same profile functions as in the parametrization of the surface fluxes. Therefore there was no change in the computation of 2 m air temperature in ERA5 and ERAI. However, the lowest model levels are different in ERA5 and ERAI due to the higher vertical resolution in ERA5.

Confused why you need to interpolate the ERAI data to the ERA5 grid before interpolating to the buoy position. Guessing this won't be a big issue as ERA5 is of intermediate resolution but still seems odd to me. How do you deal with the temporal differences between ERAI and ERA5? ERA5 is hourly and ERA-I is 6-hourly?

The original text is indeed unclear, thank you for pointing this out. We do NOT interpolate ERAI to ERA5 for buoy comparison. To clarify, we have rewritten the text in the manuscript as follows: "For comparison and evaluation against buoy observations, ERA5 is bilinearly interpolated to the buoy positions, and ERA-I is first linearly interpolated to hourly data, and

then bilinearly interpolated to the buoy positions. For the comparison between ERA-I and ERA5 over the Arctic sea ice, the ERA-I data are first bilinearly interpolated to the grid of ERA5, and then T2M is averaged in the season, and total precipitation and snowfall is integrated over the season to calculate the seasonal mean."

I'm confused why you don't show the actual ice thickness for the buoys (I think you just show the estimated ice thickness change from the FDD model?). Also confused as to whether you initialize the FDD model with zero ice thickness or not, as you show ice growth, not ice thickness. Any reason for this? Again, I see little value in this analysis so suggest dropping this and improving the rest of the analysis presented in the paper.

The new Figures 8 and 9 show the snow and ice thickness from the buoy observations and from the 1-D model runs (this is indicated in the legend), NOT for the FDD model. FDD is often used to estimate ice thermodynamic growth from zero, so ice thickness is from zero in the FDD model. For clarity, we have rewritten the FDD model part and made it much shorter as you suggest (Please refer to our response to your General Comments above).

Specific comments

P1 L3-4: 'The decline of Arctic sea ice has been attributed to various interrelated causes, including a general overall warming trend (Steel et al., 2008; Polyakov et al., 2010).' Seems pretty vague so would recommend you either improve this or drop it.

This sentence was deleted.

P1 L10: I would replace 'in-situ atmospheric observations' with something like 'direct observations of the atmosphere, sea ice and ocean conditions'?

Replaced with "direct observations of the atmosphere, sea ice and ocean conditions"

P1 L15: I suggest you combine this line 'Atmospheric reanalyses etc..' with the one about their use earlier on L12-13.

This sentence was moved to directly follow the earlier sentence about reanalyses other uses, and rewritten as follows: "In addition, reanalyses are also frequently used to force snow and sea ice models (Schweiger et al., 2011; Merkouriadi et al., 2017; Stroeve et al., 2018).".

P1 L22: I think you need to make the point here that the 1950 onwards data isn't yet available yet? Unless you're guessing it will be at the time of publication. . .

We now clarify the data availability as, "The entire ERA5 dataset, extending back to 1950, will be available for use in late 2019." on P4 L5-6 in the revised manuscript.

I also think you need to provide a better discussion of these supposed improvements and how they might increase ERA5's utility for Arctic studies, e.g.: - What do you mean by improved representation of troposphere and global balance? - More consistent? How?

We are limited in what we can say here, because there is still no official peer-reviewed publications documenting the ERA5 product, and neither many validation studies of ERA5 as far as we know, that is why this study is of value. This text has been revised as follows:

"There are several major improvements in ERA5 compared with ERA-I, including much higher spatial and temporal resolutions and more consistent sea surface temperature and sea ice concentration (Hersbach and Dee, 2016). Evaluations of the performance of ERA5 have been conducted over the land and revealed a higher performance of ERA5 than ERA-I (Albergel et al., 2018; Urraca et al., 2018), and other commonly used reanalysis, such as, MERRA-2 (the second version of the Modern-Era Retrospective Analysis for Research and Applications) (Olausen, 2018; Urraca et al., 2018). However, the performance of ERA5 over Arctic sea ice is yet to be fully investigated."

P3 L20: Where were these buoys deployed?

This sentence has been revised to provide this information: "For additional coverage, we also use observations from 3 snow buoys deployed in 2015, two of which in the Laptev Sea and one in the Central Arctic (Table 1; Fig. 1).". Please also refer to P3 L30-31 and P4 L1.

P4 L23: Boisvert not Biosvert P6 L3-4: Biosvert is corrected to Boisvert.

And any thoughts on how the cloud physics might have changed in ERA5 to cause this big change in snowfall/precip ratio?

We have added the following text in to the revised manuscript: "In ERA-I, the split between liquid and ice in clouds is determined diagnostically as a function of temperature from −23 to 0 °C, with ice-only below −23 °C and liquid-only above 0 °C. In contrast, the IFS Cy41r2 used in ERA5 includes a  prognostic microphysics scheme, with separate cloud liquid, cloud ice, rain and snow prognostic variables (Sotiropoulou et al., 2015; see also ECMWF IFS documentation –Cy41r2; https://www.ecmwf.int/sites/default/files/elibrary/2016/16648-part-iv-physical-processes.pdf). Our findings indicate that ERA5 has significantly less Arctic rainfall than ERA-I, particularly in August-September (Fig. 7, Figs. S4-5)." (Please also refer to P7 L25-29 in the revised manuscript).

I think you should plot this ratio out as a separate figure as it seems like a crucial part of the story.

We feel that this moves beyond the scope of this study. The Snowfall/Total precipitation ratio can clearly be seen by comparing the cumulative snowfall /total precipitation plots in Figure 7 of the revised manuscript, and so does not necessitate an additional figure.

P6 L4-5: I'm not quite sure how Figure 5 shows it is less anomalous as these are just showing the reanalysis data not compared to anything. Think you need to plot the buoy results too despite the big issues of representation etc.

We now plot the buoy snow depth as snow water equivalent in Figure 7 (formerly figure 5). Further buoys are shown in Figure S4 and S5 in the supplementary information. The figures are shown below. We have removed the word "anomalous" from the manuscript.

[Figure]

**Figure 7. Cumulative total precipitation (prec.) and snowfall (snow) for ERA5 and ERA-I and snow depth for buoys (a) 2012D, (b) 2013E, (c) 2011M, (d) 2012H, and (e) 2012L. Accumulation starts from 15 August for panels (a) and (b), and from 1 October for panels (c)-(e). Note there was no snow depth data for buoy 2013E during the accumulation period.**

[Figure]

**Figure S4. Cumulative total precipitation (prec.) and snowfall (snow) for ERA5 and ERA-I and snow depth for buoys (a) 2010A, (b) 2012C, (c) 2013B, (d) 2014E, and (e) 2015D. Accumulation starts from 15 August. Note that Buoy_2012C does not have snow depth.**

[Figure]

**Figure S5. Same as Figure S4, but for buoys (a) 2012I, (b) 2012J, (c) s16, (d) s20, and (e) s29, and accumulation starts on 1 October.**

P6 L10: why not use a daily climatology of density? You cite Warren1999 for the 350 value but this seems overly simplistic considering the results presented in Warren1999.

Thanks for the suggestion. The constant snow density was replaced with climatological monthly snow densities based on Warren et al. (1999). The results are shown in the new Figure 7, Figure S4 and S5, and Table 1.

P8, L4-5: can you briefly describe what this ocean heat flux is? E.g. 2 W/m2?

We have added description for the ocean heat flux in the manuscript on P10 L17-19 as "For all of the simulations we apply a seasonally variant ocean heat flux according to McPhee et al. (2003), which is large in October (10-20 $Wm^{-2}$), and decreases to nearly zero from mid-November." Accordingly, subplots for the used ocean heat flux were provided in the supplementary information Figure S7.

P9, L30: drop the warm summer bias comment here as you repeat it later.

This section now reads: "Overall, we find a warm bias in ERA-I and ERA5, when compared with the buoys. In both reanalyses, these biases are smallest in summer months, and larger during the autumn, winter and spring. The warm bias in ERA5 is smaller than ERA-I during the summer months. However, we find a larger warm bias in ERA5 than in ERA-I during the cold season, especially when the observed T2M was lower than -25 °C in the Atlantic sector and the Pacific sector."

P10, L1-3: think you should mention the caveat here that the buoy probably isn't giving the 2 m air temperatures.

We now include the sentence: "The near surface warm bias in ERA5 and ERA-I may also partly be attributed to the difference in height with observations."

Can we be sure resolving the boundary layer is the actual problem here?

We cannot be sure, as we do not analyze vertical profiles here. However, several studies have highlighted this problem in a range of reanalysis products, and many studies indicate this is the cause, for example (Beesley et al., 2000; Tjernstöm and Graversen, 2009; Graham et al., 2017b; Kayser et al., 2017)

P10, L5-14: I think this needs a bit of rewording for clarity. Really worth stressing that the total precip is lower but the snowfall is higher, right?! I would start with that difference then explain what it means in terms of the comparisons with the buoys. - Think you also need to make the point later regarding which precip was used to force the 1D model and that care needs to be taken regarding how precip is used in the products perhaps.

We have rewritten this section to clarify. The snowfall product in ERA5 is only larger than ERA-I during summer months. The text now reads: "The total precipitation over Arctic sea ice in ERA5 was lower than in ERA-I in all seasons. This is surprising, as ERA-I is known to be drier in the Arctic compared with some other recent reanalyses (Lindsay et al., 2014; Merkouriadi et al., 2017; Boisvert et al., 2018). However, the snowfall is higher in ERA5 than in ERA-I during the summer months. This indicates that ERA5 has a higher snowfall to precipitation ratio than ERA-I during summer. ERA-I is known to have an anomalously large fraction of liquid precipitation and low snowfall to precipitation ratio in the Arctic, especially during August-September (Dutra et al., 2011; Leeuw et al., 2015). The total precipitation accumulated along the buoys drift trajectories, during the cold season (from 15 August/1

October until buoy failed or 30 April), was lower in ERA5 than in ERA-I for every buoy. Similarly, the accumulated snowfall in ERA5 is lower than in ERA-I for all buoys with an accumulation date starting from 1 October. In contrast, the total accumulated snowfall in ERA5 is higher than in ERA-I for buoys with an accumulation date starting from 15 August, due to likely anomalous summer/autumn rainfall in ERA-I being classified as snow in ERA5. The accumulated total precipitation and/or snowfall in ERA5 are often closer to the SWE content of buoy measured snow pack, compared with ERA-I. Nonetheless, the lack of representative in-situ observations and difficulty in measuring snow accumulation on sea ice in the Arctic makes it a challenge to accurately evaluate precipitation products over sea ice (e.g. Rasmussen et al., 2012; Lindsay et al., 2014; Sato et al., 2017; Blanchard-Wrigglesworth et al., 2018; Boisvert et al., 2018). Given snow is such a critical factor in sea ice evolution, more representative observations are therefore needed (e.g. Merkouriadi et al., 2017; Webster et al., 2018)." (Please refer to P12 L22-31, P13 L1-8 in the revised manuscript)

Any particular recommendations here? I.e. do you think we should be using the snowfall product or deriving this from the total precip? There are other ways of doing this also (using higher level temps).

We do not make any specific recommendations here, because we do not have sufficient data to validate the reanalyses products. We simply highlight that the choice of snowfall / precipitation products will affect the simulation of snow and sea ice in a model. We conclude that further precipitation and snow pack measurements over Arctic sea ice are essential.

Figure 2: why does the green line seem dashed? Can you move the difference line lower so it's easier to see? - I also think you should show not the difference between ERA5 and ERA-I but two lines representing the differences between the reanalyses and the buoys. Maybe just pick a couple as good examples of the seasonal cycles you mention in the text and make these bigger/clearer, then put the rest in the supplementary information? As it is, it's hard to really get a sense of what these figures show quantitatively.

Thank you for the suggestions. We replotted Figures 2 and 3.

In the new Figure 3, we show the variation of 2 m air temperature in ERA5, ERA-I and the buoys. The differences of T2M between ERA5 and ERA-I are shown in a separate subplot below.

The differences between ERA5/ERA-I and the buoy measurements are snow shown in a separate new Figure (Figure 4).

We have reduced the number of buoys shown in the manuscript to five examples. All the other buoys are shown in supplementary Figure S1 and S2.

Figures 3-4 and Figure S1-S3 are shown below.

[Figure]

Figure 3. Variation of 2 m air temperature (T2M) in ERA5, ERA-I and the buoys and the differences of T2M between ERA5 and ERA-I for buoys (a) 2012D, (b) 2013E, (c) 2011M, (d) 2012H, and (e) 2012L. Note the different time-axis.

[Figure]

**Figure S1. Variation of 2 m air temperature (T2M) in ERA5, ERA-I and the buoys and the difference of T2M between ERA5 and ERA-I for buoys (a) 2010A, (b) 2012C, (c) 2013B, (d) 2014E, and (e) 2015D.**

[Figure]

**Figure S2. Same as Figure S1, but for buoys (a) 2012I, (b) 2012J, (c) s16, (d) s20, and (e) s29. Note there is no buoy data in Figure (b) for buoy 2012J.**

[Figure]

**Figure 4. Variation of T2M differences between ERA5/ERA-I and buoys for (a) buoy 2012D, (b) buoy 2013E, (d) buoy 2011M, (d) buoy 2012H, and (e) buoy 2012L.**

[Figure]

**Figure S3. Variation of T2M differences between ERA5/ERA-I and buoys for buoys (a) 2010A, (b) 2012C, (c) 2012I, (d) 2013B, (e) 2014E, (f) 2015D, (g) s16, (h) s20, and (i) s29.**

Figure 4: this is a good figure!

Thank you.

Figure 5: you don't need the second y-axis, just state in the legend that the dashed lines are snow. You should add the units to the label and legend. Why does this not include the buoys change in snow depth? As a second y-axis?! Or converted to cumulative precip and plotted on the same axis.

We have replotted the Figure 5 (now Figure 7) as mentioned in our response to your General comments. We now show the snow depth as snow water equivalent from buoys, as suggested. Please refer to our reply in the General comments part.

Figure 7: I don't think you need to show the FDD values as they don't mean much physically.

As mentioned in our response to your General comments, we made the FDD part more concise, and moved the figure to supplementary information (Figure S6).

Figure 8: why the weird staircase in buoy 2011M? Lower temporal resolution for some reason?

The staircase pattern is an artifact of the automated system that is being applied to archive the data. If the acoustic sounders fail and the temperature string is still working, the positions of the ice surface and bottom are determined from the temperature string (with much less accuracy than from the acoustic sounder). We have added one sentence to explain how the ice thicknesses were determined for the buoys on P3, L22-23 in section 2.1, and pointed out the reason for the staircase ice thickness for buoy 2011M and 2012J on P10, L31-33.

---

## Author Comment (AC3) · 1 Mar 2019

**We would like to thank the reviewer for the evaluation of our study and the constructive comments that helped us to improve the manuscript. Please find below the reviewer's comments in black font and the author's response in blue font.**

5                                          **Responses to Anonymous Referee #1**

- In the revised manuscript, we added one new subsection 3.1 to show the seasonal mean difference of T2M, total precipitation and snowfall over the pan-Arctic sea ice and added one new figure (Figure 2). Accordingly, the original section 3.1 changed to 3.2, and the original subsection 3.1

10       and 3.2 changed to 3.2.1 and 3.2.2, respectively.
- We replotted the original Figure 2 and 3 which named as Figure 3 and 4 in the revised manuscript,
  - In the new Figure 3 we show the variation of T2M from ERA5, ERA-I and buoys and the differences of T2M between ERA5 and ERA-I for 5 buoys, plots for the other buoys are

15          provided in the Supplementary Information as Figure S1 and S2.
  - In the new Figure 4 we show the variation of T2M differences between ERA5/ERA-I and buoys for the same 5 buoys as in the new Figure 2. Plots for the other buoys are provided in the Supplementary Information as Figure S3.
- We added a new Figure 6 to show the T2M difference between ERA5/ERA-I and buoys in four

20       regions (Central Arctic, Atlantic sector, Pacific sector, and Laptev Sea) in the Arctic. Correspondingly, we added one paragraph at the end of the original section of 3.1 to describe the regional differences.
- We replotted the original Figures 5 and 6, the new figure is now Figure 7 and Figure S4-5. In these new figures, we show the accumulated total precipitation and snowfall from ERA5 and

25       ERA-I and snow depth measured by buoys.
- The original Figure 7 (showing the FDD and sea ice growth) was moved to Supplementary Information as Figure S6, and the discussion on the FDD model was shortened.
- Forcing data of wind speed (V), relative humidity (Rh) and total cloud (CN) and ocean heat flux were plotted and provided in the Supplementary Information as Figure S7.

30   Our specific responses are given below.

In the abstract, I would not say that ERA-I is drier than 'most' reanalyses, I will say ERA-I is drier than 'some' reanalyses - see Boisvert et al., 2018 Journal of Climate

Agree. Changed to "ERA-I is drier than some modern reanalyses".

Figure 3 caption. Do you mean panel (D), not (K)? Because there is no panel K in the figure.

Thank you for spotting this typo. In the revised manuscript, we have replotted Figure 3. We now show only five buoys, and moved the remaining buoys to Figures S1-2 in the Supplementary Information (see below) to keep the main text concise.

It would be great to see a little more conversation dealing with the differences in Temp and Precipitation compared to the buoys and to themselves. It seemed like some regions where the buoys were/times of the year produce larger differences between the buoys and the reanalyses. For example, there appeared to be larger differences between realanyses and the buoys in the Beaufort sea areas.

We now include a new Figure (Fig. 2), showing the regional and seasonal differences of Temp, total precipitation and snowfall over the pan-Arctic sea ice between ERA5 and ERA-I. These analyses are discussed in the new subsection 3.1. In addition, we have made a new figure (Figure 6), showing the Temp differences between buoys and the reanalyses for different regions, including: the Central Arctic (north of 86° N), Pacific sector (90° W – 150°E), Atlantic sector (30° W – 60° E) and Laptev Sea (60° E – 150° E). This reveals that the differences are large in the Atlantic sector and small in the Central Arctic. New Figure 2 and 6 are shown below. The subsection 3.1 reads as,

**"3.1 Spatial distribution of seasonal difference of reanalysis near surface temperature and precipitation**

Figure 2 shows the seasonal mean differences of T2M, total precipitation and snowfall between ERA5 and ERA-I over Arctic sea ice during 2010-2015. We classify spring as March, April and May, summer as June, July and August, autumn as September, October and November, and winter as December, January and February. The seasonal mean ice extent is obtained from the monthly sea ice concentration from NOAA/NSIDC during 2010-2015 (Meier et al., 2017).

The difference in T2M between ERA5 and ERA-I clearly varies with season (Fig. 2a-d). ERA5 is generally warner than ERA-I in spring and winter, and colder than ERA-I during summer and autumn over most regions of Arctic sea ice. These temperature differences are small during summer, but large during the other seasons. Near the North Pole, ERA5 is warmer than ERA-I in summer, but colder than ERA-I in winter. Whether warmer or colder, the differences between ERA5 and ERA-I are small (<±0.4 °C) in this region.

ERA-I is known to be a relatively "dry" global reanalysis product in the Arctic compared with most other modern reanalyses (e.g. MERRA-2, CFSR, and JRA-55) (Lindsay et al., 2014; Merkouriadi et al., 2017; Boisvert et al., 2018). However, the total precipitation in ERA5 is lower than in EAR-I over Arctic sea ice in all seasons (Fig. 2e-h). The lower precipitation in ERA5 is most pronounced in summer, and in the eastern Arctic. Differences in the snowfall between ERA5 and ERA-I are smaller

than for total precipitation (Fig. 2 i-j vs. Fig. 2e-h). The snowfall in ERA5 is lower than in ERA-I in spring, autumn and winter, but larger than ERA-I in summer."

[Figure]

**Figure 2. Seasonal mean difference between ERA5 and ERA-I (ERA5-ERA-I) for T2M (a-d), total precipitation (e-h), and snowfall (i-l) in spring (a, e, i), summer (b, f, j), autumn (c, g, k) and winter (d, h, l) over Arctic sea ice during 2010-2015. The mean sea ice extent in the seasonal is used for classification of sea ice and open ocean.**

[Figure]

**Figure 6. Scatter plot of T2M from ERA5 and ERA-I vs. from buoys in (a) Central Arctic, (b) Atlantic sector, (c) Pacific sector, and (d) Laptev Sea, and number of data (daily) in (e) Central Arctic, (f) Atlantic sector, (g) Pacific sector, and (h) Laptev Sea.**

The text for describing Figure 6 read as follows "The performance of reanalysis near surface temperature varies with region over Arctic sea ice (Fig. 6, also refer to Fig. 2). According to the buoys' positions (Fig. 1), we define four regions in the Arctic: the Central Arctic (north of 86 °N), and the Pacific sector (90 °W – 150 °E), the Atlantic sector (30 °W – 60 °E), and the Laptev Sea (60 °E – 150 °E). The later three sectors are south of 86 °N. The ERA5/ERA-I near surface temperature performs best in the Central Arctic (Fig. 6a), and well in the Pacific sector (Fig. 6c). It performs well in the Atlantic sector when the T2M is above -25 °C, but poorly when the T2M is below -25 °C (Fig. 6b). The performance of reanalysis near surface temperature in the Laptev Sea needs to be further investigated due to small number of observations in this region (Fig. 6d & 6h). However, there is also some seasonal bias in the availability of data from buoys in the different regions, largely due to when buoys are deployed in different regions of the Arctic and ice drift patterns." Please also refer to P6 L25-33.

Figures 2 and 3. It would be beneficial to also have the differences between ERA5 and ERA-I and the buoy temperatures perhaps in a different figure? Because it is a little hard to see how well the reanalyses compare with the buoys the way it is now. Or perhaps provide a table with the differences and biases for each buoy.

To show the differences clearly, we replotted Figures 2 and 3 (see below). These are now Figures 3 and 4 in the revised manuscript.

In the new Figure 3, we show the variation of 2 m air temperature in ERA5, ERA-I and the buoys in a subplot, and the differences of T2M between ERA5 and ERA-I in a separate subplot below. Overall, there are five buoys shown in the new Figure 3. All the other buoys were shown as supplementary Figure S1 and S2.

In the new Figure 4, we show the differences between ERA5/ERA-I and the buoy measurements for five buoys as in Figure 3. The other buoys are shown in supplementary Figure S3.

[Figure]

**Figure 3. Variation of 2 m air temperature (T2M) in ERA5, ERA-I and the buoys and the difference of T2M between ERA5 and ERA-I for buoys (a) 2012D, (b) 2013E, (c) 2011M, (d) 2012H, and (e) 2012L. Note the different time-axes.**

[Figure]

**Figure S1.** Variation of 2 m air temperature (T2M) in ERA5, ERA-I and the buoys and the difference of T2M between ERA5 and ERA-I for buoys (a) 2010A, (b) 2012C, (c) 2013B, (d) 2014E, and (e) 2015D.

[Figure]

**Figure S2. Same as Figure S1, but for buoys (a) 2012I, (b) 2012J, (c) s16, (d) s20, and (e) s29. Note there is no buoy data in Figure (b) for buoy 2012J.**

[Figure]

**Figure 4. Variation of T2M differences between ERA5/ERA-I and buoys for (a) buoy 2012D, (b) buoy 2013E, (c) buoy 2011M, (d) buoy 2012H, and (e) buoy 2012L.**

[Figure]

**Figure S3. Variation of T2M differences between ERA5/ERA-I and buoys for (a) buoy 2010A, (b) buoy 2012C, (c) buoy 2012I, (d) buoy 2013B, (e) buoy 2014E, (f) buoy 2015D, (g) buoy s16, (h) buoy s20, and (i) buoy s29.**

Page 4, line 16: Might be best to say where these 2 buoys are located in the text. 2013 E and 2012 J? Perhaps the reanalyses are better at producing accurate temperatures in certain regions of the Arctic and perhaps this could be elaborated on more.

Text of "which are both deployed in central Arctic, the former near the North Pole and the later closer to the Laptev Sea (Fig. 1)" was added to clarify where the 2 buoys were deployed (See P5 L22-23). To elaborate, we added a new figure (Figure 6) as mentioned above for T2M difference between buoys and the reanalyses in different regions: Central Arctic (north of 86° N), and south of 86° N we have the Pacific sector (90° W – 150°E), Atlantic sector (30° W – 60° E) and Laptev Sea (60° E – 150° E). Our new Figure 6 shows the reanalysis are best at producing accurate temperature in the Central Arctic.

I know that snow depths are fairly uncertain, but perhaps instead of taking a constant snow density of 350 kg/m3, why not time vary it throughout the winter season and based on locations based on the Warren climatology. This might improve your results.

Thanks for your suggestion. Instead of a constant snow density, a climatological monthly mean snow density was applied based on Fig. 11 of Warren et al. (1999). This results in the new Figure 7 and Figures S4 and S5, in which we show the precipitation/snowfall from reanalysis of ERA5 and ERA-I, and snow depth from buoys (see below).

[Figure]

**Figure 7. Cumulative total precipitation (prec.) and snowfall (snow) for ERA5 and ERA-I and snow depth for buoys (a) 2012D, (b) 2013E, (c) 2011M, (d) 2012H, and (e) 2012L. Accumulation starts from 15 August for panels (a) and (b) and from 1 October for panels (c)-(e). Note there was no snow depth data for buoy 2013E during the accumulation period.**

[Figure]

**Figure S4. Cumulative total precipitation (prec.) and snowfall (snow) for ERA5 and ERA-I and snow depth from for buoys (a) 2010A, (b) 2012C, (c) 2013B, (d) 2014E, and (e) 2015D. Accumulation starts from 15 August. Note that Buoy_2012C does not have snow depth.**

[Figure]

**Figure S5. Same as Figure S4, but for buoys (a) 2012I, (b) 2012J, (c) s16, (d) s20, and (e) s29, and accumulation starts on 1 October.**

Line 18 page 6 should be ERA5

It was corrected to ERA5.

[revised manuscript text omitted]

**3.1 Spatial distribution of seasonal difference of reanalysis near surface temperature and precipitation**

Figure 2 shows the seasonal mean differences of T2M, total precipitation and snowfall between ERA5 and ERA-I over Arctic sea ice during 2010-2015. We classify spring as March, April and May, summer as June, July and August, autumn as September, October and November, and winter as December, January and February. The seasonal mean ice extent is obtained from the monthly sea ice concentration from NOAA/NSIDC during 2010-2015 (Meier et al., 2017).

The difference in T2M between ERA5 and ERA-I clearly varies with season (Fig. 2a-d). ERA5 is generally warner than ERA-I in spring and winter, and colder than ERA-I during summer and autumn over most regions of Arctic sea ice. These temperature differences are small during summer, but large during the other seasons. Near the North Pole, ERA5 is warmer than ERA-I in summer, but colder than ERA-I in winter. Whether warmer or colder, the differences between ERA5 and ERA-I are small ($<\pm0.4$ °C) in this region.

ERA-I is known to be a relatively "dry" global reanalysis product in the Arctic compared with most other modern reanalyses (e.g. MERRA-2, CFSR, and JRA-55) (Lindsay et al., 2014; Merkouriadi et al., 2017; Boisvert et al., 2018). However, the total precipitation in ERA5 is lower than in EAR-I over Arctic sea ice in all seasons (Fig. 2e-h). The lower precipitation in ERA5 is most pronounced in summer, and in the eastern Arctic. Differences in the snowfall between ERA5 and ERA-I are smaller than for total precipitation (Fig. 2 i-j vs. Fig. 2e-h). The snowfall in ERA5 is lower than in ERA-I in spring, autumn and winter, but larger than ERA-I in summer.

[revised manuscript text omitted]
. 5e7a) to 38.4 mm water equivalent (buoy 2011M; Fig. 6a7c). This is interestingindicates that ERA5 is drier than ERA-I, which is because ERA-I is known to be a relatively "dry" global reanalysis product in the Arctic compared with most other modern reanalyses (e.g. MERRA-2, CFSR, and JRA-55) (Lindsay et al., 2014; Merkouriadi et al., 2017; Boisvert et al., 2018).

Unlike the accumulated total precipitation, the accumulated snowfall (SfSF) in ERA5 is sometimes can be larger than that in ERA-I (Figs. 5-7 & 6S4-5; Table 1). Specifically, Ffor buoys deployed near the North Pole, which that started operating on 15 August, the accumulated Sf 
[revised manuscript text omitted]

[Figure]

[Figure]

**Figure 4. Variation of T2M differences between ERA5/ERA-I and buoys for (a) buoy 2012D, (b) buoy 2013E, (d) buoy 2011M, (d) buoy 2012H, and (e) buoy 2012L.**

[Figure]

**Fig. 45. Statistics of T2M from ERA5, ERA-I and all the buoys. (a) Scatter plot for all data (small dots) and average T2M at 5 degree bins between -45 °C and +5 °C, (b) Daily temperature differences between the reanalysis and between the reanalysis and the buoys corresponding to 5 degree bins between -45 °C and +5 °C, and (c) monthly mean differences and standard deviation (std). In panel a, the black solid line is for 1:1.**

[Figure]

**Figure 6. Scatter plot of T2M from ERA5 and ERA-I vs. from buoys for (a) Central Arctic, (b) Atlantic sector, (c) Pacific sector, and (d) Laptev Sea, and number of buoy data (daily) per month for (e) Central Arctic (f) Atlantic sector, (g) Pacific sector, and (h) Laptev sea. The definition of sectors are shown in Figure 1.**

[Figure]

[Figure]

**Figure 7. Cumulative total precipitation (prec.) and snowfall (snow) for ERA5 and ERA-I and snow depth for buoys (a) 2012D, (b) 2013E, (c) 2011M, (d) 2012H, and (e) 2012L. Accumulation starts from 15 August for panels (a) and (b), and. **

precipitation, and dashed lines for the cumulative snowfall from 1 October for panels (c)-(e). Note there was no snow depth data for buoy 2013E during the accumulation period.

[Figure]

Figure 6. Same as Figure 5, but with accumulation starting from 1 October. The dashed blue lines overlap with the solid blue lines in the panels due to the small differences between the cumulative total precipitation and snowfall in ERA5.

[Figure]

Fig. 7. FDD and estimated ice growth using cumulative FDD based on equation 1 along the trajectories of (a) Buoy 2011M, (b) Buoy 2012H, (c) Buoy 2012L, and (d) Buoy 2012J for freeze-up on 1 Oct.

[Figure]

[Figure]

**Fig. 8.** Evolution of snow and sea ice  thickness during freezing season based on simulations with HIGHTSI for (a) uoy 2012H, (b)  buoy 2011M, (c)  buoy 2012L, and (d)  buoy 2012J in the runs of TPI_T2MI, TP5_T2M5, FI_T2MI, and SF5_T2M5.

[Figure]

[Figure]

**Fig. 9.** Same as Fig. 8, but for the runs of TPI_T2M5, and TP5_T2MI.

---

## Author Response (AR2)

Dear Dr. Derksen,

We would like to thank the two reviewers for their constructive comments. Indeed while double and triple checking our calculations for precipitation in ERA5 we noticed that an error had slipped in our calculations. As both reviewers noted, the increase in precipation and snowfall are consistent in ERA5 relative to ERA-I. This change is now reflected in the revised manuscript. And we have addressed the reviewers concerns and suggestions.

Given the comments from Alek Petty, we also added a new figure (new Fig. 8) to show the ratio of snowfall to total precipitation, which has changed substantially (i.e. increased) in the new ERA5. Given that the readership will also have use of absolute amounts of precipitation and snowfall (and not only differences or ratios between ERA5 and ERA-I shown in Fig. 2 and 7), we added new figures to the Supplementary Material (S1 and S2), as we consider that these are also valuable to show (and we refer to this in the revised text. The new results are reflected in a rewritten abstract, section 3.1-3.2, and Conclusions.

We also read through the manuscript again thoroughly to edit the text for clarity and fluency.

Overall, we find that with the insights of the two reviewers, this paper has now become much improved, and valuable contribution for the community. It will hopefully spur some more in-depth studies on the exact causes of the changes from ERA-I to ERA5, which are beyond the scope of the current study.

Detailed responses to the reviews are given below.

We hope you find this work publishable in The Cryosphere.

Yours truly,

Caixin Wang

on behalf all co-authors

**Reply to reviewer Alex Petty**

Thanks for your insightful comments. Our reply to your comments are written with blue text. In brief, we identified an error in our ERA5 precipitation calculations, and corrected the corresponding figures and updated the text to reflect these changes. We also added the figure on the snowfall to precipitation ratio, thank you.

Comments from Alek Petty:

**Suggestions for revision or reasons for rejection (will be published if the paper is accepted for final publication)**

Most importantly, I recently had a look at the ERA5 snowfall and my brief analysis suggested it's pretty consistently higher than ERA-I snowfall over the Arctic in all seasons, which isn't what you find in this paper. I think it's worth you (and me!) taking a closer look at your ERAI and ERA5 snowfall results. Perhaps the time period matters here, but there was some difference with the data from ERAI to ERA5 (summing hourly ERA5 values within a day instead of 6-hourly ERA-I data). This needs to be resolved and could have a significant impact on your results that would need correcting if I am right.

Thanks for pointing this out!! (as did the 2nd reviewer). You are correct. We double checked our analysis, and found errors in our calculation of the ERA5 total precipitation and snowfall accumulation. Now we corrected our results, and included them in the updated manuscript. In the revised manuscript, we show that the total precipitation and snowfall in ERA5 are larger than those in ERA-I in all seasons over Arctic sea ice and along the buoy drift trajectories. Following this, we replaced the old Fig. 2 (e-h) and (i-l), Fig. 7 and Fig. S4-5 with updated figures, and rewrote the corresponding text in section 3.1, section 3.2.2. Accordingly, the old Figs. 8-9 for model runs were replotted, and renamed as Fig. 9 and 10, respectively. The main results concerning our model runs have no big change.

We have also updated the abstract and conclusions to reflect these changes, in addition to the section discussion snowfall and precipitation.

Other comments:

I still think you need to make clearer the changing ratio of snowfall to precip as it seems a big part of the story and not beyond the scope of this work at all.

We now agree this aspect is indeed valuable to show. For this, we added ratio values in the new Fig. 7 and Figs. S6-7, and one new Fig. 8 for its ratio in four seasons, and also added some discussion on this matter in text (refer to P7) (including abstract and conclusions).

New Figure 2: Drop the colorbars in the first three columns as they are the same as the colorbar on the right. Add the variable (i.e. 2m air temperature (C)). Unclear what you've done with the ice extent as I can't make it out on the maps.

Figure 2 was re-made. The new Figure 2 only shows the colorbars on the right-hand side. In the figure, the ice extent is used as a mask to distinguish ocean and sea ice.

New Figures 3 and 4: thanks for making this change, I think it's an improvement but I still think it could be better! It's tough to jump between the figures and interpret the differences, and again I think the comparison to the buoys is more important so I would suggest merging these figures and just showing one or to buoy profiles instead in each figure. Can you place the Figure 4 buoy comparison panel between the raw and difference plots in Figure 3? It would also be nice to have RMSE values included here too for the buoy comparisons to ERAI and ERA5.

Thank you for this suggestion. Figures 3 and 4 were merged. The new Figure 3 and 4 only both show two buoys each, with all panels for one buoy in one of the figures. Additional buoys are shown in the Supplementary Material (Fig. S6-7). Each buoy consists of three panels, in the upper panel for the variation of ERA5, ERA-I and buoys, the mid-panel for the difference between ERA5 and ERA-I, and the lower panel for the ERA5/ERA-I and buoys. In addition, the RMSE values are included in the new Figure 3 and 4.

New Figure 6: nice!

Thanks!

New Figure 7: looks better! I would suggest dropping the second y-axis as this has the same units and just stating that in the left y labels (snow water equivalent (mm)) and make clear that you have cumulated reanalysis values and measured buoy snow depth values shown.

The second y-axis was dropped.

**Reply to Anonymous reviewer**

Thanks for your insightful comments. Our detailed replies are written with blue text. In brief, we identified an error in our ERA5 precipitation calculations, and corrected the corresponding figures and updated the text to reflect these changes.

Comments from Anonymous Referee #1
**Suggestions for revision or reasons for rejection (will be published if the paper is accepted for final publication)**
Review of Wang et al., Round 2

Thanks to the authors for addressing my concerns in the previous round of edits. I feel like the changes made improve the paper, and that it should be suitable for publication after these changes are made.

Section 3: I find it a little odd that the ERA5 total precipitation and snowfall values are lower than ERA-I. I thought and saw some preliminary results that the ERA5 snowfall was biased high from other reanalyses and CloudSat and was more similar to MERRA2? I thought their overall total precipitation magnitudes didn't increase but the partition between rain and snow changed from ERA-I to ERA5 so the snowfall year round was much larger in ERA5 compared to ERA-I. Not just in the summer months as is shown in Figure 2. Also it would be beneficial to try to explain or hypothesize why there are differences between ERA-I and ERA5 fields. Was cloud microphysics changed? Representation of the sea ice? Etc.

Thanks for pointing this out (as did the other reviewer). We double and triple checked our calculations for ERA5 and found a mistake in the integration of the ERA5 precipitation/snowfall in our calculations. You are right, the total precipitation and snowfall was larger in ERA5 than in ERA-I (see updated Figure 2). We corrected our results, and included them in the new manuscript. In the revised manuscript, we show that the total precipitation and snowfall in ERA5 are larger than those in ERA-I in all seasons over the Arctic and along all the buoy drift trajectories. Following this, we replaced the old Fig. 2 (e-h) and (i-l), Fig. 7 and Fig. S4-5 with new figures, and rewrote the corresponding text in section 3.1 and section 3.2.2. Accordingly, the old Figs. 8-9 for model runs were replotted, and renamed as Fig. 9 and 10, respectively. The main results concerning our model runs have no big change.

In response to the other reviewer, we also added a new figure (now figure 8), of the ratio of snowfall to total precipitation in ERA-I and ERA5 (refer to P7), to show the increase of snowfall at the expense of rain in ERA5 over Arctic sea ice, likely due to the improved cloud physics scheme.

In our manuscript, we had tried to explain or hypothesize why there are differences between ERA-I and ERA5 fields through talking about the different (improved) cloud physics scheme used in ERA5 (see section 3.2.2). The representation of sea ice in physics seems not have changed (see IFS Cy41r2 and Cy31r2). But there is more consistent sea ice concentration (SIC) product used in ERA5 due to using OSI-SAFr, the reprocessed version of the Ocean and Sea Ice Satellite Application Facilities (OSI-SAF) (Hersbach and Dee, 2016), which produces a large ice extent when the SIC is over 10% or 50% compared with previous version of OSI-SAF (ERA-I). Large sea ice concentration means less open water, possibly less water vapor. However, this is quite speculative, although could indeed also affect T2M.

After reading on I notice that you try to explain some of the differences in the cloud physics. But what about the sea ice representation? Has this changed?
There is more consistent sea ice concentration product used in ERA5 (Herbash and Dee, 2016). However, how the sea-ice-atmosphere interface is treated in ERA5 compared to ERA-I has not changed to our knowledge, and is simply for the boundary condition in the model. The full effect on the model sensitivity to this change has not as far we know been examined to any detail, and is beyond the scope of our work. Please also see our reply to your comments above. We have now indicated this in the text (refer to P4).

Any thoughts on why the temperature bias is larger for ERA5 than ERA-i?

It is difficult for us to answer this question at the moment, as there has not been a thorough comparison of the ERA-I and ERA5 model systems over sea ice as far as we know. In the reanalysis output we see this result, and an in-depth study is required by the model development team to answer this question.

Page 6 line 2: Could the warmer t2m in ERA5 be due to the representation of the sea ice cover and thickness in the model? This will likely affect that as well.
We know the sea ice concentration used in the ERA5 is different than in ERA-I, however, there are no detailed studies to show that this only would change the t2m in the Arctic. Given that largest deviations are in winter when ice concentrations are nearly 100% it would be unlikely to assume sea ice concentration would affect this. In any case, including this would be highly speculative on our part.

Page 7 line 11: Remove 'is' before 'indicates'.
"is" was removed

Page 8 Line 15: are the "some periods" where the ERA5/I snow depths different from the buoys seasonally or regionally dependent? Or is it just random? At the buoy locations where ERA5/I are always producing more snow regionally dependent?

From the analysis of the buoys in question we do not see any definite patterns in the differences by region. But we see a seasonality effect in the snowfall, as most of autumn snowfall in ERA5 was in fact rain in ERA-I, and as noted snowfall in ERA5 is especially higher in summer/autumn. We also note this in the text. As we note in the manuscript, we have mentioned that " snow drifting up against the buoy structure, snow erosion/sublimation around the buoy, or reflect anomalously high/low precipitation in the reanalyses may contribute to the difference between ERA5/I and buoy", among others, can cause dynamic changes of snow depth, not captured by reanalysis precipitation/snowfall fields

Page 11 Linen 18: "Sp-ERAI runs" should be changed.

[revised manuscript text omitted]

The supplementary Information includes 9 figures. Figure captions are:

Figure S1. Seasonal mean total precipitation (TP) for ERA-I (a-d) and ERA5 (e-h), and seasonal mean snowfall (SF) for ERA-I (i-l) and EAR5 (m-p) in spring (a, e, i, m), summer (b, f, j, n), autumn (c, g, k, o) and winter (d, h, l, p) over Arctic sea ice for 2010-2015.

Fig. S2. Annual total precipitation (TP) (a-c) and snowfall (SF) (d-f) for ERA5 (a, d), ERA-I (b, e) and their differences (ERA5 minus ERA-I) (c, f) over Arctic sea ice for 2010-2015.

Figure S3. Variation of 2 m air temperature (T2M) in ERA5, ERA-I and the buoys (upper panel) and the differences of T2M between ERA5 and ERA-I (mid-panel; green color) and comparisons for ERA5 and ERA-I with buoys (ERA5 minus buoy; ERA-I minus buoy) for buoys (a) 2010A, (b) 2012C, and (c) 2013B. RMSE values for the comparison between ERA products and buoys are shown as text, blue for ERA5-buoy, red for ERA-I-buoy. Note the different time-axis.

Figure S4. Same as Figure S3, but for buoys (a) 2014E, (b) 2015D, (c) 2012I, and (d) 2012L. Note the different time-axis.

Figure S5. Same as Figure S3, but for buoys (a) 2012J, (b) s20, (c) s16, and (d) s29.

Figure S6. Cumulative total precipitation (TP) and snowfall (SF) for ERA5 and ERA-I and snow depth for buoys (a) 2010A, (b) 2012C, (c) 2013B, (d) 2014E, and (e) 2015D. Accumulation starts from 15 August. The ratio of snowfall to total precipitation (SF/TP) in ERA5 (blue text) and ERA-I (red text) is also shown in the figure. Note that Buoy_2012C does not have snow depth data.

Figure S7. Same as Figure S6, but for buoys (a) 2012I, (b) 2012J, (c) 2012L, (d) s16, (e) s20, and (f) s29, and accumulation starts on 15 August for (a) and starts on 1 October for (b)-(f).

Figure S8. Cumulative FDD and estimated ice growth using cumulative FDD model along the trajectories of buoys (a) 2011M, (b) 2012H, (c) 2012L, and (d) 2012J for freeze-up from 1 October.

Figure S9.  Forcing data of wind speed (V), relatively humidity (Rh), total cloud (CN) and ocean heat flux (fw) used in the model runs.

[Figure]

**Figure S1. Seasonal mean total precipitation (TP) for ERA-I (a-d) and ERA5 (e-h), and seasonal mean snowfall (SF) for ERA-I (i-l) and ERA5 (m-p) in spring (a, e, i, m), summer (b, f, j, n), autumn (c, g, k, o) and winter (d, h, l, p) over Arctic sea ice for 2010-2015.**

[Figure]

**Figure S2. Annual total precipitation (TP) (a-c) and snowfall (SF) (d-f) for ERA5 (a, d), ERA-I (b, e) and their differences (ERA5 minus ERA-I) (c, f) over Arctic sea ice for 2010-2015.**

[Figure]

**Figure S3.** Variation of 2 m air temperature (T2M) in ERA5, ERA-I and the buoys (upper panel) and the differences of T2M between ERA5 and ERA-I (mid-panel; green color) and comparisons for ERA5 and ERA-I with buoys (ERA5 minus buoy; ERA-I minus buoy) for buoys (a) 2010A, (b) 2012C, and (c) 2013B. RMSE values for the comparison between ERA products and buoys are shown as text, blue for ERA5-buoy, red for ERA-I-buoy. Note the different time-axis.

[Figure]

**Figure S4. Same as Figure S3, but for buoys (a) 2014E, (b) 2015D, (c) 2012I, and (d) 2012L. Note the different time-axis.**

[Figure]

**Figure S5.** Same as Figure S3, but for buoys (a) 2012J, (b) s20, (c) s16, and (d) s29.

[Figure]

**Figure S6. Cumulative total precipitation (TP) and snowfall (SF) for ERA5 and ERA-I and snow depth for buoys (a) 2010A, (b) 2012C, (c) 2013B, (d) 2014E, and (e) 2015D. Accumulation starts from 15 August. The ratio of snowfall to total precipitation (SF/TP) in ERA5 (blue text) and ERA-I (red text) is also shown in the figure. Note that Buoy_2012C does not have snow depth data.**

[Figure]

**Figure S7.** Same as Figure S6, but for buoys (a) 2012I, (b) 2012J, (c) 2012L, (d) s16, (e) s20, and (f) s29, and accumulation starts on 15 August for (a) and starts on 1 October for (b)-(f).

[Figure]

**Figure S8. Cumulative FDD and estimated ice growth using cumulative FDD model along the trajectories of (a) buoy 2011M, (b) buoy 2012H, (c) buoy 2012L, and (d) buoy 2012J for freeze-up from 1 October.**

[Figure]

**Figure S9.** **Forcing data of wind speed (V), relatively humidity (Rh), total cloud (CN) and ocean heat flux (fw) used in the model runs.**

---

## Author Response (AR4)

Dear Dr. Derksen,

Thanks for your comments.

We have changed/removed the text as you suggested. Our detailed response is shown as blue fonts in the following.

Comments to the Author:
Thank-you for your revised manuscript. The reviewers have accessed your updated results, and the paper is now suitable for publication in the The Cryosphere. I have noted some minor corrections below. Thanks for your contribution to The Cryosphere.
Chris Derksen

(Note that page and line references are from the tracked changes version of the revised manuscript)
Page 4 line 25: change to "…with the exception of the Atlantic section…"
changed
Page 4 line 27: change to " …and with the highest annual TP…"
changed
Page 6 line 2: change to "misrepresentation"
changed
Page 7 line 21: remove "as we found"
removed
Page 11 line 28: Not clear why simplified snow and ice thickness in both reanalyses would lead to larger warm bias in ERA5 compared to ERA-I?
We removed the sentence of "It may also be partly attributed to the simplified representation of snow and ice thickness in the reanalyses". Simplified snow and ice thickness may affect the surface heat budget, for example, overestimated conductive heat flux. But without full study, we will not give such a claim in this study.
Page 12 line 21: change to "The effects on ice growth are very small (one the order of the centimeters) during the freezing period.
Changed

Your truly
Caixin Wang
On behalf of all co-authors